# Fumarate induces vesicular release of mtDNA to drive innate immunity

Vincent Zecchini[1,16], Vincent Paupe[2,16], Irene Herranz-Montoya[1,8], Joëlle Janssen[1,9], Inge M. N. Wortel[1,10], Jordan L. Morris[2], Ashley Ferguson[1], Suvagata Roy Chowdury[2], Marc Segarra-Mondejar[1,11], Ana S. H. Costa[1,12], Gonçalo C. Pereira[2], Laura Tronci[1,13], Timothy Young[1], Efterpi Nikitopoulou[1], Ming Yang[1,11], Dóra Bihary[1,14], Federico Caicci[3], Shun Nagashima[2,15], Alyson Speed[1], Kalliopi Bokea[4], Zara Baig[5], Shamith Samarajiwa[1], Maxine Tran[4], Thomas Mitchell[6,7], Mark Johnson[2], Julien Prudent[2,17 ✉] & Christian Frezza[1,11,17 ✉]

Mutations in fumarate hydratase (FH) cause hereditary leiomyomatosis and renal cell carcinoma[1]. Loss of FH in the kidney elicits several oncogenic signalling cascades through the accumulation of the oncometabolite fumarate[2]. However, although the long-term consequences of FH loss have been described, the acute response has not so far been investigated. Here we generated an inducible mouse model to study the chronology of FH loss in the kidney. We show that loss of FH leads to early alterations of mitochondrial morphology and the release of mitochondrial DNA (mtDNA) into the cytosol, where it triggers the activation of the cyclic GMP–AMP synthase (cGAS)–stimulator of interferon genes (STING)–TANK-binding kinase 1 (TBK1) pathway and stimulates an inflammatory response that is also partially dependent on retinoic-acid-inducible gene I (RIG-I). Mechanistically, we show that this phenotype is mediated by fumarate and occurs selectively through mitochondrial-derived vesicles in a manner that depends on sorting nexin 9 (SNX9). These results reveal that increased levels of intracellular fumarate induce a remodelling of the mitochondrial network and the generation of mitochondrial-derived vesicles, which allows the release of mtDNA in the cytosol and subsequent activation of the innate immune response.

Fumarate hydratase (FH), a metabolic enzyme of the tricarboxylic acid (TCA) cycle that catalyses the reversible conversion of fumarate to malate, has been identified as a bona fide tumour suppressor[2]. Loss of FH predisposes to hereditary leiomyomatosis and renal cell carcinoma (HLRCC), a syndrome that is characterized by an aggressive form of kidney cancer. A hallmark of both FH-deficient cells and tumours is the aberrant accumulation of fumarate[3], which has been shown to drive malignant transformation and tumour progression through the activation of a series of oncogenic cascades[2]. However, when analysed in tumours or other cellular models, all of these signalling cascades are already simultaneously activated and are likely to be intertwined, which makes it difficult to elucidate their hierarchy, cross-talk and causal contribution to the disease.

Here we generated an inducible transgenic mouse model to investigate the chronology of the loss of *Fh1*, the mouse orthologue of human *FH*, in the adult kidney. By crossing a previously generated *Fh1*[fl/fl] strain[4] with mice that contain a ubiquitously expressed tamoxifen-inducible Cre recombinase at the *Rosa26* locus[5], we obtained a tamoxifen-inducible *Fh1*[fl/fl] strain (Fig. 1a). Treating *Fh1*[fl/fl] mice with tamoxifen resulted in the efficient excision of *Fh1* exons 3 and 4 (hereafter termed *Fh1*[−/−]), compared to the control strain (*Fh1*[+/+]) (Fig. 1b and Extended Data Fig. 1a,b). Loss of FH causes profound metabolic changes, including the aberrant intracellular accumulation of argininosuccinate[6] and of products of the reaction by which fumarate is added to the thiol groups of free cysteine and to the thiol groups of proteins[7], generating *S*-(2-succinyl)cysteine (2SC) and succinated proteins, respectively. Both reactions buffer the excess of fumarate production, which eventually occurs when these and other buffering systems have reached capacity (Extended Data Fig. 1c). We observed an early increase of 2SC at day 5, followed by an increase of all these metabolic markers, including fumarate, at day 10 in *Fh1*[−/−] kidneys (Fig. 1c). Macroscopically, there were no gross morphological differences between wild-type and *Fh1*-deficient kidneys (Fig. 1d).

[1]Medical Research Council Cancer Unit, University of Cambridge, Cambridge, UK. [2]Medical Research Council Mitochondrial Biology Unit, University of Cambridge, Cambridge, UK. [3]Department of Biology, University of Padova, Padova, Italy. [4]Department of Surgical Biotechnology, Division of Surgery and Interventional Science, UCL, London, UK. [5]Division of Infection and Immunity, Institute of Immunity and Transplantation, UCL, London, UK. [6]Wellcome Sanger Institute, Wellcome Genome Campus, Hinxton, UK. [7]Department of Surgery, University of Cambridge, Cambridge, UK. [8]Present address: Molecular Oncology Programme, Growth Factors, Nutrients and Cancer Group Centro Nacional de Investigaciones Oncológicas (CNIO), Madrid, Spain. [9]Present address: Human and Animal Physiology, Wageningen University and Research, Wageningen, The Netherlands. [10]Present address: Department of Data Science, Institute for Computing and Information Sciences, Radboud University, Nijmegen, The Netherlands. [11]Present address: CECAD Research Centre, University of Cologne, Cologne, Germany. [12]Present address: Matterworks, Somerville, MA, USA. [13]Present address: Cogentech SRL Benefit Corporation, Milan, Italy. [14]Present address: VIB KU Leuven Center for Cancer Biology, Leuven, Belgium. [15]Present address: Laboratory of Regenerative Medicine, School of Life Sciences, Tokyo University of Pharmacy and Life Sciences, Tokyo, Japan. [16]These authors contributed equally: Vincent Zecchini, Vincent Paupe. [17]These authors jointly supervised this work: Julien Prudent, Christian Frezza. ✉e-mail: julien.prudent@mrc-mbu.cam.ac.uk; christian.frezza@uni-koeln.de

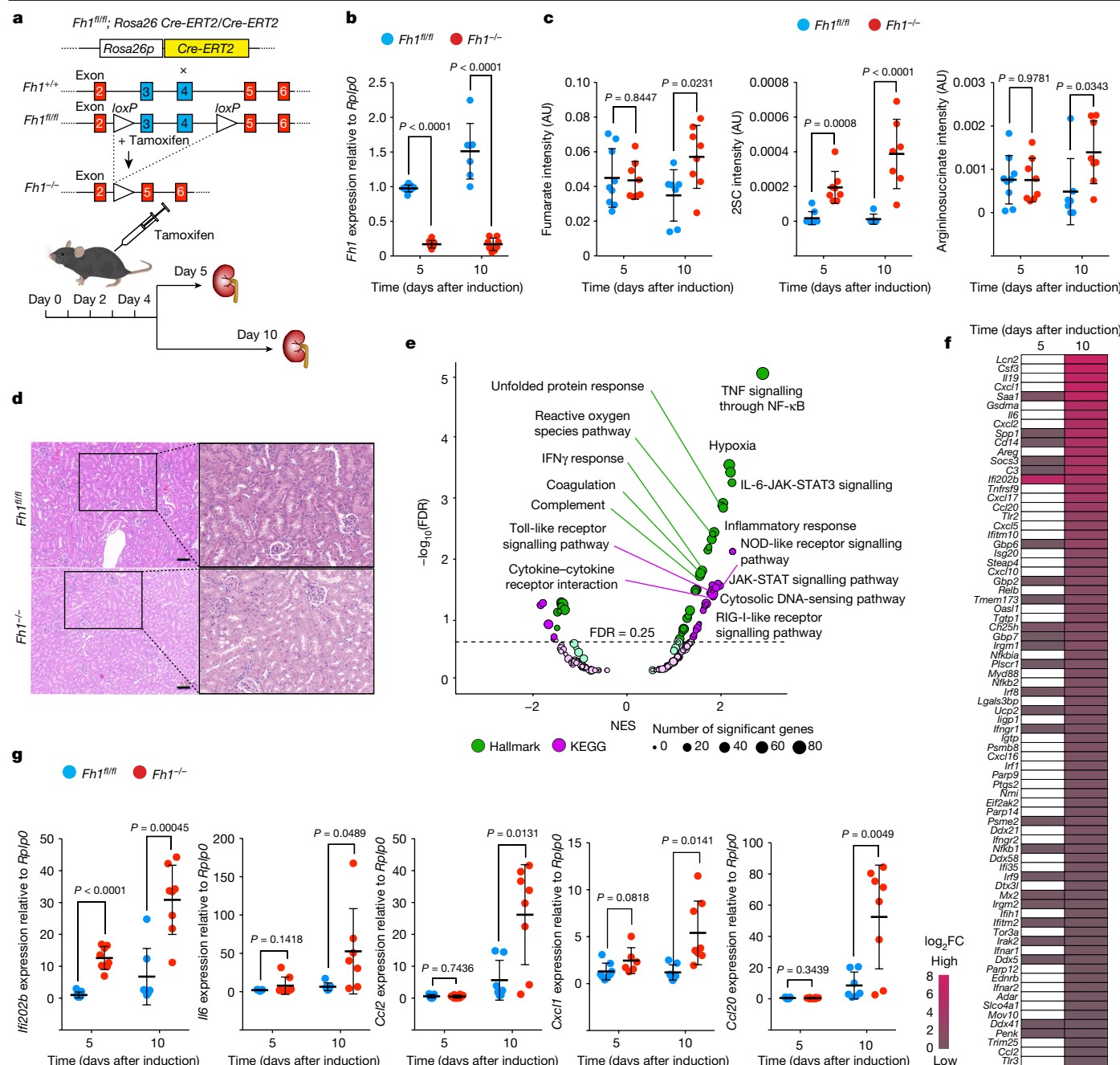

**Fig. 1 | Loss of *Fh1* in the adult mouse kidney triggers an early inflammatory response. a**, Genome-editing strategy for generating inducible *Fh1* knockout alleles. *Rosa26-Cre-ERT2* mice carry the tamoxifen-responsive *Cre* recombinase transgene downstream of the *Rosa26* promoter[5]. *Fh1*^fl/fl mice[13] contain two *loxP* sites flanking *Fh1* exons 3 and 4. Intraperitoneal injection of tamoxifen in the adult mouse induces the nuclear translocation of the ubiquitously expressed Cre–ERT2 fusion protein, resulting in the excision of the genomic fragment located between the *loxP* sites to generate *Fh1* null alleles (*Fh1*^−/−). Mice of about 90 days of age are treated with tamoxifen and the kidneys are collected for analysis at day 5 and day 10 after injection. **b**, *Fh1* mRNA expression levels in wild-type control (*Fh1*^+/+) and *Fh1*-deficient (*Fh1*^−/−) adult mouse kidneys, measured by qRT–PCR. **c**, Metabolite abundance (normalized peak ion intensity, arbitrary units (AU)) in *Fh1*^+/+ and *Fh1*^−/− adult mouse kidney measured by liquid chromatography–mass spectrometry (LC–MS). **d**, Haematoxylin and eosin (H&E) staining of *Fh1*^+/+ and *Fh1*^−/− adult mouse kidney at day 10 after induction. Scale bars, 100 μm. **e**, Volcano plots of the GSEA, highlighting the differentially regulated pathways in *Fh1*^−/− versus *Fh1*^+/+ kidney tissue at day 10 after induction. FDR, false discovery rate; NES, normalized enrichment score. **f**, Heat map showing upregulated inflammation-related genes in *Fh1*^−/− versus *Fh1*^+/+ kidney tissue at day 5 and day 10 after induction (white cells mean no difference between the two groups). **g**, Expression levels of ISGs in *Fh1*^−/− versus *Fh1*^+/+ kidney tissue at day 5 and day 10 after induction, measured by qRT–PCR. Data are mean ± s.e.m. **b**,**c**,**g**, $n$ = minimum 8 mice in each group, Student's *t*-test corrected for multiple comparisons with the Holm–Sidak method.

## Loss of *Fh1* triggers an inflammatory response

To investigate the acute response to the loss of *Fh1*, we performed a transcriptomic analysis of the mouse kidney (Fig. 1a and Supplementary Table 1). The most prevalent altered biological processes were linked to pathways that are involved in inflammation and the innate immune response (Fig. 1e and Supplementary Tables 2 and 3). We also observed an increase in interferon-stimulated genes (ISGs), with *Ifi202b* being the most upregulated transcript (Fig. 1f, Extended Data Fig. 1d and Supplementary Table 1), and found that this was associated with an

upregulation of pro-inflammatory cytokines and chemokines (Fig. 1f,g). Deconvolution of the transcriptomics data revealed that immune cells contributed to a low fraction of the signal that was observed (Extended Data Fig. 1e). In addition, although we confirmed the presence of succinated proteins, there were no changes in the expression of CD14 (a marker of immune cells) in *Fh1*[−/−] kidney tissue (Extended Data Fig. 1f), ruling out infiltrating immune cells as the cause of the immune signature observed.

To confirm the cell-autonomous nature of the phenotype, we generated epithelial cell lines from *Fh1*[fl/fl] mouse kidneys[8], hereafter referred to as inducible *Fh1* clones 29 or 33 (iFh1[−/−CL29] or iFh1[−/−CL33] after recombination and *Fh1* loss) (Extended Data Fig. 2a). Treating iFh1[fl/flCL29] and iFh1[fl/flCL33] cells with 4-hydroxytamoxifen (4-OHT) resulted in a time-dependent decrease in the mRNA and protein levels of *Fh1*, accompanied by reductions in the mitochondrial membrane potential and respiration; it also induced an early increase in the metabolic markers 2SC and argininosuccinate, followed by the accumulation of fumarate, and transcriptional hallmarks of *Fh1* loss (Fig. 2a,b, Extended Data Fig. 2b–d and Supplementary Fig. 1a–d).

We then investigated the pathways involved in the activation of inflammation. The binding of conserved features of invading pathogens or that of host components released as a result of cell or tissue damage converges on a central kinase, TBK1, which in turn activates interferon-regulatory factors (IRFs) and leads to the production of pro-inflammatory cytokines, chemokines and type I and III interferons (IFNs)[9] (Extended Data Fig. 2e). Consistent with the activation of TBK1, we observed a time-dependent phosphorylation of IRF3 and the downstream effector STAT1 in iFh1[−/−CL29] cells, accompanied by an upregulation of STING (Fig. 2c) and an increase in the transcription of ISGs in both iFh1[−/−] clones (Extended Data Fig. 2f and Supplementary Fig. 1e).

Altogether, these results show that the loss of *Fh1* triggers a cell-autonomous innate immune response that results in the upregulation of ISGs both in vivo and in vitro.

## Loss of *Fh1* induces the release of cytosolic mtDNA

Previous studies have shown that mtDNA and mitochondrial RNA (mtRNA) have a role as potent activators of nucleic-acid-sensing pathways, leading to the activation of different inflammatory pathways[10]. We observed a time-dependent increase in the number of cells containing cytosolic DNA foci, which matched the dynamics of the loss of *Fh1* expression (Fig. 2d–f and Supplementary Fig. 1f,g) and coincided with a progressive remodelling of mitochondrial morphology that was characterized by swollen and swollen-elongated mitochondria (Fig. 2d, Extended Data Fig. 2g–i and Supplementary Fig. 1h–j). A transmission electron microscopy (TEM) analysis of *Fh1*-deficient kidneys (Fig. 2g) and other mouse (Extended Data Fig. 2j) and human[11] FH-deficient lines identified similar anomalies in mitochondrial ultrastructure. Therefore, we hypothesized that the source of the DNA foci could be mtDNA leaking into the cytosol. To test this, we quantified the mtDNA copy number in isolated cytosolic fractions using droplet digital PCR (ddPCR). We found a time-dependent increase in mtDNA copy number in the cytosol of both iFh1[−/−CL29] and iFh1[−/−CL33] cell lines (Fig. 2h–j and Supplementary Fig. 1k–m).

To determine whether this phenotype was maintained over time, we resorted to the chronic model of *Fh1* loss that we previously generated[12] (hereafter termed cFh1[fl/fl] for the control line and cFh1[−/−CL1] and cFh1[−/−CL19] for the two Fh1-deficient clones) (Supplementary Fig. 2a). Similar to iFh1[−/−] cells, both cFh1[−/−] cell lines exhibited anomalies in mitochondrial morphology (Extended Data Fig. 3a–e and Supplementary Fig. 2b). In addition, both cFh1[−/−] cell lines showed the release of mtDNA into the cytosol, activation of the STING–TBK1–IRF3 cascade and upregulation of the expression of ISGs (Extended Data Fig. 3f–l). All aspects of this phenotype were fully rescued by the re-expression of exogenous *Fh1* (Extended Data Fig. 3c–l). Together, these results indicate that the loss of *Fh1* induces a chronic release of mtDNA into the cytosol, which triggers a persistent inflammatory response.

We also showed that cells deficient for the B subunit of succinate dehydrogenase (SDH)[13], another TCA-cycle enzyme, exhibited no significant enrichment in immune-related processes, no major changes in mitochondrial morphology, low levels of cytosolic mtDNA and low transcription levels of *Ifnb1*, ISGs and *Hmox1* (Supplementary Fig. 3a–k). Furthermore, expressing NDI1 (ref. [14]) to restore the defects in the electron transport chain that were observed in *Fh1*-deficient cells[15] did not rescue any phenotype (Supplementary Fig. 4).

Together, these results suggest that the loss of FH activity, rather than an overall impairment of the TCA cycle or mitochondrial metabolism, is responsible for the activation of the inflammatory response.

## Fumarate phenocopies *Fh1* loss

We then tested whether fumarate could be the main driver behind this effect (Extended Data Fig. 1c), by treating cFh1[fl/fl] cells with a cell-permeable derivative of fumarate, monomethyl fumarate (MMF), which is known to increase the cellular levels of fumarate without substantially affecting its reactivity[16] (Supplementary Fig. 5a). Treatment with MMF phenocopied the effects of *Fh1* loss, inducing the dose- and time-dependent expression of hallmarks of *Fh1* loss, 2SC and fumarate at levels similar to those in iFh1[−/−] cells, early mitochondrial accumulation of succinated proteins, mitochondrial network remodelling, cytosolic mtDNA release, activation of the TBK1 cascade and expression of ISGs (Fig. 3a–f and Extended Data Figs. 4 and 5). Expression of cytosolic *Fh1*, which partially restores the oxygen consumption rate (OCR) and the levels of fumarate, but still leads to increased mitochondrial succination, only partially rescued the phenotype (Extended Data Fig. 6), suggesting that increased mitochondrial fumarate levels are primarily responsible for mtDNA release. Treatment with a cell-permeable form of succinate, which accumulates in SDH-deficient cells and tissues and to a much lesser extent also in FH-deficient conditions[17], had no such effect (Extended Data Fig. 7a–d and Supplementary Fig. 5b–f). Similarly, treatment with the TCA-cycle-derived metabolites α-ketoglutarate or 2-hydroxyglutarate had no effect compared to MMF treatment (Extended Data Fig. 7e–g and Supplementary Fig. 5g).

To further confirm the role of mtDNA release in our model, we generated mtDNA-depleted iFh1[fl/flCL29ρ0] (that is, *Fh1*[+/+ρ0]) cells (Extended Data Fig. 8a). MMF-treated *Fh1*[+/+ρ0] cells did not show any increase in phosphorylated TBK1 (pTBK1) and pIRF3 (Fig. 3g), or other downstream ISG targets (Extended Data Fig. 8b).

These results show that fumarate and cytosolic mtDNA are the trigger and the ensuing stimulus, respectively, that account for the activation of the innate immune response that is observed in *Fh1*-deficient cells.

## Fumarate induces activation of cGAS and RIG-I

The presence of cytosolic mtDNA and the activation of the TBK1–IRF3 cascade in *Fh1*-deficient cells suggests that the cGAS–STING pathway might be involved in eliciting the observed inflammatory phenotype. We observed increased cytosolic cGAS in MMF-treated cells (Extended Data Fig. 8c,d) and showed that the pharmacological inhibition of cGAS in MMF-treated cells, iFh1[−/−CL29] and cFh1[−/−CL1] cells using RU.521 (ref. [18]) lowered the expression levels of ISGs (Extended Data Fig. 8e–g). The in vivo pharmacological inhibition of STING also partially rescued the upregulation of ISGs (Fig. 3h and Extended Data Fig. 8h), indicating that the cGAS–STING pathway is a crucial part of the inflammatory response induced both in vitro and in vivo.

Notably, we found that silencing the cytosolic RNA sensor RIG-I (*Rigi*), but not its adaptor protein *Mavs* or the RNA sensor *Mda5* (also known as *Ifih1*), also reduced the levels of TBK1 and IRF3 phosphorylation in iFh1[−/−CL29] cells (Extended Data Fig. 8i and Supplementary Fig. 6a). Similarly, although the individual silencing of *Sting1*, *Cgas* or *Rigi* in

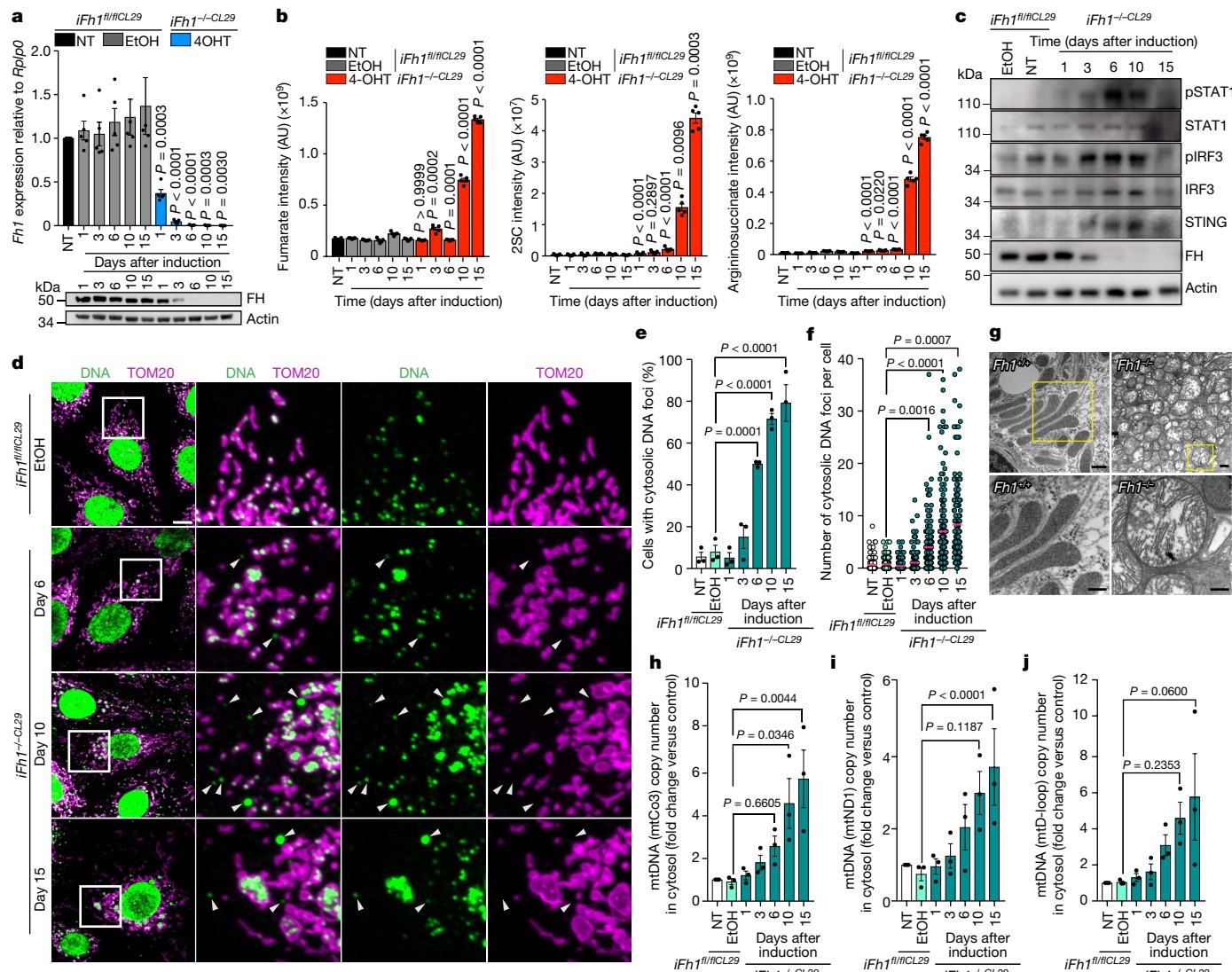

**Fig. 2 | Activation of the cGAS–STING pathway in *Fh1*-deficient cells is triggered by cytosolic mtDNA. a**, qRT–PCR (top) and immunoblots (bottom) showing the expression levels of *Fh1* (top) or FH protein (bottom) in inducible *iFh1* epithelial kidney cell lines clones 29 (*iFh1*^fl/flCL29^) that were not treated (NT) or were treated with either vehicle (ethanol; EtOH) or 4-OHT (*iFh1*^−/−CL29^) for the indicated period of time. *n* = 5 independent experiments. **b**, Relative abundance (normalized peak ion intensity) of fumarate (left), 2SC (middle) and argininosuccinate (right) in *iFh1*^CL29^ cells measured by LC–MS. *n* = 5 independent experiments. **c**, Immunoblots of specified proteins in *iFh1*^CL29^ cells. **d**, Representative confocal images of mitochondrial morphology (TOM20) and DNA foci (DNA) in *iFh1*^CL29^ cells. White arrowheads indicate cytosolic DNA foci.

Scale bar, 10 μm. **e**,**f**, Percentage of *iFh1*^CL29^ cells showing cytosolic DNA foci (**e**) and number of cytosolic DNA foci per cell (**f**) from **d**. *n* = 3 independent experiments. **g**, TEM images of mitochondria from *Fh1*^+/+^ and *Fh1*^−/−^ adult mouse kidney tissue. Scale bars, 1 μm (top); 500 nm (bottom). **h**–**j**, Quantification of mtDNA copy number by ddPCR using a mtCo3 (**h**), mtND1 (**i**) or mtD-loop (**j**) probe, from isolated cytosolic fractions of *iFh1*^CL29^ cells at day 1 to day 15 after induction. *n* = 3 independent experiments. Data are mean ± s.e.m. **a**,**b**, Student's *t*-test corrected for multiple comparisons with the Holm–Sidak method; *P* values are indicated above each condition and relative to the corresponding vehicle-treated control; **e**,**f**,**h**–**j**, one-way ANOVA with Tukey's multiple comparisons test.

MMF-treated cells had a mild effect, their simultaneous knockdown enhanced the rescue of TBK1 and IRF3 phosphorylation and ISG expression (Fig. 3i,j and Supplementary Fig. 6b).

Together, these results indicate that a complex network that includes at least two cytoplasmic nucleic-acid sensors—cGAS and RIG-I—activates the TBK1–IRF3-dependent response after the release of mtDNA into the cytosol.

## mtDNA is released through SNX9-dependent MDVs

To gain insight into the mechanism of mtDNA release mediated by FH deficiency, we used *cFh1*^−/−CL1^ cells to perform a targeted small interfering RNA (siRNA) screen. The double silencing of *Bax* and

*Bak1*, which are involved in mtDNA release[19], did not rescue cytosolic mtDNA release (Extended Data Fig. 9a–d and Supplementary Fig. 7). Of note, deletion of *Fh1* did not lead to the release of cytochrome *c* or the recruitment of BAX to mitochondrial membranes (Extended Data Fig. 9e,f). Knockdown of *Vdac1*, which has been shown to facilitate mtDNA release[20], was also unable to rescue the phenotype (Extended Data Fig. 9a–d and Supplementary Fig. 7). Knockdown of the main regulators of mitochondrial fusion and fission[21], or *Cgas*, *Sting1* and *Rigi*, did not affect mtDNA release (Extended Data Fig. 9a–d and Supplementary Fig. 7). Finally, the levels of TFAM (transcription factor A, mitochondrial) were unchanged in our models (Supplementary Fig. 8), ruling out a role for TFAM depletion in fumarate-dependent mtDNA release[22].

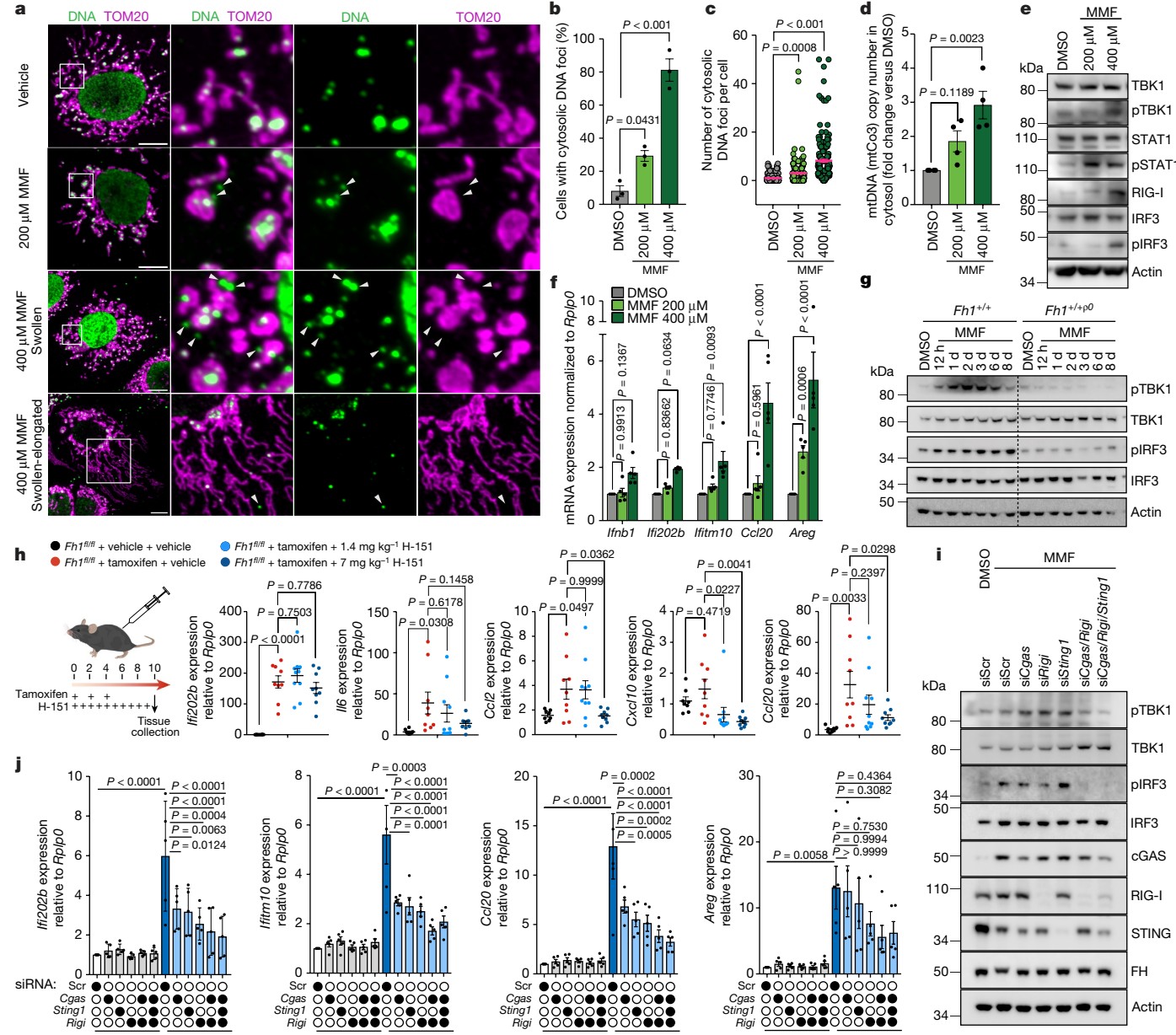

**Fig. 3 | Fumarate induces a remodelling of mitochondrial morphology and the release of mtDNA. a–f,** Chronic *Fh1* kidney cells (*cFh1^fl/fl^*) were treated with 200 μM or 400 μM MMF or vehicle (dimethyl sulfoxide; DMSO) for 8 days. *n* = 3 independent experiments (unless otherwise specified). In **g**, *iFh1^fl/flCL29^* cells were treated with 400 μM MMF or DMSO for 8 days. **a,** Representative confocal images of mitochondrial morphology (TOM20) and DNA foci (DNA) in *cFh1^fl/fl^* cells. White arrowheads indicate cytosolic DNA foci. Scale bars, 10 μm. **b,c,** Percentage of *cFh1^fl/fl^* cells showing cytosolic DNA foci (**b**) and number of cytosolic DNA foci per cell (**c**) from **a**. **d,** Quantification of mtDNA copy number by ddPCR using a mtCo3 probe, from isolated cytosolic fractions of *cFh1^fl/fl^* cells. **e,** Immunoblots of specified proteins in *cFh1^fl/fl^* cells. **f,** ISG expression in

*cFh1^fl/fl^* cells measured by qRT–PCR. *n* = 5 independent experiments. **g,** Immunoblots of specified proteins in *iFh1^fl/flCL29^* and mtDNA-depleted *iFh1^fl/flCL29ρ0^* cells. **h,** mRNA expression of a panel of ISGs in mouse kidney tissue treated with the STING inhibitor H-151, measured by qRT–PCR. *n* = 9 mice per group. **i,j,** Immunoblots of specified proteins (**i**) and ISG expression measured by qRT–PCR (**j**) in *cFh1^fl/fl^* cells treated with 400 μM MMF or DMSO for 8 d, and transfected with the indicated siRNAs (Scr, scramble). *n* = 5 independent experiments. Data are mean ± s.e.m. **b–d,f,j,** One-way ANOVA with Tukey's multiple comparison test; **h,** one-way ANOVA with Dunnett's multiple comparison test.

As an alternative way to communicate with other organelles and transport mitochondrial content without affecting the integrity of membranes, mitochondria can generate small-vesicle carriers known as mitochondrial-derived vesicles (MDVs)[23]. TEM imaging of *Fh1^−/−^* mouse kidneys revealed double-membrane-bound, low-density-content vesicles protruding from mitochondria, similar to budding MDVs[24] (Fig. 4a). The role of SNX9 (an endocytic accessory protein) and RAB9 in the release of MDVs was previously reported[25]. Here, the silencing of *Snx9*, but not *Rab9*, in

significantly reduced the release of mtDNA in both *cFh1^−/−CL1^* and *cFh1^−/−CL19^* cells (Extended Data Figs. 9a–d and 10a–c and Supplementary Fig. 7). Loss of *Snx9* also decreased the activation of TBK1–IRF3 phosphorylation, without restoring defects in mitochondrial morphology (Extended Data Fig. 10d–f). Finally, these results were confirmed in *iFh1^−/−^* cells at 10 and 15 days after induction (Extended Data Fig. 10g–l and Supplementary Fig. 9a). Together, these results indicate that SNX9 has a crucial role in the inflammatory response that is induced by cytosolic mtDNA release.

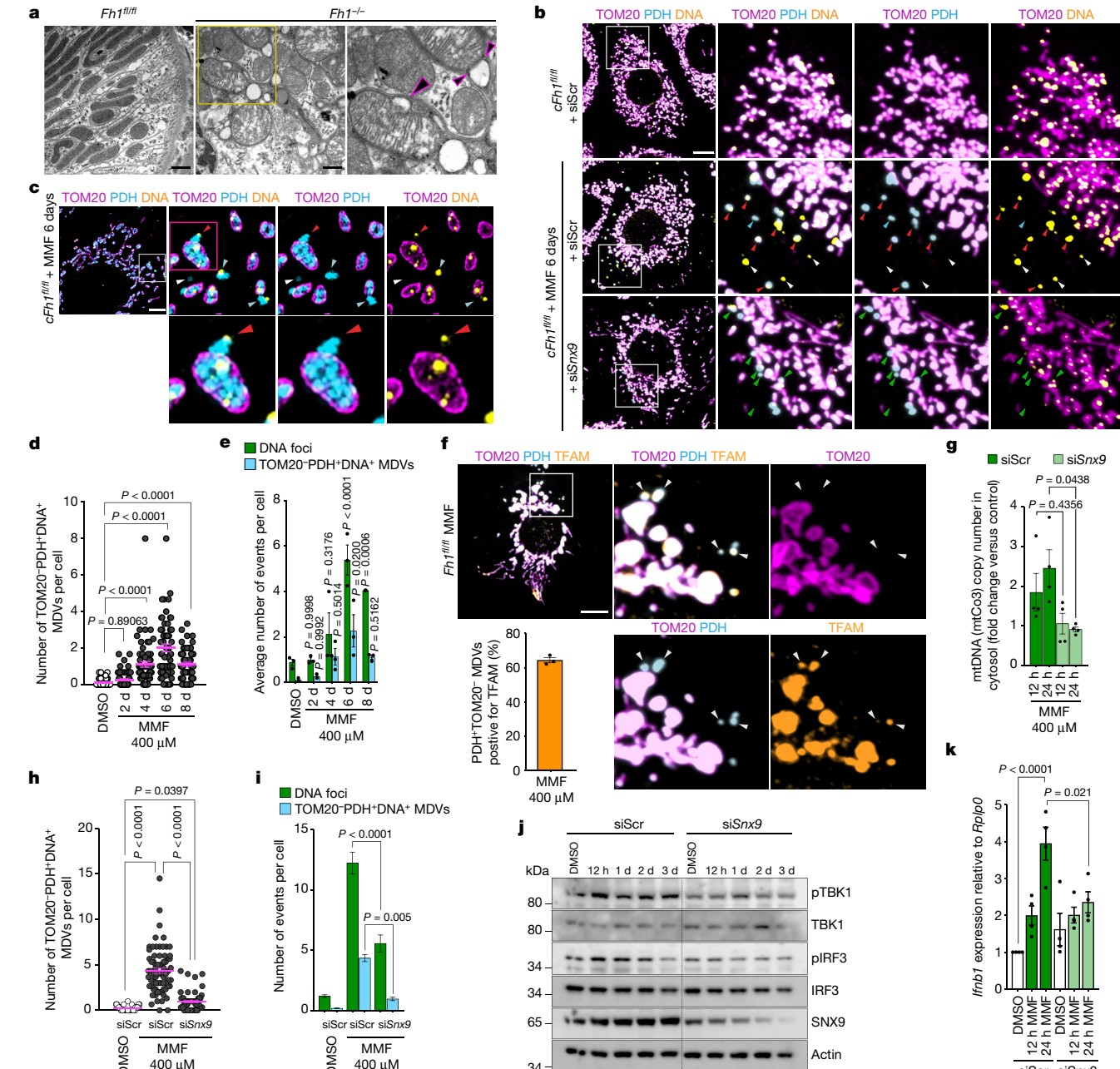

**Fig. 4 | mtDNA is conveyed to the cytosol by MDVs.** For all panels: $n = 3$ independent experiments (unless otherwise specified). In **b**,**c**,**f**,**h**,**i**, $cFh1^{fl/fl}$ cells were treated with 400 μM MMF or vehicle (DMSO) for 6 days. For immunofluorescence experiments, mitochondria or MDVs were labelled using anti-TOM20, anti-PDH and anti-TFAM antibodies and DNA with an anti-DNA antibody. **a**, TEM images of $Fh1^{+/+}$ and $Fh1^{-/-}$ mouse kidney showing potential vesicle budding at the mitochondrial surface (black arrowheads). Scale bars, 1 μm. **b**, Representative confocal images of $cFh1^{fl/fl}$ cells transfected with the indicated siRNAs. Red arrowheads: TOM20⁻PDH⁺DNA⁺ MDVs; blue arrowheads: TOM20⁻PDH⁺DNA⁻ MDVs; white arrowheads: cytosolic DNA foci; green arrowheads: stalled TOM20⁻PDH⁺DNA⁺ MDV budding events. Scale bar, 10 μm. **c**, Representative N-structured illumination microscopy (N-SIM) super-resolution images of MMF-treated $cFh1^{fl/fl}$ cells. Red arrowheads: TOM20⁻PDH⁺DNA⁺ MDV budding event; blue arrowheads: TOM20⁻PDH⁺ DNA⁺-released MDVs. Scale bar, 5 μm. **d**,**e**, Quantification of TOM20⁻PDH⁺DNA⁺

MDVs (**d**), and compared to cytosolic DNA foci (**e**) in $cFh1^{fl/fl}$ cells; $P$ values are relative to the corresponding DMSO-treated control. **f**, Representative confocal images of MMF-treated $cFh1^{fl/fl}$ cells. White arrowheads: TOM20⁻PDH⁺TFAM⁺ MDVs. Scale bar, 10 μm. Bottom left, histogram showing the quantification of TOM20⁻PDH⁺ MDVs that are positive for TFAM. **g**, Quantification of mtDNA copy number by ddPCR using a mtCo3 probe from isolated cytosolic fractions of MMF-treated $cFh1^{fl/fl}$ cells (12 h or 24 h), and pre-transfected with the indicated siRNAs. $n = 4$ independent experiments. **h**,**i**, Quantification of TOM20⁻PDH⁺ DNA⁺ MDVs (**h**) and compared to cytosolic DNA foci (**i**) in MMF-treated $cFh1^{fl/fl}$ cells and pre-transfected with the indicated siRNAs. **j**,**k**, Immunoblots of specified proteins (**j**) and $Ifnb1$ mRNA expression measured by qRT–PCR (**k**) in $cFh1^{fl/fl}$ cells transfected with the indicated siRNAs and treated with MMF for the indicated period of time. Data are mean ± s.e.m. **d**,**g**–**i**,**k**, One-way ANOVA with Tukey's multiple comparisons test; **e**, two-way ANOVA with Tukey's multiple comparisons test.

One of the hallmarks of MDVs is their cargo selectivity[23]. We observed an increase in vesicles that were positive for the matrix protein pyruvate dehydrogenase (PDH) but negative for the outer-mitochondrial-membrane protein TOM20 (TOM20⁻PDH⁺) in MMF-treated cells (Fig. 4b–e and Extended Data Fig. 11a–c). Notably, most of these vesicles also contained DNA (TOM20⁻PDH⁺DNA⁺) (Fig. 4d,e and

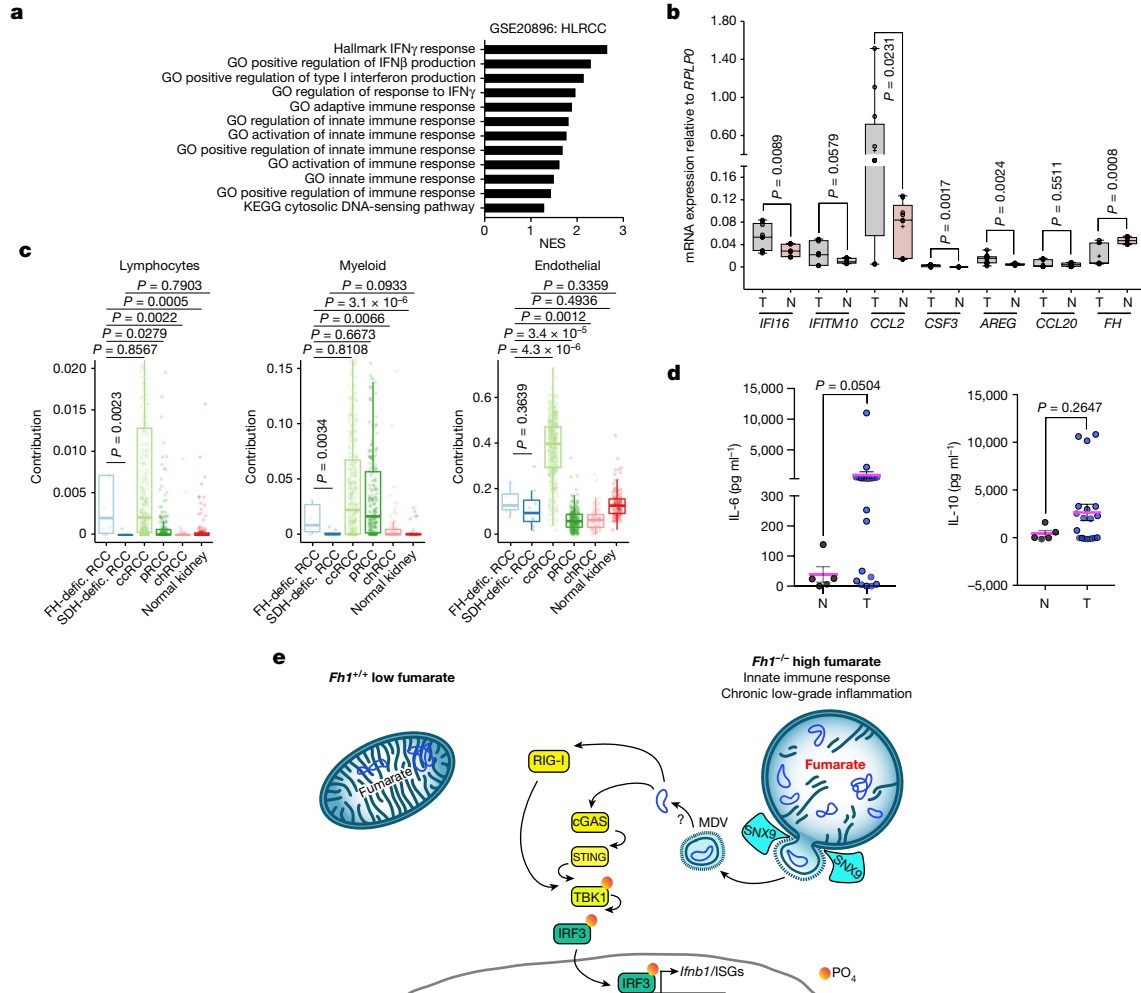

**Fig. 5 | FH-deficient tumour tissue is characterized by an inflammatory signature. a**, GSEA of the gene expression profile of HLRCC tumour versus normal tissue. **b**, mRNA expression of a panel of ISGs in HLRCC tumour versus normal tissue measured by qRT–PCR; N, normal (healthy individuals); T, tumour (patients with HLRCC). $n = 3$ serum samples for each N and T, in 3 technical replicates each. **c**, Cellular composition of the bulk RNA-sequencing datasets from *FH*-deficient RCCs, *SDH*-deficient RCCs and common RCC subtypes from TCGA (as described in the Methods). ccRCC, clear cell RCC; pRCC, papillary RCC; chRCC, chromophobe RCC. **d**, Levels of IL-6 and IL-10 in HLRCC tumour versus normal tissue measured by enzyme-linked immunosorbent assay

(ELISA). $n = 5$ and 20 samples for normal tissue (N) and *FH*-deficient HLRCC tumour tissue (T), respectively. **e**, Schematic of the pathway. Loss of *Fh1* in the mouse kidney results in profound metabolic adaptations and mitochondrial network remodelling mediated by fumarate within the cell. Increased levels of fumarate lead to the release of mtDNA in the cytosol through an SNX9-dependent MDV-mediated mechanism, which results in the activation of the cGAS–STING pathway as well as RIG-I, upregulation of genes involved in the innate immune response and chronic low-grade inflammation. Data are mean ± s.e.m. **b**, Unpaired two-tailed *t*-test; **c**, Wilcoxon rank sum test, $P < 0.05$ (Benjamini–Hochberg adjustment); **d**, unpaired two-tailed Mann–Whitney test.

Extended Data Fig. 11a,b). Microscopy analysis (Fig. 4f and Extended Data Fig. 11d) also revealed the presence of the nucleoid-associated protein TFAM inside these TOM20⁻PDH⁺ and TOM20⁻PDH⁺DNA⁺ MDVs, confirming the mitochondrial origin of the cytosolic DNA foci. Lattice structured illumination microscopy (SIM) analysis showed that these TOM20⁻PDH⁺DNA⁺ MDVs exhibited, on average, a Feret's diameter of 0.45 μm and an area of 0.2 μm² (Extended Data Fig. 11e–g and Supplementary Fig. 9b). We also confirmed the presence of TOM20⁻PDH⁺DNA⁺ MDVs from as early as day 3 after the loss of *Fh1* in *iFh1⁻/⁻CL29* cells (Extended Data Fig. 11h,i), corroborating the early activation of the ISG response that was observed in this inducible model. Finally, treating *Fh1⁺/⁺ρ0* cells with MMF did not reveal the presence of TOM20⁻PDH⁺ MDVs (Extended Data Fig. 12a,b), indicating that the presence of mtDNA is necessary for the formation of MDVs triggered by fumarate.

Silencing *Snx9* before treatment with MMF resulted in a reduction not only in mtDNA release in the cytosol (Fig. 4b,g and Extended

Data Fig. 12c), but also in the number of TOM20⁻PDH⁺DNA⁺ MDVs in the cytosol (Fig. 4h,i), and was accompanied by the detection of arrested budding vesicles at the mitochondrial membrane (Fig. 4b). Finally, blocking the release of MDVs also affected the activation of the nucleic-acid-sensing pathway, as silencing *Snx9* mitigated the activation of TBK1–IRF3 and the expression of *Ifnb1* and ISGs that was observed after treatment with MMF (Fig. 4j,k and Extended Data Fig. 12d).

Altogether, these results indicate that the accumulation of fumarate drives the SNX9- and MDV-dependent release of mtDNA into the cytosol to trigger innate immunity.

## Loss of FH causes inflammation in renal cancer

We next investigated whether this inflammatory response could be relevant in patients with HLRCC. Gene set enrichment analysis (GSEA) of a published HLRCC gene expression dataset revealed a

chronic inflammatory response, with an activation of innate immunity and DNA-sensor pathways similar to that observed in vivo (Fig. 5a), and quantitative PCR with reverse transcription (qRT–PCR) showed an upregulation of the same key markers in *FH*-deficient tumours (Fig. 5b). We also estimated the cellular composition of bulk RNA-sequencing datasets of *FH*-deficient renal cell carcinomas (RCCs), *SDH*-deficient RCCs and common RCC subtypes from The Cancer Genome Atlas (TCGA) by deconvolution of the transcriptomics data (Fig. 5c). This showed that there were significant differences in the contributions of major cell lineages, with the contribution of leukocytes being higher in *FH*-deficient RCCs than in normal tissue and other subtypes of RCCs. In line with our data, this contribution was also lower in *SDH*-deficient RCCs (Fig. 5c). Finally, we also found that *FH*-deficient tumour tissue exhibited increased levels of interleukin-6 (IL-6), but not IL-10, confirming the presence of an inflammatory milieu in these tumour tissues (Fig. 5d). These results suggest that the immune phenotype associated with loss of *Fh1* that we observed in mice may be relevant to human tissue that lacks *FH*.

## Discussion

In this work, we have shown that fumarate results in a remodelling of the mitochondrial network, the release of mtDNA into the cytosol and the activation of the innate immune response (Fig. 5e). Although this response emerges quickly, it persists over time, and is also seen in patients with HLRCC, suggesting that there is a direct interplay between mtDNA release, chronic inflammation and tumorigenesis in HLRCC.

Our results reveal that in addition to cGAS, RIG-I, which can recognize short viral double-stranded RNA and DNA[26] and is linked to the STING sensing pathway[27], also has a role. However, the silencing of *Mavs* had no effect on TBK1–IRF3 signalling, suggesting that RIG-I has a non-canonical function.

Although VDAC1 oligomerization[20], the opening of the mitochondrial permeability transition pore[28] and BAX and BAK pores[19] have been proposed to allow mtDNA release, we found that mtDNA is transported to the cytosol through SNX9-dependent MDVs, an alternative mechanism that preserves membrane integrity (Fig. 5e).

Overall, this study expands our understanding of how mitochondria can affect innate immune responses and provides a basis for further investigations of metabolite-driven immunopathology.

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

# Methods

## Mice

Mice were of mixed genetic background C57BL/6 and 129/SvJ. Mice were bred and maintained under specific-pathogen-free conditions at the Breeding Unit (BRU) at the CRUK Cambridge Institute. *Fh1[fl/fl]* (ref. [4]) and *R26-Cre-ERT2* (ref. [5]) mice were gifts from E. Gottlieb and D. Winton, respectively. Experimental mice were homozygous for the conditional LoxP-exon3/4-LoxP *Fh1* allele and expressed the Cre recombinase–ERT2 fusion under control of the *Rosa26* promoter (*Fh1[fl/fl];R26 CreERT2/CreERT2*). Littermate controls lacked the LoxP-exon3/4-LoxP allele but also expressed the *CreERT2* allele under the control of the *Rosa26* promoter (*Fh1[+/+]; R26 CreERT2/CreERT2*). Control mice were induced and euthanized at the same time as their experimental littermates. In vivo experiments (tamoxifen induction) were performed under specific-pathogen-free conditions at the BRU at the CRUK Cambridge Institute. All mouse experiments were performed in individually ventilated cages under the Animals (Scientific Procedures) Act 1986 (project licence P8A516814). The experiments were not randomized, and investigators were not blinded to treatment status during experiments and outcome assessment. Mice of both sexes were used and the sample size was initially set at 10 per group on the basis of previous experiments to account for potential losses during experiments and allow statistical analysis.

## Tamoxifen and H-151 preparation and treatment

In brief, 5 ml ethanol was added slowly to 1 g of tamoxifen (Sigma-Aldrich) in a 50-ml Falcon tube. The tamoxifen and ethanol mix was sonicated at 40% amplitude in 20-s pulses until the tamoxifen was completely dissolved. Then, 50 ml of corn oil (Sigma-Aldrich) pre-heated at 50 °C was added immediately to the tamoxifen and ethanol mix to obtain a 20 mg ml$^{-1}$ tamoxifen stock solution. The tube was then vortexed and incubated for up to 12 h in an orbital shaker at 50 °C to ensure adequate solubilization. Aliquots of 5 ml were then stored indefinitely at −20 °C. H-151 (InvivoGen) was diluted in DMSO to obtain 20 mg ml$^{-1}$ or 4 mg ml$^{-1}$, and was further diluted in sterile phosphate-buffered saline (PBS) to obtain a 5% DMSO concentration before injection into mice. Aliquots of 2 ml were then stored at −20 °C. Before injection, the mixture was allowed to warm up to room temperature. Age-matched mice (between 10 and 12 weeks old) were used in all experiments. Each mouse received three doses of 2 mg tamoxifen by intraperitoneal injection. Tamoxifen was administered every other day to allow mice to recover between doses. Mice received a daily dose of 7 mg per kg or 1.4 mg per kg H-151 by intraperitoneal injection.

## Tissue collection

Mice were killed by cervical dislocation and the kidneys were speedily collected and processed for further analysis using electron microscopy, gene expression, mass spectrometry or staining (see below for specific applications).

## Electron microscopy

Tissue samples were fixed overnight at 4 °C in 2.5% glutaraldehyde (Sigma-Aldrich) and 2% paraformaldehyde (PFA; Polysciences) in 0.1 mol l$^{-1}$ sodium cacodylate buffer pH 7.4. The samples were post-fixed with 1% osmium tetroxide 1% (EMS) in 0.1 mol l$^{-1}$ sodium cacodylate buffer for 1 h at 4 °C. After three water washes, samples were dehydrated in a graded ethanol series and embedded in epoxy resin (Sigma-Aldrich). Ultrathin sections (60–70 nm) were obtained with an Ultrotome V (LKB) ultramicrotome, counterstained with uranyl acetate and lead citrate and viewed with a Tecnai G$^2$ (FEI) transmission electron microscope operating at 100 kV. Images were captured with a Veleta (Olympus Soft Imaging System) digital camera.

## Tissue processing for gene expression analysis

For gene expression analyses, kidneys were snap-frozen in liquid nitrogen and stored at −80 °C until processing for RNA extraction. In brief, approximately 10–20-mg chunks of snap-frozen kidney (either mouse or human tissue) were placed in PreCellys tubes (Stretton Scientific Ltd) containing 600 μl ice-cold RNALater solution (Qiagen AllPrep DNA/RNA) for homogenization at 5,500 rpm for 2 min and spun at maximum speed for 10 min in a refrigerated centrifuge to pellet cell debris. The supernatant was used for total RNA isolation using the AllPrep DNA/RNA Kit (Qiagen) following the manufacturer's instructions. For RNA sequencing, RNA was resuspended in water and the concentration was determined using a Qubit 2.0 fluorimeter (Life Technologies) following the manufacturer's instructions. RNA quality was determined using the RNA 6000 Nano Kit (Agilent) and Agilent 2100 system Bioanalyzer system following the manufacturer's instructions.

## RNA sequencing

RNA sequencing was performed at the CRUK Cambridge Institute Genomics Core Facility using the Illumina Truseq stranded mRNA (HT) kit for RNA-sequencing polyA capture on a Illumina HiSeq4000 sequencer.

Reads were mapped to the mouse reference genome GRCm38 with the STAR (v.2.6.0c) aligner[29]. Low-quality reads (mapping quality < 20) as well as known adapters and artefacts were filtered out using Cutadapt (v.1.10.0). Read counting was performed using the Bioconductor package Rsubread (v.1.28.1) (https://github.com/LTLA/csawUsersGuide) and gene annotations from GENCODE (release M17). Differential expression analysis was carried out with DESeq2 (v.1.18.1)[30]. The conditions were contrasted against the wild-type samples. Genes were identified as differentially expressed with a false discovery rate (FDR) cut-off of 0.01 and an absolute value of log$_2$FC (log$_2$ of the fold change) > 0.58.

## LC−MS for untargeted metabolomics analysis of mouse tissues

Snap-frozen tissue specimens were cut and weighed into PreCellys tubes (Stretton Scientific). An exact volume of extraction solution (30% acetonitrile, 50% methanol and 20% water) was added to obtain 40 mg specimen per ml of extraction solution. Tissue samples were lysed using a PreCellys 24 homogenizer (Stretton Scientific) at 5,500 rpm for 2 min. The suspension was mixed and incubated for 1 h at −20 °C, followed by 15 min at 4 °C in a thermomixer (Eppendorf), followed by centrifugation (16,000g, 15 min at 4 °C). The supernatant was collected and transferred into autosampler glass vials, which were stored at −80 °C until further analysis. To avoid bias due to machine drift, samples were randomized and processed blindly. A Dionex U3000 UHPLC system coupled to a Q Exactive mass spectrometer (Thermo Fisher Scientific) was used to perform the LC−MS analysis. Five microlitres of sample was injected onto a Sequant ZIC-pHILIC column (150 × 2.1 mm, 5 μm) and guard column (20 × 2.1 mm, 5 μm) (both Merck Millipore) for the chromatographic separation, as previously described[13]. The column oven temperature was maintained at 45 °C. The mobile phase was composed of 20 mM ammonium carbonate and 0.1% ammonium hydroxide in water (mobile phase A), and acetonitrile (mobile phase B). The flow rate was set at 0.2 ml per min with the following gradient: 80% B for 2 min, linear decrease to 20% of B 15 min. Both solvents were then brought back to the initial conditions and maintained for 8 min. The mass spectrometer was operated in full MS (50−750 *m/z*) and polarity switching mode. The acquired spectra were analysed using XCalibur Qual Browser and XCalibur Quan Browser software (Thermo Fisher Scientific) by referencing to an internal library of compounds.

## Metabolite analysis

LC−MS analysis was performed on a Q Exactive Orbitrap mass spectrometer coupled to a Dionex UltiMate 3000 Rapid Separation LC system (Thermo Fisher Scientific). The LC system was fitted with either

a SeQuant Zic-HILIC column (column A, 150 mm × 4.6 mm, 5 μm), or a SeQuant Zic-pHILIC (column B, 150 mm × 2.1 mm, 5 μm) with the corresponding guard columns (20 mm × 2.1 mm, 5 μm) (both from Merck Millipore). With column A, the mobile phase was composed of 0.1% aqueous formic acid (solvent A) and 0.1% formic acid in acetonitrile (solvent B). The flow rate was set at 300 μl per min and the gradient was as follows: 0–5 min 80% B, 5–15 min 30% B, 15–20 min 10% B, 20–21 min 80% B, hold at 80% B for 9 min. For column B, the mobile phase was composed of 20 mM ammonium carbonate and 0.1% ammonium hydroxide in water (solvent C), and acetonitrile (solvent D). The flow rate was set at 180 μl per min with the following gradient: 0 min 70% D, 1 min 70% D, 16 min 38% D, 16.5 min 70% D, hold at 70% D for 8.5 min. The mass spectrometer was operated in full MS and polarity switching mode. Samples were randomized to avoid machine drift, and were blinded to the operator. The acquired spectra were analysed using XCalibur Qual Browser and XCalibur Quan Browser software (Thermo Fisher Scientific) by referencing to an internal library of compounds. Calibration curves were generated using synthetic standards of the indicated metabolites.

## Tissue staining

For H&E staining, kidney tissue was fixed in a 10% neutral buffered formalin (NBF) solution for 24 h at room temperature. The tissue was then transferred to 70% ethanol and processed for paraffin embedding. Embedding, sectioning and H&E staining were performed at the Histopathology facility at CRUK Cambridge Institute. Slides were scanned using an Axioscan Z1 and analysed using Zen Blue.

For immunohistochemistry, paraffin sections were dewaxed in two xylene baths (5 min each) and rehydrated through a series of graded ethanol solutions (100%, 90% and 70%) (5 min each). Sections were placed in distilled water before heat-induced epitope retrieval (HIER) was performed. In brief, slides were immersed in IHC-Tek Epitope Retrieval Solution (IHC World) and placed in the IHC-Tek Epitope Retrieval Steamer Set (IHC World) for 45 min to unmask the antigens using the steaming method. Once completely cooled, the slides were gently rinsed with Tris-buffered saline containing 0.1% Tween-20 (TBST) wash buffer and endogenous peroxidase activity was blocked by incubating the slides with Peroxidase Block (Dako, Agilent) for 5 min. Slides were rinsed with TBST and placed in a fresh TBST bath (5 min) before incubating for 5 min with Protein Block (Dako, Agilent). Slides were then incubated for 2 h with the primary antibody at an appropriated dilution in Antibody Diluent (Dako, Agilent). Antibodies used: anti-CD14 (1:500, anti-rabbit, PA5-95334, Invitrogen, Thermo Fisher Scientific) and anti-2SC (1:500, anti-rabbit, crb2005017, Cambridge Research Biochemicals). Slides were rinsed with TBST and placed in three fresh TBST baths (5 min each) before incubating them with Labelled Polymer-HRP Anti-Rabbit (Dako, Agilent) for 2 h. After incubation, the slides were again rinsed with TBST and placed in three fresh TBST baths (5 min each). For the detection, the EnVision+ System-HRP(DAB) kit containing DAB chromogen concentrate (Dako, Agilent) was used for 10 min. The slides were gently rinsed with distilled water and counterstained in Mayer's Hematoxylin Solution (Sigma-Aldrich, Merck) for 10 min, followed by rinsing with tap water for 2 min. Slides were dehydrated through graded ethanol washes (70%, 90%, 100%) (5 min each) before mounting using Histomount (Thermo Fisher Scientific).

## Tissue lysates for immunoblotting

Tissue lysates were prepared in RIPA (50 mM Tris-HCl (pH 7.4), 150 mM NaCl, 1 mM EDTA, 1% Triton X-100, 0.5% Na-deoxycholate, 0.1% SDS in 1× PBS) buffer supplemented with phosphatases and protease inhibitor cocktails (Roche) as follows: approximately 20–30 mg of frozen tissue was homogenized in CK14/2ml PreCellys tubes for 2 min at 5,500 rpm in ice-cold RIPA buffer. Lysates were clarified by centrifugation at maximum speed for 20 min in a benchtop centrifuge (Eppendorf) at 4 °C. After clarification, the supernatant was transferred to a clean pre-chilled tube and the protein concentration was determined using the Pierce BCA Protein Assay Dye (Thermo Fisher Scientific) following the manufacturer's instructions.

## Genomic DNA amplification

Verification of the excision of exons 3 and 4 after tamoxifen recombination was performed by PCR on genomic DNA extracted from the tissues. In brief, genomic DNA was extracted using the Qiagen AllPrep DNA/RNA Kit following the manufacturer's instruction. PCR was performed on an Applied Biosystems Verity Thermo Cycler. Primers[12] were obtained from Sigma-Aldrich. PCR product was loaded on a 2% agarose gel and visualized with SybrGreen.

## Bioinformatics processing of data

The results of the differential expression analysis were visualized using volcano plots, representing the $\log_2$ of the fold change between the conditions and the adjusted $P$ values obtained. GSEA was performed using the GSEA software from Broad Institute. Pre-ranked GSEA with default settings was run to determine the enriched pathways in the different conditions versus controls, using DESeq2 metrics to rank the genes, and the Hallmark and KEGG pathway collections. The results are presented in volcano plots showing the NES and the FDR-adjusted $P$ values obtained.

## Estimates of kidney tumour composition by deconvolution of transcriptomics data

**Bulk RNA-sequencing datasets.** Bulk transcriptomic data for clear cell RCCs, papillary RCCs and chromophobe RCCs were downloaded from TCGA[31]. This dataset was further annotated for samples with germline *FH* mutational status[32]. Transcriptomic data for additional *FH*-deficient tumours sampled from primary and metastatic sites were also downloaded[33]. *FH*-deficient RCCs were defined as only those with a known germline mutation, and derived unequivocally from the primary kidney cancer. Samples that might have been derived from metastatic sites[33] were not classified as *FH*-deficient RCCs, as the microenvironment in metastases will differ from the primary site. In total, six tumours were unequivocally derived from primary tumour, and these were added to the four *FH*-deficient RCCs sequenced as part of the TCGA consortium. Finally, we added bulk transcriptomes from *SDH*-deficient RCCs[34].

**Single-cell-sequencing reference data.** Annotated single cells derived from adult RCCs were used as a comparative reference[35]. For the mouse data, we used a mouse single-cell RNA-sequencing reference dataset[36] as reference. Bulk RNA counts are from GSE183745. Methods as above.

**Deconvolution method.** We collated the reference single-cell RNA-sequencing data according to their major lineage (such as leukocytes, myeloid or endothelial), rather than using the finely annotated subtypes. The collation to major lineage types helped to alleviate batch effects and the difficulty in discerning the presence of subtly different cell subtypes from bulk RNA data. This single-cell reference was used as input alongside the bulk RNA-sequencing data above into a Bayesian deconvolution method to allow the cellular composition of the bulk data to be estimated[37]. We compared the cellular composition of the above bulk RNA-sequencing datasets (*FH*-deficient RCCs, *SDH*-deficient RCCs and common RCC subtypes from TCGA as described above).

## Cell culture and treatment

All cell lines were grown in high-glucose (4.5 g l⁻¹) Dulbecco's modified Eagle's medium (DMEM) containing 1 mM sodium pyruvate and 2 mM L-glutamine (Gibco, Thermo Fisher Scientific) supplemented with 10% fetal bovine serum (FBS) (Gibco, Thermo Fisher Scientific). Cells were tested for mycoplasma contamination using the Lookout Mycoplasma PCR detection kit (Sigma-Aldrich). To generate $FH^{+/+p0}$

cell lines, $FH^{+/+}$ cells were treated with 50 ng ml$^{-1}$ ethidium bromide for 25 days. Clones were isolated after treatment and grown in the same medium supplemented with 50 µg ml$^{-1}$ uridine (Sigma-Aldrich). MMF (Sigma-Aldrich) and dimethylsuccinate (DMS) (Sigma-Aldrich) powder were resuspended in DMSO (Thermo Fisher Scientific) at 500 mM and added to the medium at the indicated concentration for the indicated time. For long treatments, the medium was changed every two days. Cells were transfected with Lipofectamine RNA iMAX (Invitrogen) following the manufacturer's instructions, with the siRNA (Dharmacon) at a final concentration of 20 nM.

**siRNA oligonucleotides.** Non-targeting siRNA scramble (ON-TARGETplus SMARTpool, D-0001810-10-20), *Bak1* (ON-TARGETplus SMARTpool, L-042978-00-0005), *Bax* (ON-TARGETplus SMARTpool, L-061976-00-0005), *Ddx58* (*Rigi*) (ON-TARGETplus SMARTpool, L-065328-00-0005), *Drp1* (*Dnm1l*) (ON-TARGETplus SMARTpool, L-054815-01-0005), *Ifih1* (*Mda5*) (ON-TARGETplus SMARTpool, L-048303-00-0005), *Mavs* (ON-TARGETplus SMARTpool, L-053767-00-0005), *Mb21d1* (*Cgas*) (ON-TARGETplus SMARTpool, L-055608-1-0005), *Mfn1* (ON-TARGETplus SMARTpool, L-065399-01-0005), *Mfn2* (ON-TARGETplus SMARTpool, L-046303-00-0005), *Opa1* (ON-TARGETplus SMARTpool, L-042427-01-0005), *Rab9* (L-040861-01-0005), *Snx9* (ON-TARGETplus SMARTpool, L-057505-01-0005), *Tmem173* (*Sting1*) (ON-TARGETplus SMARTpool, L-055528-00-0005) and *Vdac1* (ON-TARGETplus SMARTpool, L-047345-00-0005).

## Plasmids and constructs
pEGFP and pEGFP:NDI1 were obtained from the laboratory of E. Dufour[38] and were stably expressed in $cFh1^{-/-CL1}$ cells. Lentiviral particles were generated in HEK293T cells by co-transfection of the target vector together with packaging psPAX2 (Addgene, 12260) and envelope pMD2.G (Addgene, 12259) vectors. Twenty-four hours after transduction, cells were selected for puromycin resistance. The *Fh1* reconstituted cell line ($Fh1^{-/-CL1}$+*pFH-GFP*) was generated as previously described[16]. The cytoplasmic *Fh1* cell line ($Fh1^{-/-CL1}$+*cytoFh1-GFP*) was generated by transfecting a previously generated cytoFh1-GFP plasmid[39] into cells seeded into a six-well plate using Lipofectamine 2000 (Thermo Fisher Scientific, 11668027) using 2.5 µg plasmid DNA and 9 µl Lipofectamine 2000 reagent per well. Selection was performed by incubating cells in the presence of 2 mg ml$^{-1}$ G418 (Gibco, 10131027) for two weeks.

## Antibodies
For immunoblots: goat polyclonal anti-FH/fumarase (ab113963) and anti-β-actin (AC-15) (ab6276), rabbit polyclonal anti-GRP75 (ab2799), anti-MFN1 (ab126575), anti-phospho-IRF3S386 (ab76493) and mouse monoclonal anti-VDAC1 (ab14734) antibodies were purchased from Abcam. Mouse monoclonal anti-actin (A2228) and anti-vinculin (V4505) antibodies were purchased from Sigma-Aldrich. Mouse monoclonal anti-DRP1 (611113) and anti-OPA1 (612607) antibodies were obtained from BD Transduction Laboratories. Rabbit polyclonal anti-MFN2 (11925), anti-BAK (12105), anti-BAX (2772), anti-cGAS (31659S), anti-IRF3 (4302S), anti-STAT1 (9172S), anti-pSTAT1 Tyr701 (9167S), anti-TBK1/NAK (3013S), anti-phospho-TBK1/NAK Ser172 (5483S), anti-RAB9A (5118S), anti-RIG-I (3743S) and anti-STING (50494S) antibodies were obtained from Cell Signaling Technology. Rabbit polyclonal anti-SNX9 (15721-1-AP) was purchased from Proteintech. Mouse monoclonal anti-cytochrome *c* (556433) antibody was purchased from BD Pharmingen. Rabbit polyclonal anti-mtTFAM (GTX103231) antibody was obtained from GeneTex.

For immunofluorescence: rabbit polyclonal anti-mtTFAM (GTX103231) antibody from GeneTex was used and mouse monoclonal anti-DNA (CBL186) antibody was purchased from Millipore. Rabbit polyclonal anti-TOM20 (ab232589), mouse monoclonal anti-TOM20 (ab56783), mouse anti-PDH (ab110333), rabbit anti-pTBK1 (ab109272) and mouse anti-TOM22 (ab57523) antibodies were purchased from Abcam. Mouse anti-cytochrome *c* (556432) antibody was purchased from BD Pharmingen, mouse anti-GM130 (610822) antibody was purchased from BD BioSciences and rabbit polyclonal anti-cGAS (D3080) antibody was purchased from Cell Signaling Technology. Donkey anti-mouse, goat anti-mouse IgG1, goat anti-mouse IgG2a, goat anti-mouse IgGM and goat anti-rabbit Alexa Fluor 488, 565, 594 or 647 were used as secondary antibodies (Invitrogen).

## Immunofluorescence
Immunofluorescence were performed as previously described[40]. In brief, cells were fixed in 5% PFA in PBS at 37 °C for 15 min, then washed three times with PBS, followed by quenching with 50 mM ammonium chloride in PBS. After three washes in PBS, cells were permeabilized in 0.1% Triton X-100 in PBS for 10 min, followed by three washes in PBS. Cells were then blocked with 10% FBS in PBS, followed by incubation with primary antibodies in 5% FBS in PBS, for 2 h at room temperature. Cells were then washed three times in 5% FBS in PBS and incubated with the corresponding secondary antibody (dilution 1:1,000) prepared in 5% FBS in PBS. After three washes in PBS, coverslips were briefly rinsed in water and mounted onto slides using Dako fluorescence mounting medium (Dako) or in ProLong Diamond Antifade Mountant (P369621) purchased from Thermo Fisher Scientific.

## Confocal microscopy
Stained cells were imaged using a 100× objective lense (NA 1.4) on a Nikon Eclipse TiE inverted microscope with appropriate lasers using an Andor Dragonfly 500 spinning disk system, equipped with a Zyla 4.2 PLUS sCMOS camera (Andor), coupled with Fusion software. For mitochondrial morphology analysis, nine stacks of 0.2 µm each were acquired using the 100× objective. Images acquired in the same conditions of laser intensity and exposure time from the same experiment were then compiled by 'max projection' and mitochondrial morphology was analysed and presented as intermediate, elongated, swollen, swollen-elongated or fragmented. Mitochondrial morphology was also quantified in a semi-automated and unbiased manner using a modified version of MitoMAPR (ref. [41]). In brief, regions of interest (ROIs) were pre-processed using background subtraction (rolling ball 50), followed by 1× unsharp mask and 1× smooth filter functions in ImageJ. The images were then thresholded and converted to binary. To separate closely connected mitochondrial particles, we used the built-in watershed algorithm in ImageJ (ref. [42]). The images were batch processed using this workflow to calculate the mitochondrial area (µm$^2$) of each mitochondrial particle and the number of mitochondrial particles within a ROI using the 'analyse particle' function of ImageJ. For the quantification of cytosolic DNA foci, the number of DNA foci outside the nucleus and the mitochondrial perimeter was counted manually. Cells were considered as 'releasing cytosolic DNA foci' when the number of DNA foci counted was higher than two per cell. The number of TOM20$^-$PDH$^+$ and TOM20$^-$PDH$^+$DNA$^+$ MDVs outside mitochondria was assessed manually. Cells were considered as 'releasing vesicles' when the number of DNA foci counted was higher than one per cell.

Representative images were processed once with the 'smooth' function in Fiji.

## Super-resolution microscopy
Super-resolution images (Fig. 4c and Extended Data Fig. 3a) were acquired with a Nikon N-SIM microscope. Nine *z*-stacks of 0.2 µm each were acquired using a SR Apo TIRF 100× 1.49 N.A. oil objective and a DU897 Ixon camera (Andor). Raw images were computationally reconstructed using the reconstruction slice system from NIS-Elements software (Nikon). Mitochondrial diameter was calculated using the Fiji software.

Super-resolution images related to TOM20$^-$PDH$^+$TFAM$^+$ MDVs (Extended Data Fig. 11d) were acquired with a Zeiss LSM 880 microscope with an Airyscan detector (Carl Zeiss Microscopy), using a Zeiss 63× oil

lens, NA 1.4, as the primary objective. Four-colour excitation was performed with a blue diode laser for 405 nm, argon laser for 488 nm, He 543 laser for 561 nm and He 633 laser for 647 nm. The Airyscan detector was used in SR mode using all 32 pinholes, thus increasing the resolution to around 140 nm in $x$, $y$ and $z$ to capture the image. The image was reconstructed by pixel reassignment and deconvolution on the Zen Black platform (Carl Zeiss Microscopy). The final image was produced on ImageJ by background subtraction (rolling ball 50) and one run of the smooth filter.

### Characterization of MDVs by super-resolution microscopy
Images were acquired with a Zeiss Elyra7 equipped with Lattice SIM$^2$ (Carl Zeiss Microscopy) with a 15-phased imaging protocol for all channels (Extended Data Fig. 11e). A Zeiss 63× OIL: Plan-Apo 63×/1.4 Oil Corr WD: 0.35 was used as the primary objective to obtain representative images of individual cells. The images were then reconstructed on the Zen Black platform (Carl Zeiss Microscopy) with a two-step SIM$^2$ function. The reconstructed images after deconvolution have an approximate $x$, $y$ resolution of 80 nm and $z$ resolution of 200 nm. The images used for quantifying the number and dimensions of MDVs were acquired on a 40× OIL: Plan-APO 40×/1.4 Oil DIC (UV) VIS-IR objective using the SIM Apotome mode to maximize the field of view and include more than one cell at a higher resolution.

MDVs were selected on the basis of the following criteria: positive for the markers PDH, DNA, but negative for TOM20. ROIs positive for bona fide MDVs were duplicated and saved for quantification.

MDVs were batch-quantified in an unbiased manner using a MACRO script written for ImageJ. In brief, the script splits the images and selects the PDH channel, which is then converted into binary mask for quantification. Using the 'Analyse particle' function of Fiji, the following attributes of the MDVs were generated and stored as a .csv file: area of individual MDV ($\mu m^2$), maximum Feret's diameter ($\mu m$), minimum Feret's diameter ($\mu m$) and circularity of individual MDV.

As can be seen from the distribution of the circularity values, most of the MDVs are of irregular shape (circularity of a true circle ($4\pi \times$ {area/perimeter$^2$} = 1). Thus, we used an average of the maximum and minimum Feret's diameter to represent the dimensions of an MDV. For calculating the area or size of an individual MDV, ImageJ uses an ellipse fitting strategy as described previously[42]. We used 400 individual MDVs to generate an average area ($\mu m^2$) and average Feret's diameter distribution ($\mu m$) of MDVs.

### Measuring mitochondrial volume by TEM
To measure the surface volume density, a square test-grid was placed over three randomly selected cell areas of the cell of interest. The number of test-points ($P$) inside an organelle was calculated, and the number of intersections ($I$) between the mitochondria and the test-grid was used to measure the surface density of the organelle membrane ($Sv = 2\sum I/d\sum P$, in $\mu m^2/\mu m^3$, where $d$ is the real distance between the test-lines in the defined image). The absolute surface area was calculated by multiplying the surface volume density of the organelle of interest by the organelle volume. In brief, vertical serial 1-$\mu m$ sections were made through the cell of interest, and then the section where the nucleus diameter was maximal was re-embedded. Next, serial ultrathin vertical sections were obtained. The section where the width of the nucleus was maximal was considered as the central section, and the vertical axis was generated through the nuclear centre. Then the absolute organelle volumes were measured using the discretized version of vertical rotator. To test whether differences were significant (that is, $P < 0.05$), Student's $t$-tests were used. A difference was considered significant when $P < 0.05$.

### Cell fractionation
Cells were washed three times with cold PBS and scraped over ice in heavy membranes isolation buffer (HMIB) (220 mM mannitol, 70 mM sucrose, 10 mM HEPES pH 7.5, 1 mM EGTA and Roche complete protease inhibitor cocktail). Cells were broken with manual tissue grind (Kimble). Samples were centrifuged at 800$g$ for 10 min at 4 °C. Post-nuclear supernatants were then centrifuged at 2,300$g$ for 10 min at 4 °C to pellet the heavy mitochondrial fraction (crude mitochondria). Pellets were washed and re-centrifuged at 4 °C for a further 15 min at 9,000$g$. The post-heavy membrane supernatants were then centrifuged at 100,000$g$ at 4 °C for 60 min to obtain the cytosolic fraction. Total extracts, heavy membranes and cytosolic fractions were extracted with HMIB 1% Triton X-100, normalized for protein content using a BioRad protein assay (BioRad) and processed for SDS–PAGE and immunoblotting.

### Isolation of cytosolic fractions and ddPCR
Cytosolic fractions were obtained by extracting 10$^5$ cells with 25 $\mu g$ $ml^{-1}$ digitonin in isolation buffer (150 mM NaCl and 50 mM HEPES) for 10 min on ice. Samples were then centrifuged at 800$g$ for 5 min at 4 °C. Supernatants were then further centrifuged at 25,300$g$ for 10 min at 4 °C and pellets were discarded. Samples were normalized for protein concentration using a BioRad protein assay (BioRad). DNA was extracted from equal amounts of cytosolic fractions using a DNA extraction kit (Qiagen) following the manufacturer's instructions. mtDNA copy number was assessed using ddPCR (Biorad) using specific probes directed against mtCo3, mtND1 and mtD-loop.

Primer sequences (IDT):
mtND1_for: 5′-GAGCCTCAAACTCCAAATACTCACT-3′
mtND1_rev: 5′-GAACTGATAAAAGGATAATAGCTATGGTTACTTCA-3′
mtCo3_for: 5′-CCTCGTACCAACACATGATCTAGG-3′
mtCo3_rev: 5′-AGTGGGACTTCTAGAGGGTTAAGTG-3′
mtDloop_for: 5′-AATCTACCATCCTCCGTGAAACC-3′
mtDloop_rev: 5′-TCAGTTTAGCTACCCCCAAGTTTAA-3′
Fluorescent probe sequences (PrimeTime 5′HEX/FAM/3′ BHQ-1; IDT):
FAM-mtDloop1: 5′-ACCAATGCCCCTCTTCTCGCTCC-3′
HEX-mtND1: 5′-CCGTAGCCCAAACAAT-3′
HEX-mtCo3: 5′-ACCTCCAACAGGAATTTCA-3′.

### Extraction of RNA from cells
Cells were homogenized using QIAshredder (Sigma-Aldrich) and RNA was extracted using the Qiagen AllPrep DNA/RNA Kit according to the manufacturer's instructions. RNA concentration was determined using a Nanodrop (ND1000). Reverse transcription was performed using 800 ng total RNA with the Applied Biosystems High Capacity cDNA Reverse Transcription Kit (Thermo Fisher Scientific).

### qRT–PCR
qRT–PCR was performed on 5 ng of cDNA using a final concentration of 2 $\mu M$ each of forward and reverse primers (see primers list) using Quantitect Syber Green Master Mix (Qiagen) on an Applied Biosystems StepOne Plus or QuantStudio5 real-time PCR system. Experiments were analysed using the $\Delta\Delta Ct$ method. Statistical analysis was performed using Prism 7 software. Samples were normalized relative to the expression of an endogenous control gene (*Rplp0*; see primers list for sequences). Amplification primers were designed using Primer 3.1 to span one intron and purchased from Sigma-Aldrich, and are listed in Supplementary Tables 4 and 5.

### OCR measurements (Seahorse)
The OCR was measured using the real-time flux analyser XF-24e (Seahorse Bioscience) as previously described[12]. In brief, $4 \times 10^4$ *iFh1$^{fl/flCL29}$* or *iFh1$^{fl/flCL33}$* cells were treated with either vehicle (ethanol) or 4-OHT for 48 h, after which fresh medium was added. The cells were then kept in culture for 3, 10, 15 or 21 days before starting the OCR assay. For this, cells were treated with 1 $\mu M$ oligomycin, 2 $\mu M$ carbonyl cyanide-p-trifluoromethoxyphenylhydrazone (FCCP), rotenone and antimycin A (both 1 $\mu M$) (all Sigma-Aldrich). At the end of the run, cells were lysed using RIPA buffer (25 mM Tris/HCl pH 7.6, 150 mM NaCl,

1% NP-40, 1% sodium deoxycholate and 0.1% SDS). The protein content of each well was measured using a BCA kit (Pierce) following the manufacturer's instructions. OCR was normalized to the total protein content as indicated.

## Oxygen consumption experiments (Oroboros)
Cells were seeded on 10-cm dishes to reach more than 90% confluency on the day of experiments. Cells were rinsed with PBS and trypsinized followed by a short spin at 300$g$ for 3 min at room temperature. The cell pellet was resuspended in complete DMEM devoid of FBS, and the cell density was determined using a Countless 3 automated cell counter (Thermo Fisher Scientific). Oxygen consumption on intact cells was monitored polarographically with an Oxygraph-2k (Oroboros Instruments) at 37 °C in a 2-ml chamber with complete DMEM without FBS, calibrated daily using air-saturated reaction medium and a correction factor of 0.89. Cell suspension was added to the chamber and allowed to equilibrate (2–3 min) before closing the chamber and starting the record. After a stable basal respiration was achieved (5 min), oligomycin (5 μM) was added to assess ATP-dependent respiration (5 min), followed by a titration with the uncoupler FCCP at 0.5-μM pulses to determine maximal uncoupled respiration (180 s each). Finally, rotenone (0.5 μM) was added to inhibit Complex I-linked respiration. Respiration rates were calculated as the average value over a 30-s window in DatLab 7.4 (Oroboros Instruments) and are expressed in pmol $O_2$ per second per $10^6$ cells.

## SDS–PAGE and immunoblotting
Cells were lysed in RIPA buffer (20 mM Tris pH 8.0, 150 mM NaCl, 0.1% SDS, 1% deoxycholic acid, 1% NP-40 and complete protease inhibitor cocktail (Roche)). Alternatively, PathScan(R) Sandwich ELISA lysis buffer (Cell Signaling Technology) supplemented with complete protease inhibitor cocktail and phospho-stop (Sigma-Aldrich) was used to extract samples to resolve phosphoproteins. Samples were normalized for protein concentration using a BioRad protein assay (BioRad). Proteins were resolved by SDS–PAGE and transferred to nitrocellulose membranes (0.2 μm pore size, GE Healthcare) or PVDF membrane (0.2 μm pore size, GE Healthcare).

For immunoblot analysis, membranes were blocked with 2–5% non-fat milk or 2% bovine serum albumin (BSA) in PBS for 60 min at room temperature. Membranes were incubated with primary antibody at the appropriate dilution (in 2–5% non-fat milk or 2% BSA in 0.05% Tween-20 in PBS) at 4 °C overnight. Membranes were washed in 0.05% Tween-20 in PBS three times for 15 min and incubated with appropriate secondary antibodies (1:3,000 in 2–5% milk or 2% BSA in 0.05% Tween-20 in PBS). Membranes were treated with Western Lightning Plus ECL (Perkin Elmer), and exposed to chemiluminescence either on films (PROTEC) or on a digital ECL machine (Amersham) for image quantification. Uncropped immunoblots are presented in Supplementary Fig. 10.

## ELISA
Serum levels of IL-6, IL-10 and IFNβ were measured by ELISA (Biotechne) according to the manufacturer's instructions. Ninety-six-well Maxisorp (Thermo Fisher Scientific) plates were coated with 100 μl of capture antibody diluted with PBS (2 μg ml$^{-1}$ IL-6, 2 μg ml$^{-1}$ IL-10 and 4 μg ml$^{-1}$ IFNβ) and sealed overnight at room temperature. Plates were then washed four times with 0.05% PBS-Tween-20 (PBS-T). After blotting, plates were blocked with 300 μl of 1% BSA in PBS for 1 h at room temperature. After repeated washing, either 50 μl (IL-6 and IL-10) or 100 μl (IFNβ) of samples was added in duplicate; 1:10 dilutions in 1% BSA were performed for all samples. Standard concentrations ranged from 600 pg ml$^{-1}$ to 9.38 pg ml$^{-1}$ for IL-6; 2,000 pg ml$^{-1}$ to 31.3 pg ml$^{-1}$ for IL-10; and 500 pg ml$^{-1}$ to 7.81 pg ml$^{-1}$ for IFNβ. Plates were sealed and incubated at room temperature for 2 h. The washing step was repeated and either 50 μl (IL-10) or 100 μl (IL-6 and IFNβ) of detection antibody diluted in PBS was added (50 ng ml$^{-1}$ IL-6, 50 ng ml$^{-1}$ IL-10 and 250 ng ml$^{-1}$ IFNβ). The plate was sealed for 2 h at room temperature.

Plates were then washed four times and 100 μl of 1:40 streptavidin–HRP diluted in 1% BSA was added per well. Plates were sealed and incubated for 20 min at room temperature, followed by four washes. Tetramethylbenzidine (TMB) substrate solution (100 μl) was added for up to 15 min, followed by 50 μl of 1 mol l$^{-1}$ $H_2SO_4$ to stop the reaction. Optical density was read immediately in a plate reader (BioTek Synergy H1) at an absorbance of 450 nm.

## Number of cells used for image quantification
For image quantification, experiments are presented from three independent experiments (unless otherwise specified) with a total number of cells ($n$) analysed as: main figures: Fig. 2d–f: $n$ = 173, 157, 158, 170, 160, 157, 162 for NT, vehicle, d1, d3, d6, d10, d15, respectively. Fig. 3a–c: $n$ = 221, 192, 185 for vehicle, 200 μM MMF, 400 μM MMF, respectively. Fig. 4d,e: for TOM20$^-$PDH$^+$DNA$^+$ vesicle analysis: $n$ = 166, 180, 167, 155, 170 for NT, 2d, 4d, 6d, 8d, respectively. Fig. 4e: for cytosolic DNA foci analysis: $n$ = 165, 171, 162, 152, 160 for NT, 2d, 4d, 6d, 8d, respectively. Fig. 4f: $n$ = 156; number of vesicles analysed: $n$ = 155. Fig. 4h,i: for TOM20$^-$PDH$^+$DNA$^+$ vesicle analysis: $n$ = 158, 173, 158 for NT, MMF 6d siScr, MMF 6 days si$Snx9$, respectively. Fig. 4i: for cytosolic DNA foci analysis: $n$ = 151, 166, 197, for NT, MMF 6d siScr, MMF 6d si$Snx9$, respectively. Extended Data: Extended Data Fig. 2g: $n$ = 173, 157, 158, 170, 160, 157, 162 for NT, vehicle, d1, d3, d6, d10, d15, respectively. Extended Data Fig. 2h,i: $n$ = 30, 32, 35, 38, 32, 40, 36 and number of ROIs analysed: $n$ = 84, 85, 84, 85, 85, 84, 85 for NT, vehicle, d1, d3, d6, d10, d15, respectively. Extended Data Fig. 2j: number of mitochondria analysed: 11, 22, 79, 54, 46 for $cFh1^{+/+}$, $cFh1^{+/+}$ + Cre, $cFh1^{fl/fl}$, $cFh1^{-/-CL1}$, $cFh1^{-/-CL19}$, respectively. Extended Data Fig. 3b: 30 cells were analysed per condition with a number of mitochondria analysed of $n$ = 429, 827, 780 for $cFh1^{fl/fl}$, $cFh1^{-/-CL1}$, $cFh1^{-/-CL19}$, respectively. Extended Data Fig. 3c,f,g: $n$ = 272, 246, 306, 300 for $cFh1^{fl/fl}$, $cFh1^{-/-CL1}$, $cFh1^{-/-CL19}$, $cFh1^{-/-CL1}$ + pFH-GFP. Extended Data Fig. 3d: $n$ = 33, 45, 67, 39 and number of ROIs analysed: $n$ = 79, 71, 164, 73 for $cFh1^{fl/fl}$, $cFh1^{-/-CL1}$, $cFh1^{-/-CL19}$, $cFh1^{-/-CL1}$ + pFH-GFP, respectively. Extended Data Fig. 4e: $n$ = 221, 192, 185 for vehicle, 200 μM MMF, 400 μM MMF, respectively. Extended Data Fig. 5a: $n$ = 221, 192, 185 for vehicle, 200 μM MMF, 400 μM MMF, respectively. Extended Data Fig. 5e: $n$ = 169, 188 for DMSO and MMF 8d, respectively. Extended Data Fig. 6g: $n$ = 182, 156, 189, 228 for $cFh1^{fl/fl}$, $cFh1^{-/-CL1}$, $cFh1^{-/-CL1}$ + pcytoFH-GFP, $cFh1^{-/-CL1}$ + $pFH$-GFP, respectively. Extended Data Fig. 7b,c: $n$ = 152, 153, 162, 155, 155 for vehicle, DMS 200 μM, DMS 400 μM, DMS 1 mM, DMS 5 mM, respectively. Extended Data Fig. 8d: $n$ = 933, 660 for DMSO and MMF 8d, respectively. Extended Data Fig. 9a: $n$ = 175, 171, 183, 168, 166, 187, 188, 186, 154, 184, 161, 180, 169, for $cFh1^{fl/fl}$, $cFh1^{-/-CL1}$ siScr, $cFh1^{-/-CL1}$ si$Snx9$, $cFh1^{-/-CL1}$ si$Vdac1$, $cFh1^{-/-CL1}$ si$Bax/Bak1$, $cFh1^{-/-CL1}$ si$Rab9$, $cFh1^{-/-CL1}$ si$Drp1$, $cFh1^{-/-CL1}$ si$Opa1$, $cFh1^{-/-CL1}$ si$Mfn1$, $cFh1^{-/-CL1}$ si$Mfn2$, $cFh1^{-/-CL1}$ si$Cgas$, $Fh1^{-/-}$ si$Sting1$, $cFh1^{-/-CL1}$ si$Rig-I$, respectively. Extended Data Fig. 9e: $n$ = 31, 32, for $cFh1^{fl/fl}$ and $cFh1^{-/-CL1}$, respectively. Extended Data Fig. 10b,c: $n$ = 151, 154, 166, 159, 161, 163, for $cFh1^{fl/fl}$, $cFh1^{fl/fl}$ siScr, $cFh1^{fl/fl}$ si$Snx9$, $cFh1^{-/-CL1}$ siScr, $cFh1^{-/-CL1}$ si$Snx9$, $cFh1^{-/-CL19}$ siScr, $cFh1^{-/-CL19}$ si$Snx9$, respectively. Extended Data Fig. 10e,f: $n$ = 55, 50, 53, 55, 60, 53 and number of ROIs analysed: $n$ = 168, 162, 158, 348, 125, 155, 76 for $cFh1^{fl/fl}$ + siScr, $cFh1^{fl/fl}$ + si$Snx9$, $cFh1^{-/-CL1}$ + siScr, $cFh1^{-/-CL1}$ + si$Snx9$, $cFh1^{-/-CL19}$ + siScr, $cFh1^{-/-CL19}$ + si$Snx9$, respectively. Extended Data Fig. 10g,h: $n$ = 191, 167, 160, 179, 190, 147, 160, for vehicle, d6 siScr, d6 si$Snx9$, d10 siScr, d10 si$Snx9$, d15 siScr, d15 si$Snx9$, respectively. Extended Data Fig. 11a–c: $n$ = 166, 180, 167, 155, 170, for NT, 2d, 4d, 6d, 8d MMF, respectively. Extended Data Fig. 11d: $n$ = 85. Extended Data Fig. 11f,g: $n$ = 408 ROIs analysed. Extended Data Fig. 11i: $n$ = 189, 155, 272, 159, 159, 144 for vehicle, 4-OHT d1, d3, d6, d10, d15, respectively. Extended Data Fig. 12b: $n$ = 170, 167, 145, 136, 133, 136, 139 for $Fh1^{+/+}$ + DMSO, $Fh1^{+/+}$ + MMF 6d, $Fh1^{+/+r0}$ + DMSO $Fh1^{+/+r0}$ + MMF 1d, 3d, 6d, 8d, respectively. Extended Data Fig. 12c: $n$ = 151, 166, 197 for NT, MMF 6d siScr, MMF 6d si$Snx9$, respectively. Supplementary Information: Supplementary Fig. 1f–h: $n$ = 166, 159, 166, 165, 175, 181, 170 for NT, vehicle, d1, d3, d6, d10, d15 respectively. Supplementary Fig. 1i,j: $n$ = 35, 42, 43, 47, 50, 48, 36 and number of ROI analysed: $n$ = 99, 115, 137,

96, 127, 127, 76 for NT, vehicle, d1, d3, d6, d10, d15, respectively. Supplementary Fig. 3e: $n = 163, 161, 163$ for $Sdhb^{fl/fl}$, $Sdhb^{-/-CL5}$, $Sdhb^{-/-CL7}$, respectively. Supplementary Fig. 4c: $n = 152, 153, 162, 155, 155$ for vehicle, DMS 200 μM, DMS 400 μM, DMS 1 mM, DMS 5 mM, respectively. Supplementary Fig. 3f,g: $n = 163, 161, 163$ for $Sdhb^{fl/fl}$, $Sdhb^{-/-CL5}$, $Sdhb^{-/-CL7}$, respectively. Supplementary Fig. 4d: $n = 174, 165, 171$ for $cFh1^{fl/fl}$, $cFh1^{-/-CL1} +$ pEGFP, $cFh1^{-/-CL1} +$ pEGFP:NDI1, respectively. Supplementary Fig. 9b: $n = 408$ ROIs analysed.

## Human tissue samples

All samples were from adult individuals with confirmed *SDH*- or *FH*-deficient renal cancer, with informed consent (Research Ethics Committee approval reference 16/WS/0039).

## Statistical analysis

Errors bars in graphs represent the mean ± s.e.m. from at least three independent experiments. The number of cells analysed is provided in the Methods. Statistical significance was analysed using a two-tailed, unpaired, Student's *t*-test, Mann–Whitney test or one-way or two-way ANOVA. All statistical analyses were performed using Excel or Prism software.

## Reporting summary

Further information on research design is available in the Nature Portfolio Reporting Summary linked to this article.

## Data availability

All data are included within the article or the supplementary information. Full versions of all gels and blots are provided in Supplementary Fig. 10. See the Supplementary Information guide for details. Raw FastQ files for RNA-sequencing analyses are publicly available in the Gene Expression Omnibus (GEO) repository with the accession code GSE183745. All materials generated in this study are readily available from the authors. All other data supporting the findings of this study are available from the corresponding authors upon request. For the purpose of open access, the authors have applied a Creative Commons Attribution (CC BY) licence to any Author Accepted Manuscript version arising. Source data are provided with this paper.

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

**Acknowledgements** This work was supported by the Medical Research Council to J.P. (MC_UU_00015/7 and MC_UU_00028/5) and C.F. (MRC_MC_UU_12022/6). V.P. was supported by the European Union's Horizon 2020 research and innovation programme (MITODYN-749926) (2017–2019). V.Z. was supported by a WWCR grant (14-0319). I.M.N.W. was supported by a grant from the Nora Baart Foundation. T.Y. was supported by the European Research Council Consolidator Award to C.F. (ERC819920). E.N. was supported by a CRUK Programme Foundation award to C.F. (C51061/A27453). J.L.M. was supported by a MRC-funded graduate student fellowship. G.C.P. is supported by the Swiss National Science Foundation (Synergia project CRSII5_180326). S.N. was the recipient of a Daiichi Sankyo Foundation of Life Science postdoctoral fellowship (2017–2019). A.F. was supported by a Rotary Global Grant. M.S.-M is supported by CRUK. We thank M. Micaroni for the quantification of the volume of mitochondria from the TEM in Extended Data Fig. 2j. We apologize for some relevant studies not being cited in the manuscript owing to space limitations.

**Author contributions** V.Z. and V.P. provided intellectual input to the project, designed and performed the experiments, performed statistical analysis and generated the figures. I.H.-M. performed the computational analysis of the gene expression datasets. J.J. and I.M.N.W. assisted V.Z. with the generation and characterization of the inducible cell lines. J.L.M. contributed to the acquisition of super-resolution images and assisted V.P. with some confocal-image acquisition. A.F. contributed to the formulation of the project, and performed initial experiments relating to mitochondrial morphology and the observation of cytosolic DNA foci in *FH*-deficient cells. S.R.C. performed super-resolution microscopy analyses of MDVs and assisted V.P. with the quantification of mitochondrial morphology. A.S.H.C., L.T., E.N. and M.Y. performed the mass spectrometry and analysis of the tissue and cell samples. G.C.P. assisted V.P. with the respiration analysis in NDI1 experiments. T.Y. assisted V.Z. with the mouse tissue and cell line processing as well as processing of cell samples for mass spectroscopy. D.B. and S.S. processed the RNA-sequencing raw data for analysis. F.C. performed the electron microscopy imaging acquisition of the mouse kidney tissue. S.N. and M.J. assisted V.P. with immunoblot acquisition. A.S. assisted V.Z. with the in vivo experiments. M.T., K.B. and Z.B. provided HLRCC samples and performed the analyses of inflammatory markers. T.M. performed the deconvolution analyses to assess the immune contribution in the mouse tissue and human data from bulk gene expression. M.S.-M. generated the cytoFH construct and performed the experiments with the cytoFH-expressing cells. V.Z., V.P., J.P. and C.F. wrote the manuscript with input from all authors. J.P. and C.F. supervised the experiments, conceived and oversaw the study. All authors read and approved the manuscript. Equal second contribution: I.H.-M., J.J., I.M.N.W., J.L.M. and A.F.

**Funding** Open access funding provided by Universität zu Köln and MRC Mitochondrial Biology Unit, University of Cambridge.

**Competing interests** C.F. is a scientific adviser for Istesso. The remaining authors declare no competing interests.

**Additional information**
**Correspondence and requests for materials** should be addressed to Julien Prudent or Christian Frezza.

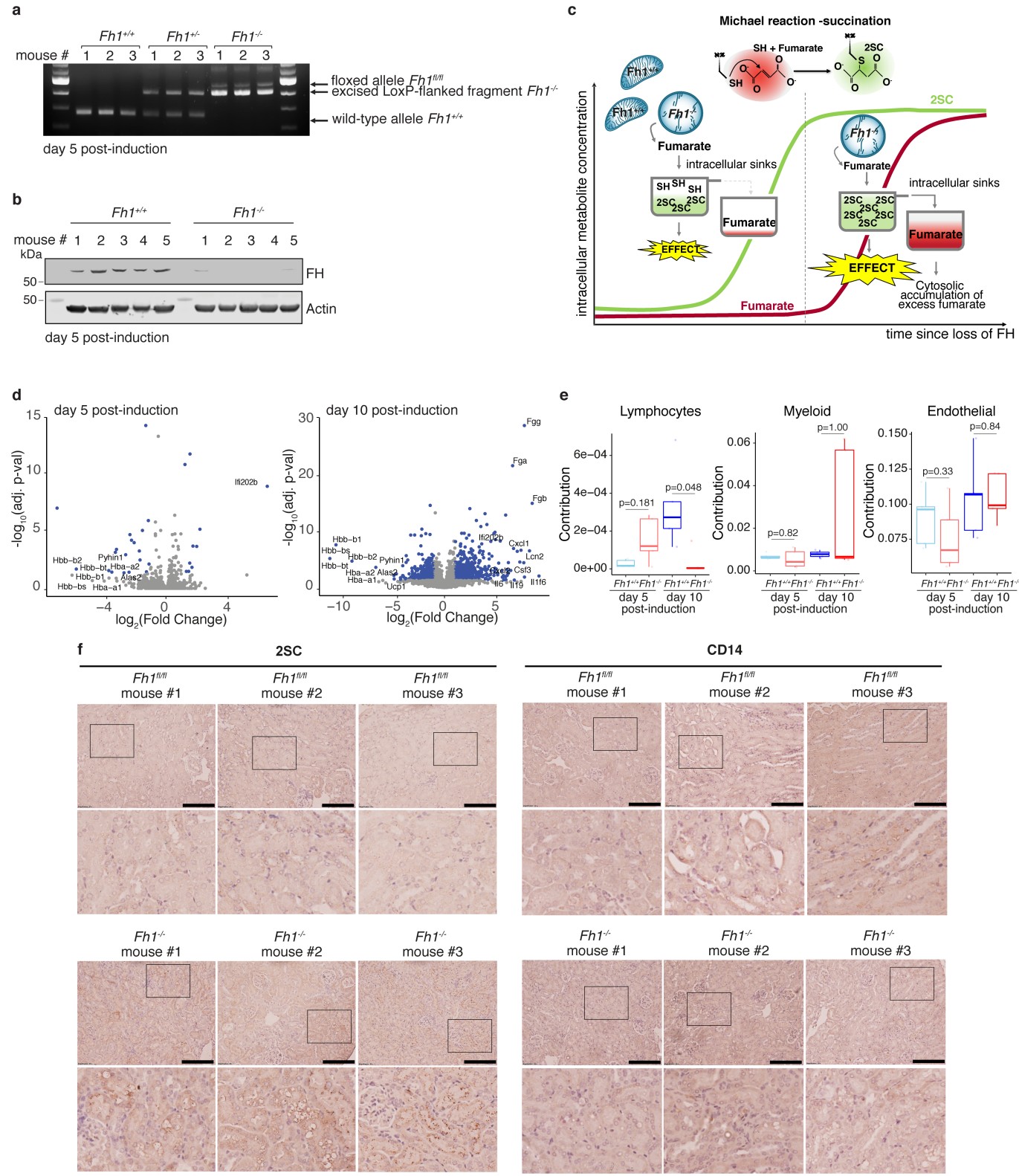

**Extended Data Fig. 1** | See next page for caption.

**Extended Data Fig. 1 | Characterization of an inducible mouse model of *Fh1* loss. a**, Genomic DNA PCR using primers[4] flanking exons 3 and 4 of *Fh1* showing amplification product for the *Fh1* wild-type allele (bottom band; *Fh1*[+/+]), the floxed (unrecombined) allele (top band; *Fh1*[fl/fl]) and the recombined allele (middle band; excised LoxP fragments; *Fh1*[−/−]) from kidney samples at day 5 post-induction. Mice carrying one allele of each *Fh1*[+/+] and *Fh1*[fl/fl] i.e. *Fh1*[+/fl]; *Rosa26-Cre-ERT2* were used for comparison with *Fh1*[+/+] and *Fh1*[−/−] mice and labelled *Fh1*[+/−]. **b**, Immunoblots of specified proteins in kidney tissue samples at day 5 post-induction. Five kidney samples are shown. **c**, Upon loss of Fh1 activity, fumarate cannot enter the enzymatic reaction that converts it into succinate but, instead, can enter a chemical reaction termed Michael addition[7] whereby it is chemically added to thiol residues of free cysteine to form 2SC, and of proteins which is considered a metabolic marker of FH activity loss. This reaction can be seen as a "buffering tank" that mop-up the excess of fumarate.

Therefore, succinated proteins and 2SC levels increase before an intracellular increase in fumarate is detected. Only when succination is "at saturation" with the 2SC intracellular sinks (or "buffering tanks") full, fumarate starts to accumulate. Thus, a modest increase in fumarate can still be accompanied by a strong effect. **d**, Volcano plots showing the differentially expressed genes in *Fh1*[−/−] *vs Fh1*[+/+] kidney at day 5 (left) and day 10 (right). **e**, Deconvolution method on bulk expression data (https://github.com/Danko-Lab/TED and Methods) was applied to determine the cellular composition of mouse kidney tissue at day 5 and day 10 post-induction. Pairwise comparison using Wilcoxon rank sum exact test and Benjamini–Hochberg p-value adjustment were used. **f**, Immunohistological staining of *Fh1*[fl/fl] (top) *vs Fh1*[−/−] (bottom) mouse kidney tissue at day 10 post-induction. Left: 2SC staining, right: CD14 staining. Scale bars: 100 μm.

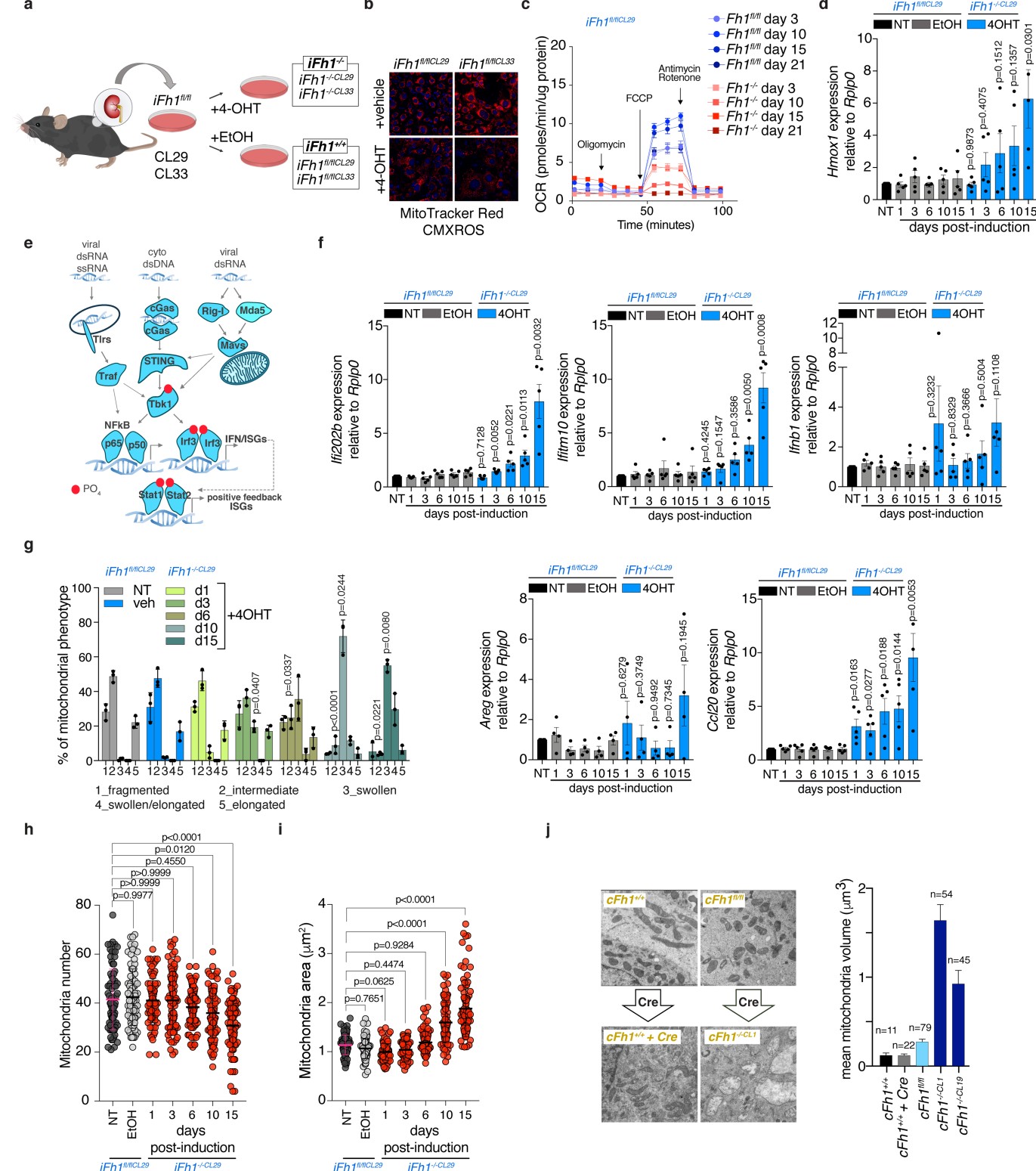

**Extended Data Fig. 2** | See next page for caption.

**Extended Data Fig. 2 | Characterization of inducible *iFh1* epithelial kidney cell lines. a**, Schematic diagram of the generation of inducible *iFh1* epithelial kidney cell lines clones 29 (*iFh1$^{fl/flCL29}$*) and 33 (*iFh1$^{fl/flCL33}$*). **b**, Mitochondrial membrane potential analysis using MitoTracker Red CMXROS in *iFh1$^{fl/flCL29}$* and *iFh1$^{fl/flCL33}$* cells treated with either vehicle (ethanol; EtOH) or 4-OHT (*iFh1$^{-/-CL29}$* and *iFh1$^{-/-CL33}$*, respectively). Scale bar: 25 µm. **c**, Mitochondrial respiration measured using Seahorse in *iFh1$^{CL29}$* cells. *n* = 3 independent experiments. **d**, qRT-PCR showing expression levels of the transcriptional marker of *Fh1* loss, *Hmox1*, in *iFh1$^{CL29}$* cells. *n* = 5 independent experiments. Bar graphs show the fold change expression, for which the expression in control samples was set to 1. Indicated p-values are relative to the corresponding vehicle (EtOH)-treated time point. **e**, Schematic of Pattern Recognition Receptors (PRR) and downstream cascades. **f**, qRT-PCR showing expression levels of *Ifnb1* and ISGs (*Ifi202b*, *Ifitm10*, *Areg* and *Ccl20*) in *iFh1$^{CL29}$* cells. *n* = 5 independent experiments. Bar graphs show the fold change expression, for which the expression in control samples was set to 1. Indicated p-values are relative to the corresponding vehicle (EtOH)-treated time point. **g-i**, Classification of mitochondrial morphology (**g**), quantification of mitochondria number (**h**), and area (**i**), in *iFh1$^{CL29}$* cells. *n* = 3 independent experiments. Indicated p-values are relative to the corresponding vehicle (EtOH)-treated morphology category. **j**, TEM images (left) and mean mitochondria volume quantification (right) in wild-type (*Fh1$^{+/+}$*) or chronic floxed *Fh1* (c*Fh1$^{fl/fl}$*) mouse kidneys epithelial cells treated with Cre-expressing Adenovirus. Data are mean ± s.e.m. **d,f**, Students t-test corrected for multiple comparison with the Holm-Sidak method, **g**, two-way ANOVA with Tukey's multiple comparison test, **h,i**, one-way ANOVA with Tukey's multiple comparison test. For (**g**) p-values are indicated above each specific phenotype and relative to the corresponding phenotype in vehicle-treated control.

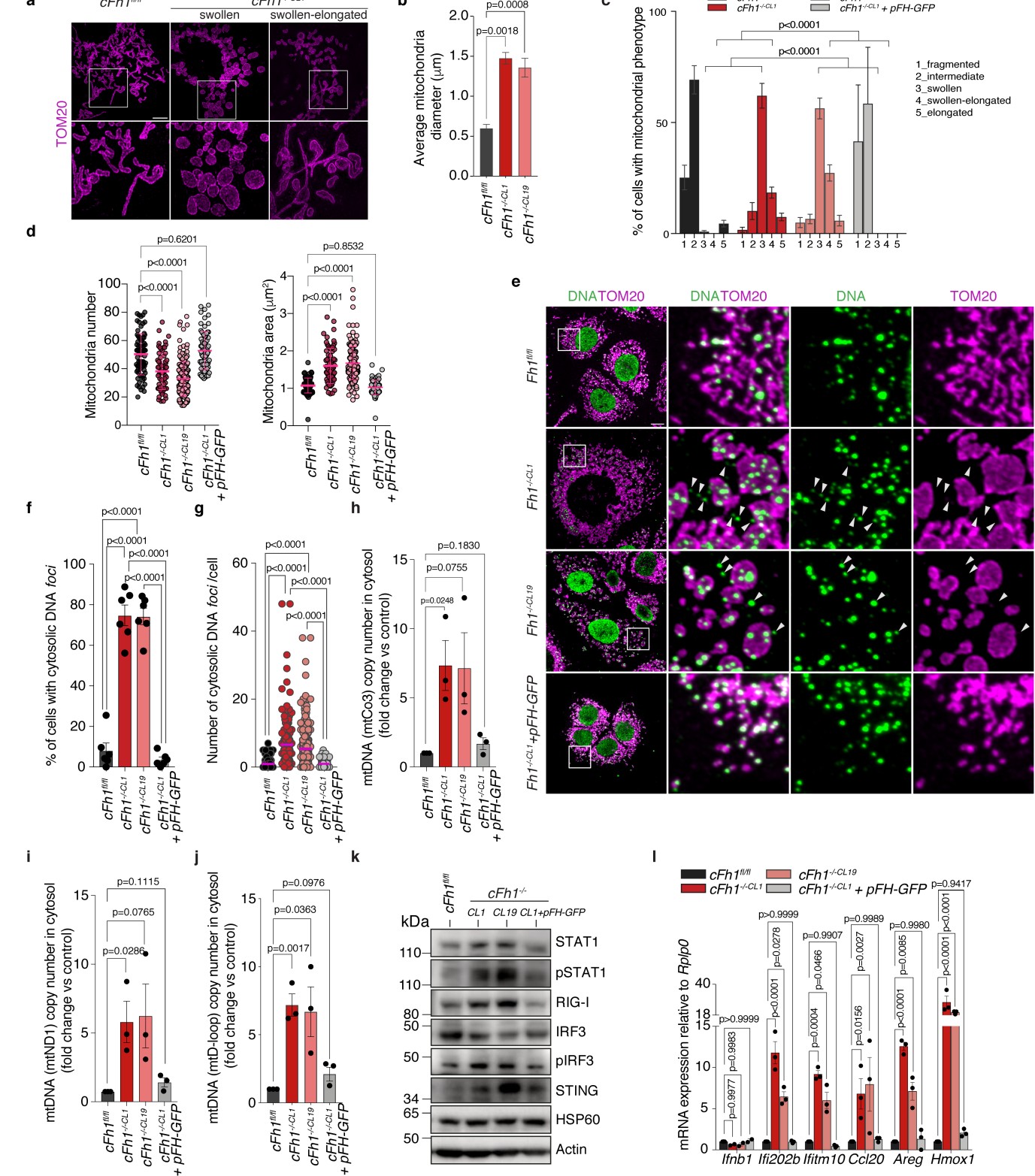

**Extended Data Fig. 3** | See next page for caption.

**Extended Data Fig. 3 | Chronic loss of *Fh1* leads to abnormal mitochondrial morphology and mtDNA release in the cytosol. a**, Representative N-SIM super-resolution images of mitochondrial morphology (TOM20) in epithelial kidney cell line with chronic *Fh1* deletion (cFh1$^{-/-CL1}$) from cFh1$^{fl/fl}$, compared to cFh1$^{fl/fl}$ cells. Scale bar: 5 μm. **b**, Quantification of mitochondrial diameter in *cFh1* cells from (**a**). *n* = 3 independent experiments. **c**, Quantification of mitochondrial morphology in *cFh1* cells. *n* = 3 independent experiments. **d**, Quantification of mitochondrial number (left) and area (right) in *cFh1* cells. *n* = 3 independent experiments. **e**, Representative confocal images of mitochondrial morphology (TOM20) and DNA foci (DNA) in *cFh1* cells. White arrows indicate cytosolic DNA foci. Scale bar: 10 μm. **f**,**g**, Percentage of *cFh1* cells showing cytosolic DNA (**f**), and number of cytosolic DNA foci per cell (**g**), from **e**. *n* = 6 independent experiments. **h-j**, Quantification of mtDNA copy number by ddPCR using either a mtCo3 (**h**), ND1 (**i**) or D-loop (**j**) probe, from isolated cytosolic fractions of *cFh1* cells. *n* = 3 independent experiments. **k**, Immunoblots of specified proteins in *cFh1* cells. **l**, Expression of inflammation-related *Ifi202b* and ISGs in *cFh1* cells measured by qRT-PCR. *n* = 3 independent experiments. Data are mean ± s.e.m. **b**,**d**,**f-j**, one-way ANOVA with Tukey's multiple comparison test, **c**,**l**, two-way ANOVA with Tukey's multiple comparison test.

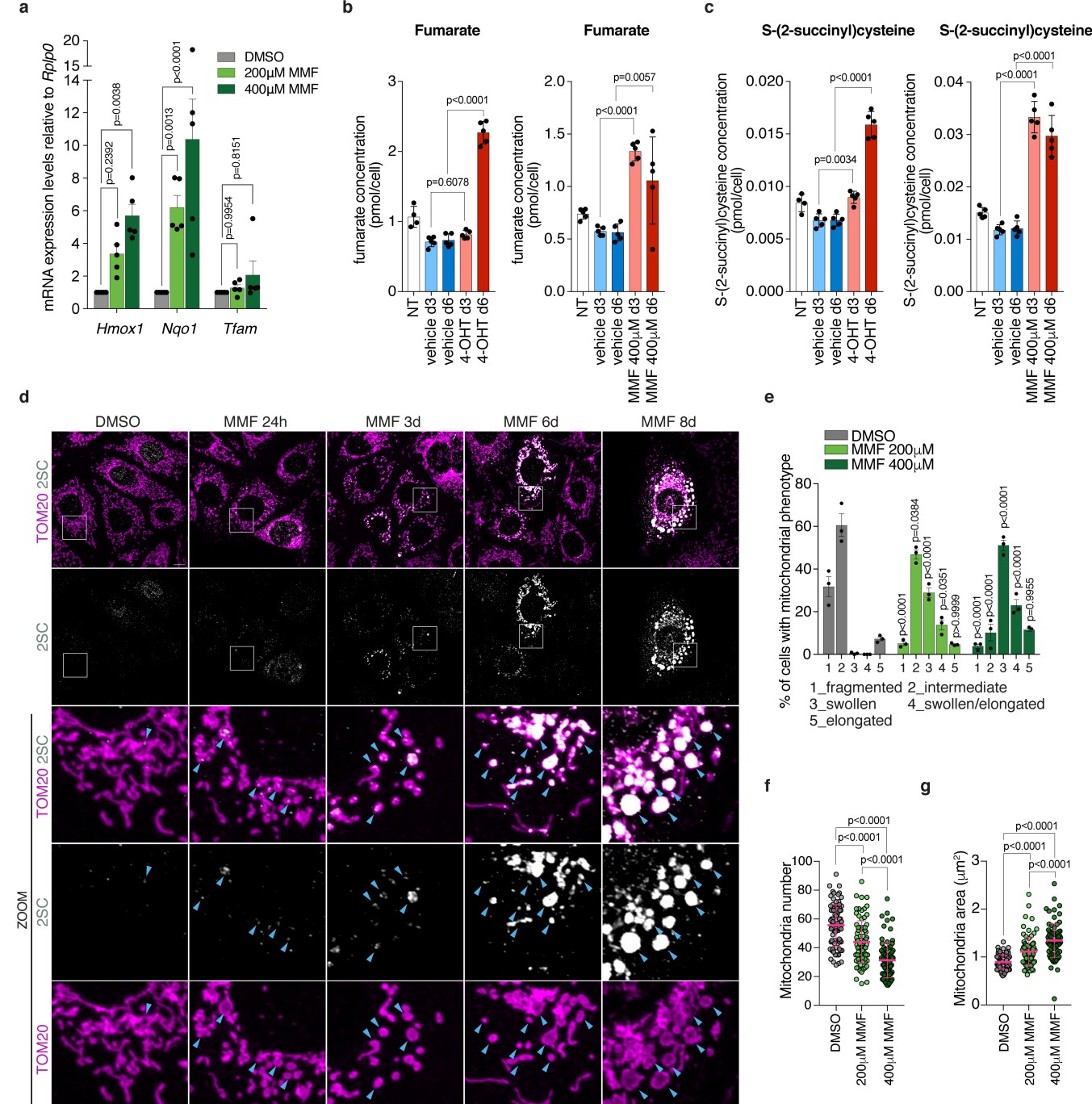

**Extended Data Fig. 4 | MMF treatment phenocopies *Fh1* loss. a**, Expression levels of the transcriptional markers of *Fh1* loss *Hmox1* and *Nqo1* as well as *Tfam* expression in monomethyl fumarate (MMF)-treated *cFh1^{fl/fl}* cells for 8 days compared to vehicle (DMSO)-treated cells, measured by qRT-PCR. *n* = 3 independent experiments. Bar graphs show the fold change expression, for which the expression in control samples was set to 1. **b**, Relative abundance of fumarate levels measured by LC–MS in *iFh1^{fl/flCL29}* untreated (NT) or treated with vehicle (*iFh1^{fl/flCL29}*) or 4-OHT (*iFh1^{−/−CL29}*) or 400 µM MMF for the indicated period of time. *n* = 5 independent experiments. **c**, Relative abundance of 2SC levels measured by LC–MS in *iFh1^{CL29}* cells and *iFh1^{fl/flCL29}* treated with 400 µM MMF for the indicated period of time. *n* = 5 independent experiments. **d**, Representative confocal images of *cFh1^{fl/fl}* cells treated with vehicle (DMSO)

or 400 µM MMF for the indicated period of time. Mitochondria and succinated proteins were labelled using anti-TOM20 and anti-2SC antibodies, respectively. Blue arrows indicate 2-SC-decorated proteins accumulation in mitochondria. Scale bar: 10 µm. **e**-**g**, Classification of mitochondrial morphology (**e**), quantification of mitochondrial number (**f**), and area (**g**) in *cFh1^{fl/fl}* cells treated with DMSO or 400 µM MMF for 8 days. *n* = 3 independent experiments. Data are mean ± s.e.m. **a**–**c**, one-way ANOVA with Tukey's multiple comparison test, **e**, two-way ANOVA with Tukey's multiple comparison test, **f,g**, one-way ANOVA with Tukey's multiple comparison test, for (**e**) p-values are indicated above each specific phenotype and relative to the corresponding phenotype in vehicle (DMSO)-treated control.

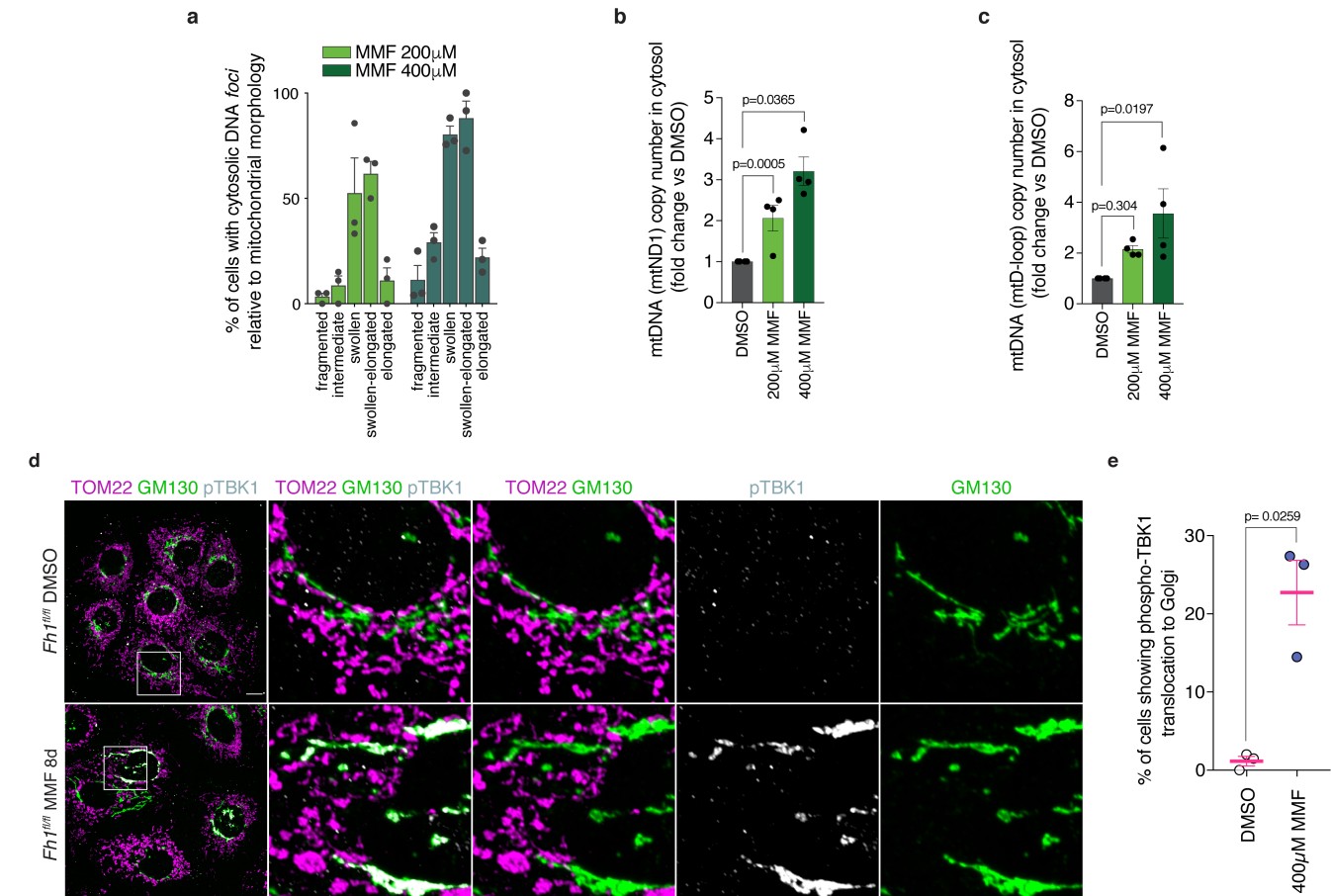

**Extended Data Fig. 5 | MMF treatment induces mtDNA release and the translocation of pTBK1 to the Golgi apparatus. a**, Percentage of MMF-treated *cFh1^{fl/fl}* cells for 8 days showing cytosolic DNA foci according to mitochondrial morphology phenotype. *n* = 3 independent experiments. **b,c**, Quantification of mtDNA copy number by ddPCR using either a ND1 (**b**) or D-loop (**c**) probe, from isolated cytosolic fractions of DMSO- or MMF-treated (8 days) *cFh1^{fl/fl}* cells. *n* = 3 independent experiments. **d**, Representative confocal images of *cFh1^{fl/fl}* treated with 400 μM MMF or DMSO for 8 days. Mitochondria, Golgi apparatus, and pTBK1 were labelled using anti-TOM20, anti-GM130 and anti-pTBK1 antibodies, respectively. Scale bar: 10 μm. **e**, Quantification of *cFh1^{fl/fl}* showing the percentage of cells with pTBK1 recruitment and colocalization with the Golgi apparatus marker, GM130 from **d**. *n* = 3 independent experiments. Data are mean ± s.e.m. **b,c**, one-way ANOVA with Tukey's multiple comparison test, **e**, Students paired t-test.

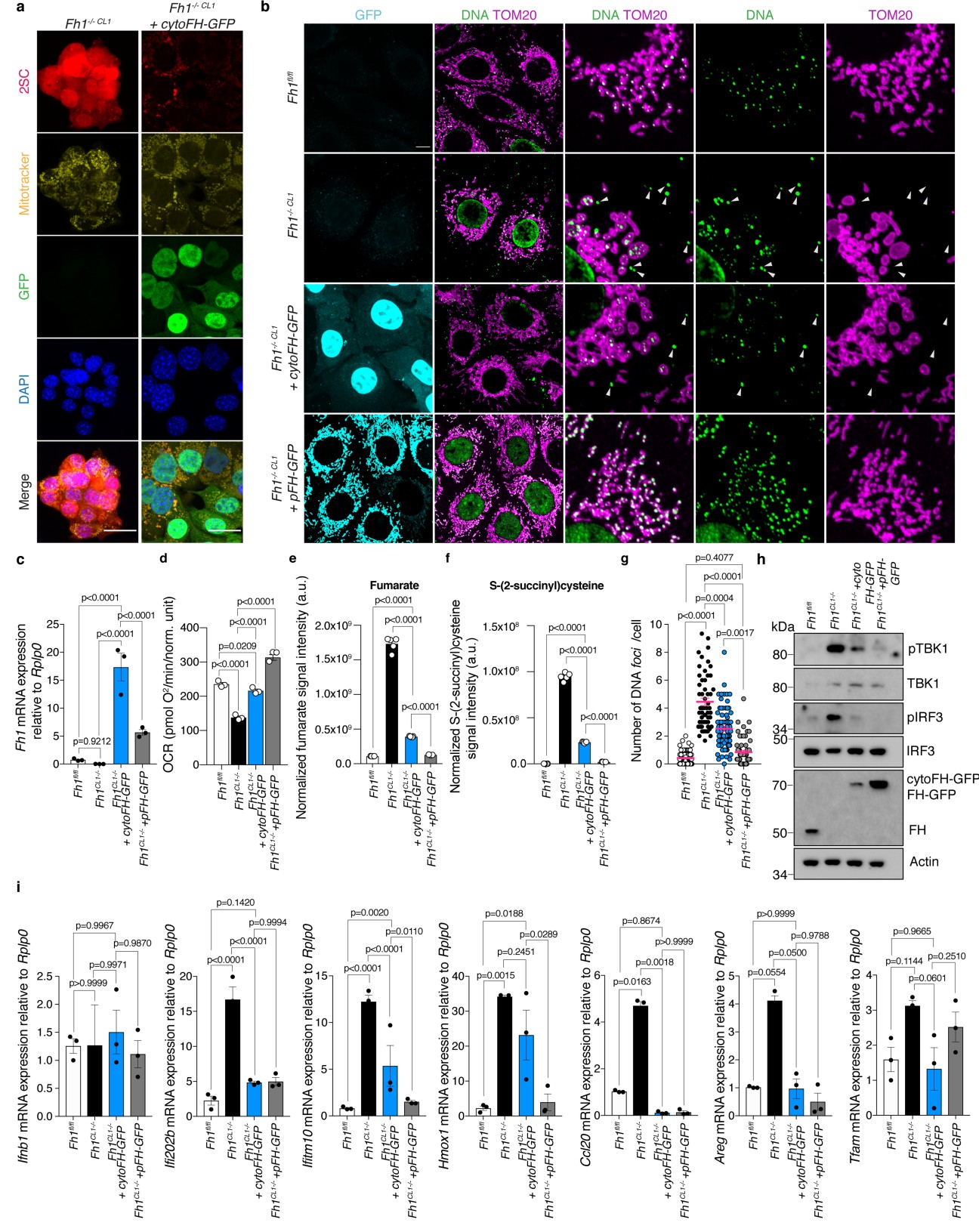

**Extended Data Fig. 6** | See next page for caption.

**Extended Data Fig. 6 | Cytosolic re-expression of FH only partially rescues the *Fh1* loss phenotype. a**, Representative confocal images of *cFh1*$^{-/-CL1}$ and *cFh1*$^{-/-CL1}$ cells stably expressing pcytoFH-EGFP (*cFh1*$^{-/-CL1}$+*cytoFh1-GFP*). Mitochondria and succinated proteins were labelled using Mitotracker and anti-2SC antibody, respectively. Nucleus was labelled using DAPI. Scale bar: 25 μm. **b**, Representative confocal images of *cFh1*$^{fl/fl}$ and *cFh1*$^{-/-CL1}$ cells stably expressing pcytoFH-EGFP (*cFh1*$^{-/-CL1}$+*cytoFh1-GFP*) or pFH-EGFP (*cFh1*$^{-/-CL1}$+ *pFH-GFP*). Mitochondria and DNA were labelled using anti-TOM20 and anti-DNA antibodies, respectively. White arrows indicate cytosolic DNA foci. Scale bar: 10 μm.

**c**, *Fh1* mRNA expression levels in *cFh1* cells measured by qRT-PCR. $n = 3$ independent experiments. **d**, Basal OCR in *cFh1* cells measured using Seahorse. $n = 5$ independent experiments. **e,f**, Relative abundance of fumarate (**e**), and 2SC (**f**), in *cFh1* cells measured by LC−MS. $n = 5$ independent experiments. **g**, Number of cytosolic DNA foci in *cFh1* cells from **b**. $n = 3$ independent experiments. **h**, Representative immunoblots of specified proteins in *cFh1* cells. **i**, mRNA expression levels of a panel of ISGs in *cFh1* cells measured by qRT-PCR. $n = 3$ independent experiments. Data are mean ± s.e.m. **c-g,i**, one-way ANOVA with Tukey's multiple comparison test.

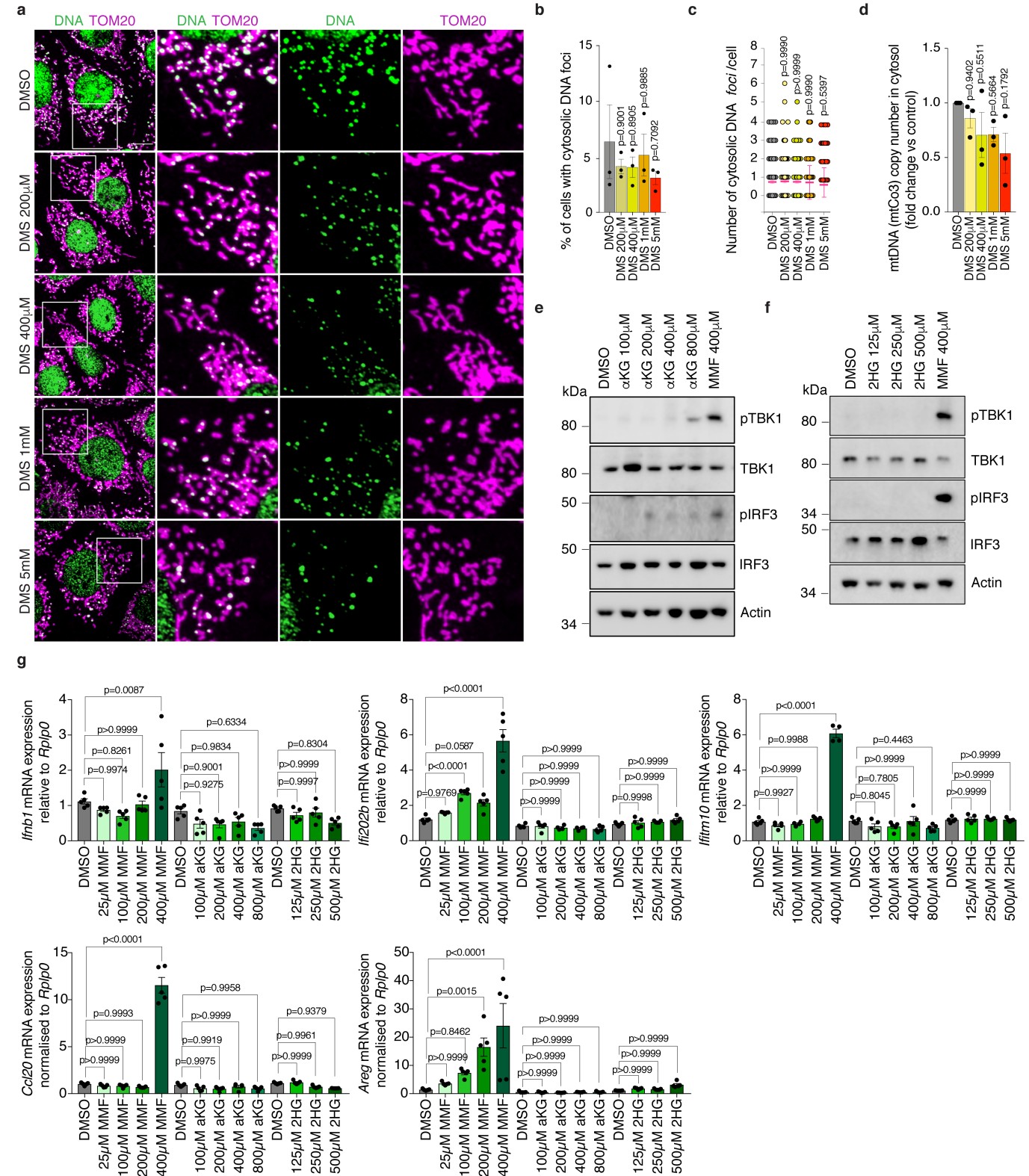

**Extended Data Fig. 7 | Effects of treatment with TCA-cycle intermediates on cytosolic mtDNA and the inflammation profile. a**, Representative confocal images of mitochondrial morphology (TOM20) and DNA foci (DNA) in *iFh1*[fl/flCL29] cells treated with dimethylsuccinate (DMS) or vehicle (DMSO) for 8 days at the indicated concentration. Scale bar: 10 μm. **b,c**, Quantification of *iFh1*[fl/flCL29] cells showing the percentage of cells with cytosolic DNA (**b**), and the number of cytosolic DNA foci per cell (**c**), from **a**. *n* = 3 independent experiments. **d**, Quantification of mtDNA copy number by ddPCR using a mtCo3 probe, from isolated cytosolic fractions of *iFh1*[fl/flCL29] cells treated with DMSO or DMS.

*n* = 3 independent experiments. **e,f**, Immunoblots of specified proteins in *iFh1*[fl/flCL29] cells treated with αKG (**e**), and 2HG (**f**), at indicated concentration. **g**, Expression of the ISGs *Ifnb1*, *Ifi202b*, *Ifitm10*, *Ccl20* and *Areg* in *iFh1*[fl/flCL29] cells treated with DMSO, MMF, αKG or 2HG at indicated concentration, measured by qRT-PCR. *n* = 5 independent experiments. Data are mean ± s.e.m. **b-d,g**, one-way ANOVA with Tukey's multiple comparison test, **b-d**, Bar graphs show the fold change expression, for which the expression in control samples was set to 1. p-values indicated above each bar are relative to the DMSO control.

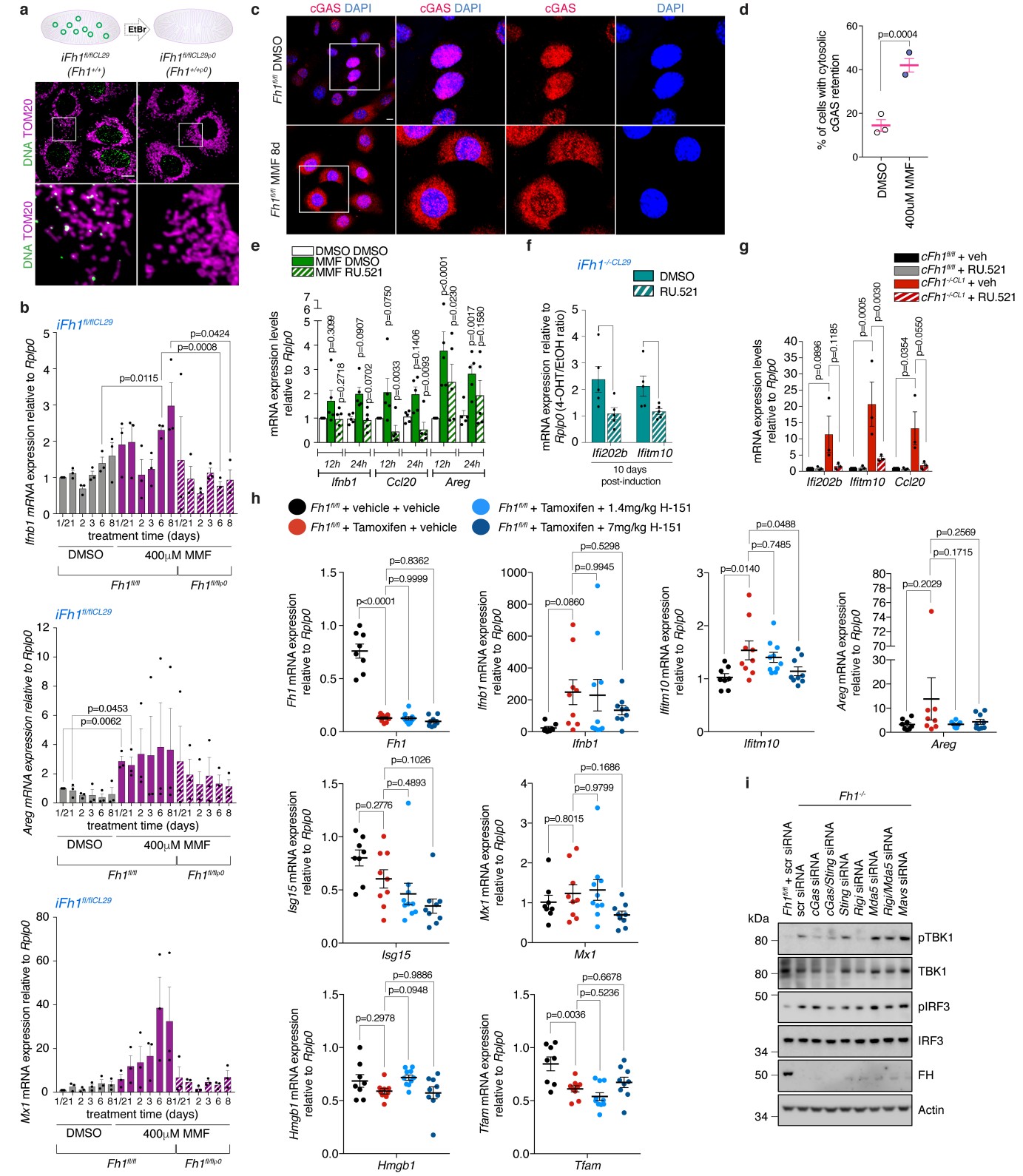

**Extended Data Fig. 8 | See next page for caption.**

**Extended Data Fig. 8 | In cellulo and in vivo inhibition of the cGAS–STING pathway. a**, Top: schematic showing depletion of mtDNA in $iFh1^{fl/flCL29}$ cells using ethidium bromide (EtBr). Bottom: representative confocal images of untreated $iFh1^{fl/flCL29}$ and mtDNA-depleted $iFh1^{fl/flCL29p0}$ cells. Mitochondria and DNA were labelled using anti-TOM20 and anti-DNA antibodies, respectively. Scale bar: 10 µm. **b**, mRNA expression of the ISGs *Ifnb1*, *Areg* and *Mx1* in non-induced $iFh1^{fl/flCL29}$ and mtDNA-depleted $iFh1^{fl/flCL29}$ ($Fh1^{p0}$) cell lines, treated with vehicle (DMSO) or 400 µM MMF for the indicated period of time, measured by qRT-PCR. $n$ = 3 independent experiments. **c**, Representative confocal images of $cFh1^{fl/fl}$ cells treated with 400 µM MMF or vehicle (DMSO) for 8 days. cGAS and nucleus were labelled using an anti-cGAS antibody and DAPI staining, respectively. Scale bar: 10 µm. **d**, Quantification of $cFh1^{fl/fl}$ cells showing the percentage of cells with cytosolic cGAS translocation from **c**. $n$ = 3 independent experiments. **e**, mRNA expression of a panel of ISGs in $iFh1^{fl/flCL29}$ cells treated with vehicle (DMSO) or MMF and DMSO or cGAS inhibitor RU.521 for the indicated period of time, measured by qRT-PCR. $n$ = 5 independent experiments. Bar graphs show the fold change expression, for which the expression in control samples was set to 1. **f**, mRNA expression of the ISGs *Ifi202b* and *Ifitm10* in $iFh1^{fl/flCL29}$ cells at day 10 post-induction, treated with vehicle (DMSO) or cGAS inhibitor (RU.521), measured by qRT-PCR. The 4-OHT *vs* vehicle (EtOH) ratios are shown. $n$ = 5 independent experiments. Bar graphs show the fold change expression, for which the expression in RU.521 samples was set to 1. **g**, mRNA expression of the ISGs *Ifi202b*, *Ifitm10*, and *Ccl20* in $cFh1^{fl/fl}$ and $cFh1^{-/-CL1}$ cells, treated with vehicle (DMSO) or cGAS inhibitor (RU.521), measured by qRT-PCR. $n$ = 3 independent experiments. (**h**) mRNA expression of a panel of ISGs in mouse kidney tissue treated with the STING inhibitor H-151, measured by qRT-PCR. $n$ = 9 mice per group. Black dots=$Fh1^{+/+}$ + vehicle, red dots = $Fh1^{-/-}$ + vehicle, light blue dots = $Fh1^{-/-}$ + 0.7 mg H-151, dark blue dots = $Fh1^{-/-}$ + 1.4 mg H-151. **i**, Immunoblots of specified proteins in $iFh1^{cl29}$ cells at 15 days post-induction transfected with indicated siRNA. Data are mean ± s.e.m. **b**, Students t-test (FDR two-stage Benjamini, Krieger and Yekutieli) at the corresponding time points, **d,f**, multiple Student's t-test, **e,g**, two-way ANOVA with Tukey's multiple comparison test, **h**, one-way ANOVA with Dunnett's multiple comparison test.

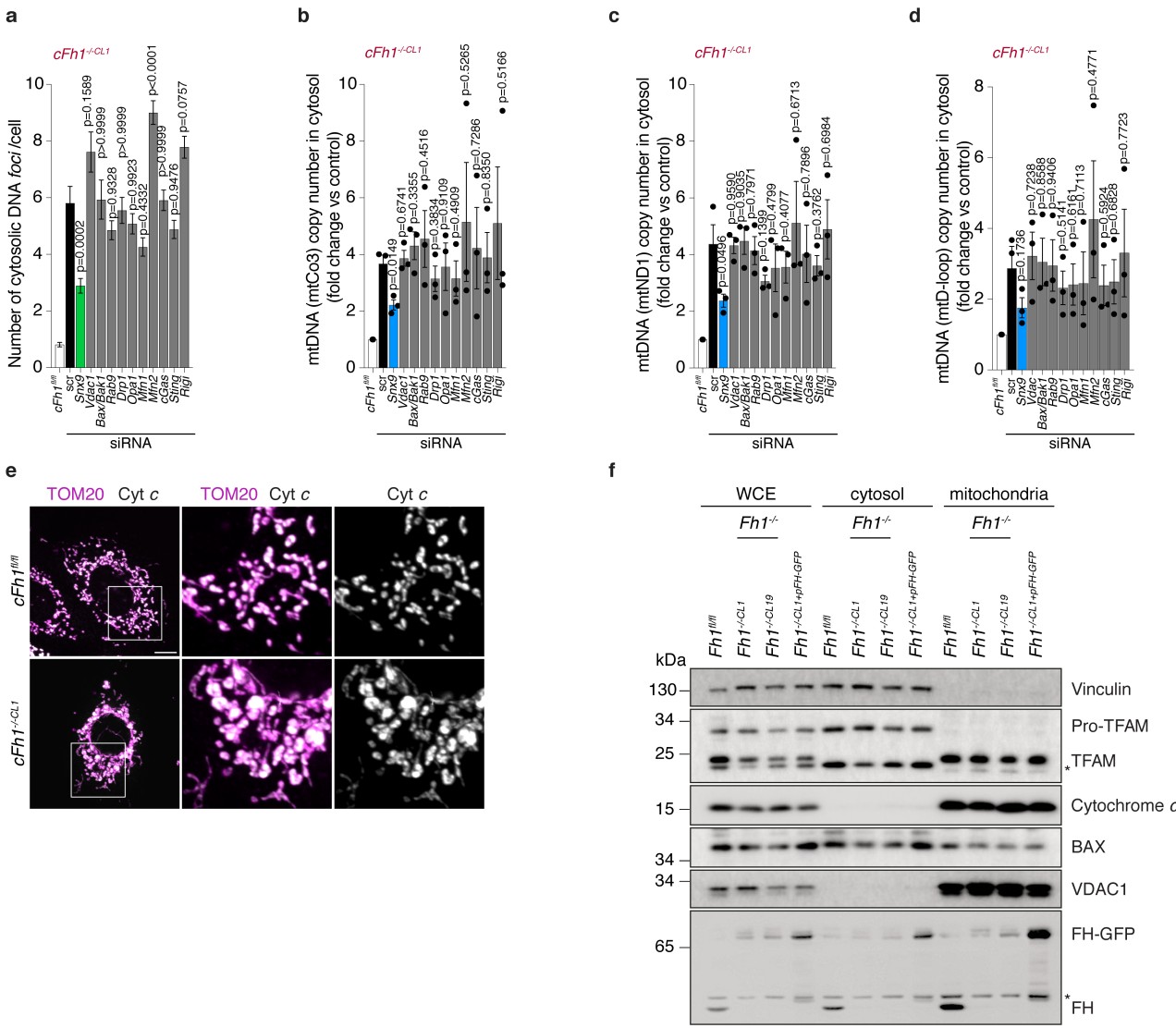

**Extended Data Fig. 9 | Targeted siRNA screen of key genes involved in mtDNA release and regulation of mitochondrial morphology. a**, Number of cytosolic DNA foci per cell in $cFh1^{-/-CL1}$ cells treated with indicated siRNAs. $n$ = 3 independent experiments. **b-d**, Quantification of mtDNA copy number by ddPCR using either a mtCo3 (**b**), ND1 (**c**) or D-loop (**d**) probe, from isolated cytosolic fractions of $cFh1^{-/-CL1}$ cells. $n$ = 3 independent experiments. **e**, Representative confocal images of $cFh1^{+/+}$ and $cFh1^{-/-CL1}$ cells. Mitochondria and cytochrome $c$ (Cyt $c$) were labelled using anti-TOM20 and anti-cytochrome $c$ antibodies, respectively. Scale bar: 10 μm. **f**, Immunoblots of specified proteins from whole cells (WCE), cytosol and heavy membrane (crude mitochondria) isolated fractions of $cFh1$ cells. Data are mean ± s.e.m. **a-d**, two-tailed unpaired t-test, p-values are indicated above each specific siRNA and relative to the scramble (scr) siRNA transfected control.

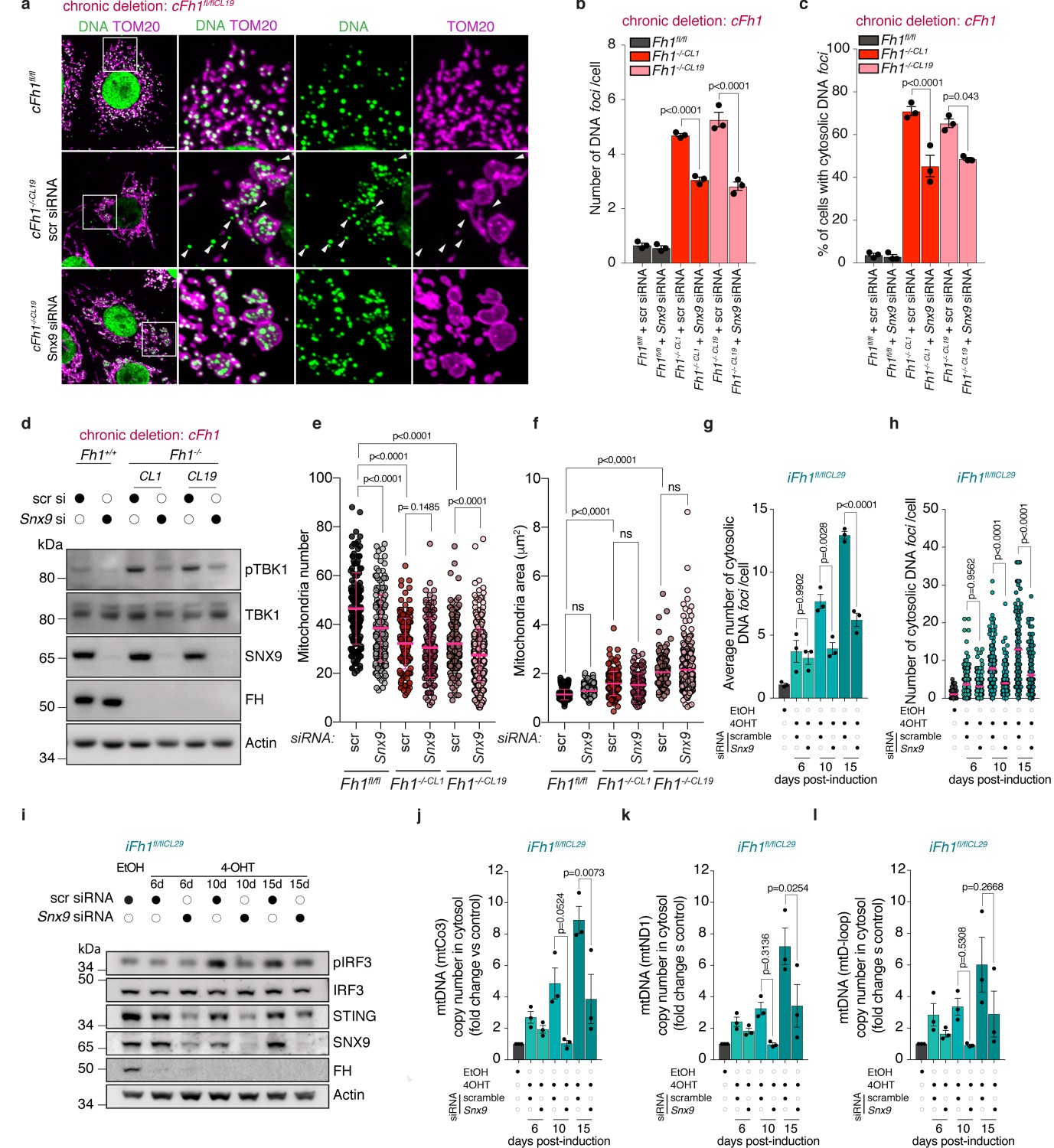

**Extended Data Fig. 10 | Loss of *Snx9* rescues mtDNA release and TBK1–IRF3 activation triggered by *Fh1* loss. a**, Representative confocal images of mitochondrial morphology (TOM20) and DNA foci (DNA) in *cFh1⁻/⁻CL19* cells transfected with scramble (scr) or *Snx9* siRNA. White arrows indicate cytosolic DNA foci. Scale bars: 10 µm. **b**, Number of cytosolic DNA foci per cell in *cFh1* cells. *n* = 3 independent experiments. **c**, Percentage of cells with cytosolic DNA foci in *c*Fh1 cells. *n* = 3 independent experiments. **d**, Immunoblots of specified proteins of *cFh1* cells. **e,f**, Quantification of mitochondrial number (**e**), and area (**f**), in *cFh1* cells. *n* = 3 independent experiments. **g,h**, Average number of cytosolic DNA foci per cell (**g**), and number of cytosolic DNA foci per cell (**h**), in *iFh1^fl/flCL29^* cells transfected with scr or *Snx9* siRNA and treated with either vehicle (EtOH) or 4-OHT (*iFh⁻/⁻CL29*) for the indicated period of time. *n* = 3 independent experiments. **i**, Immunoblots of specified proteins in *iFh1^CL29^* cells. **j-l**, Quantification of mtDNA copy number by ddPCR using either a mtCo3 (**j**), ND1 (**k**) or D-loop (**l**) probe, from isolated cytosolic fractions of *iFh1^fl/flCL29^* cells. *n* = 3 independent experiments. Data are mean ± s.e.m. **b,c,e-h,j-l**, one-way ANOVA with Tukey's multiple comparison test.

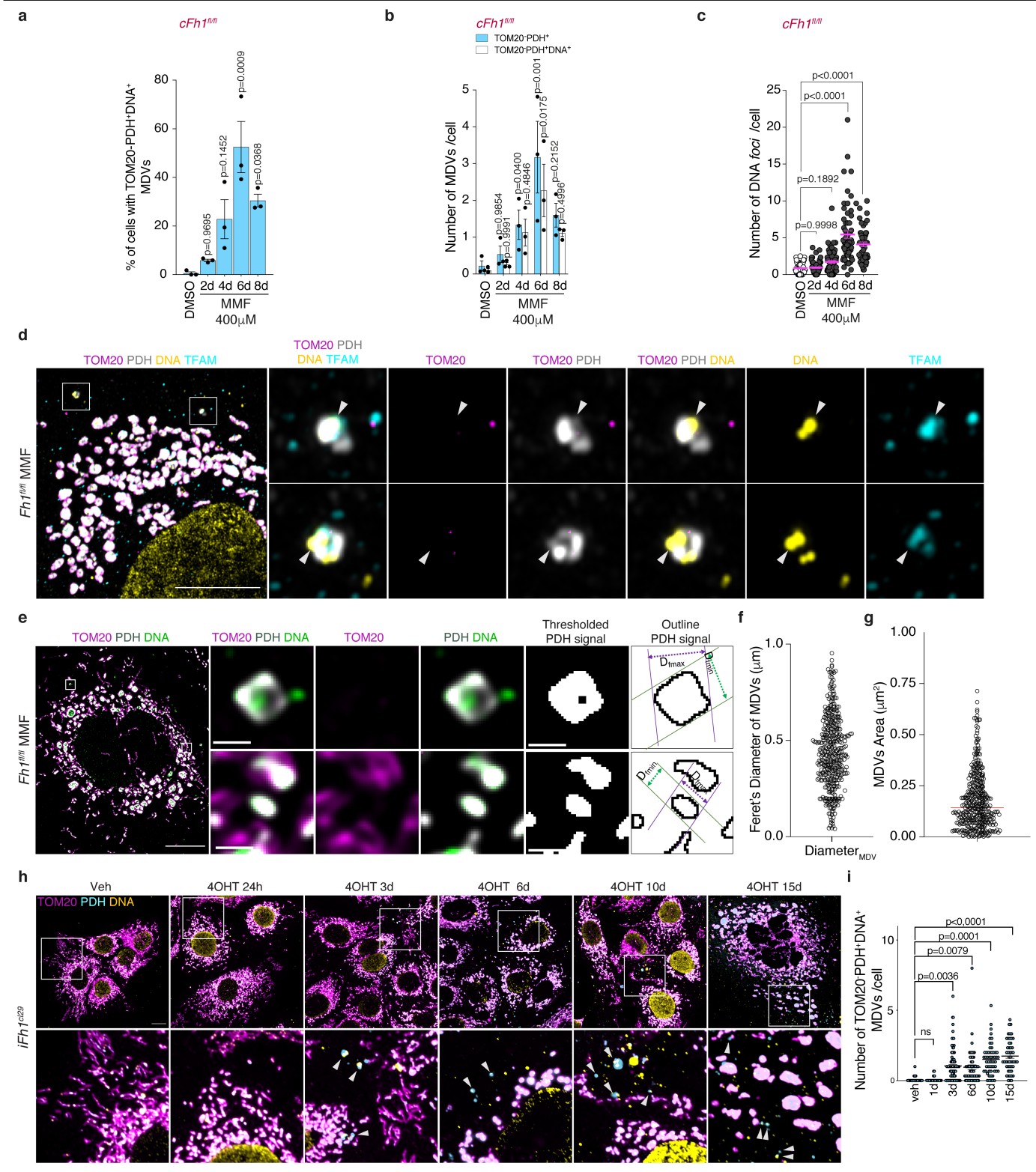

**Extended Data Fig. 11 |** See next page for caption.

**Extended Data Fig. 11 | Characterization of MDVs containing mtDNA.**
**a**,**b**, Percentage of cells harbouring TOM20$^-$PDH$^+$DNA$^+$ MDVs (**a**), and number of TOM20$^-$PDH$^+$ MDVs containing or not mtDNA per cell (**b**), in $cFh1^{fl/fl}$ cells treated with vehicle (DMSO) or 400 µM monomethyl fumarate (MMF) for the indicated period of time. $n$ = 3 independent experiments. p-values are indicated above each specific time point and relative to the untreated control for each of the one (a) or two (b) parameters. **c**, Number of cytosolic DNA foci per cell in $cFh1^{fl/fl}$ cells. $n$ = 3 independent experiments. **d**, Representative Airyscan super-resolution images of $cFh1^{fl/fl}$ cells treated with 400 µM MMF for 6 days. Mitochondria were labelled using anti-TOM20 and anti-PDH antibodies, TFAM using an anti-TFAM antibody, and DNA using an anti-DNA antibody. White arrows indicate TOM20$^-$PDH$^+$DNA$^+$TFAM$^+$ MDVs. Scale bar: 5 µm. **e**, Representative lattice super-resolution SIM image of MMF-treated $cFh1^{fl/fl}$ cells (6 days). Mitochondria were labelled using anti-TOM20 and anti-PDH antibodies, TFAM using an anti-TFAM antibody, and DNA using an anti-DNA antibody. Scale bar: 5 µm; magnification : scale bar: 1 µm. **f**,**g**, Quantification of the average maximal Feret's diameter (**f**), and average area (**g**), of TOM20$^-$PDH$^+$DNA$^+$ MDVs from **e**. $n$ = 3 independent experiments. **h**, Representative confocal images of $iFh1^{fl/flCL29}$ cells treated with either vehicle (ethanol; EtOH) or 4-OHT ($iFh^{-/-CL29}$) for the indicated period of time. Mitochondria were labelled using anti-TOM20 and anti-PDH antibodies, and DNA using an anti-DNA antibody. White arrows indicate TOM20$^-$PDH$^+$DNA$^+$ MDVs. Scale bar: 10 µm. **i**, Quantification of TOM20$^-$PDH$^+$DNA$^+$ MDVs from **h**. $n$ = 3 independent experiments. Data are mean ± s.e.m. **a**,**c**,**i**, one-way ANOVA with Tukey's multiple comparison test, **b**, two-way ANOVA with Tukey's multiple comparison test.

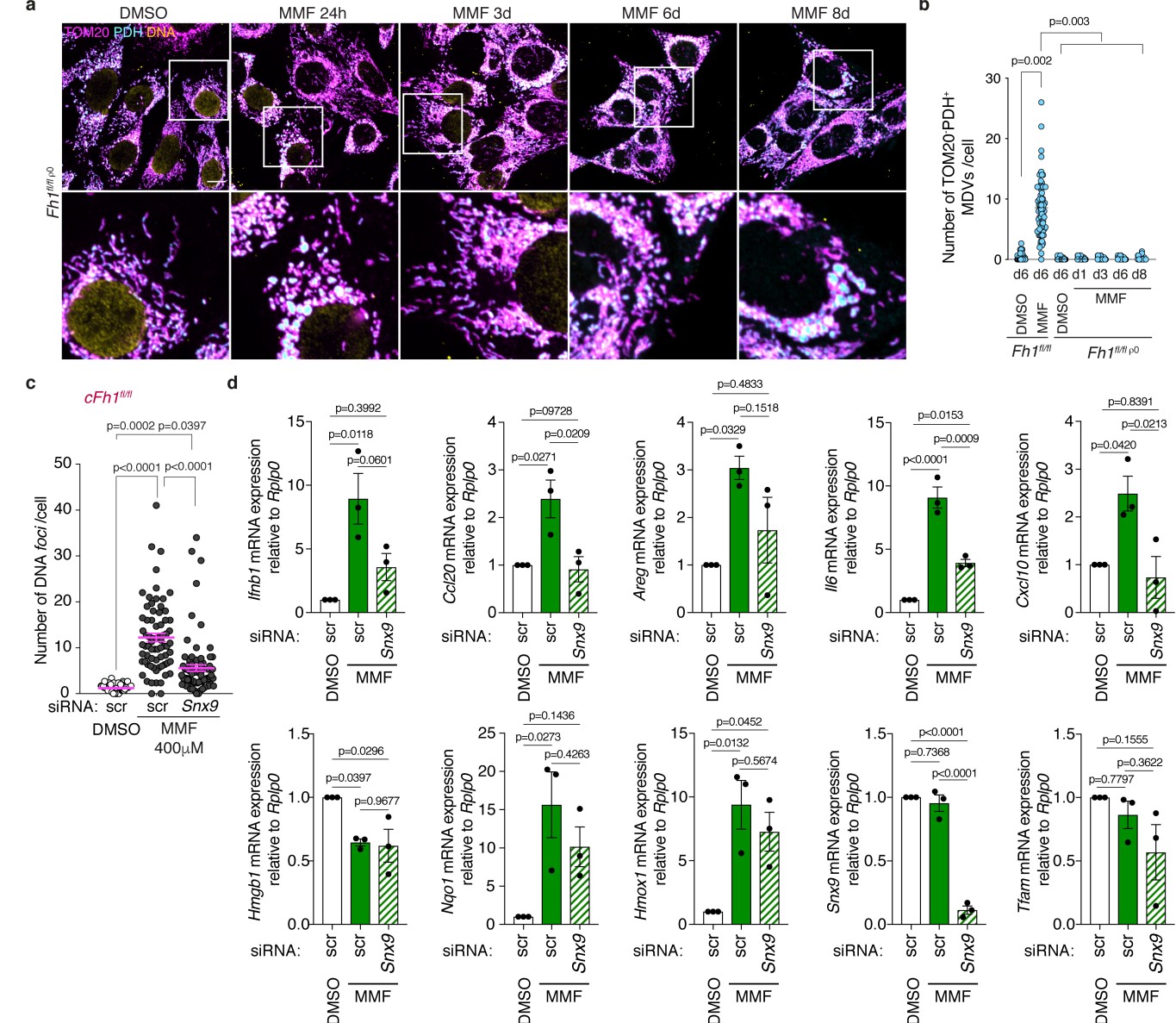

**Extended Data Fig. 12 | Loss of *Snx9* prevents MDV formation and inflammation induced by *Fh1* loss. a**, Representative confocal images of *iFh1^{fl/flCL29pO}* cells treated with 400 µM MMF for 1–8 days. Mitochondria were labelled using anti-TOM20 and anti-PDH antibodies, and DNA using an anti-DNA antibody. Scale bar: 10 µm. **b**, Quantification of the number of TOM20⁻PDH⁺ MDVs from **a**. *n* = 3 independent experiments. **c**, Number of cytosolic DNA foci per cell in *cFh1^{fl/fl}* cells pre-transfected with scramble (scr) or

*Snx9* siRNA and treated with vehicle (DMSO) or MMF for 6 days. *n* = 3 independent experiments. **d**, mRNA expression of a panel of ISGs in *cFh1^{fl/fl}* cells pre-transfected with scramble (scr) or *Snx9* siRNA and treated with 400 µM MMF or vehicle (DMSO) for the indicated period of time, measured by qRT-PCR. *n* = 3 independent experiments. Bar graphs show the fold change expression, for which the expression in control samples was set to 1. Data are mean ± s.e.m. **b**-**d**, one-way ANOVA with Tukey's multiple comparison test.

# Reporting Summary

## Statistics

For all statistical analyses, confirm that the following items are present in the figure legend, table legend, main text, or Methods section.

| n/a | Confirmed | |
|---|---|---|
| ☐ | ☒ | The exact sample size (*n*) for each experimental group/condition, given as a discrete number and unit of measurement |
| ☐ | ☒ | A statement on whether measurements were taken from distinct samples or whether the same sample was measured repeatedly |
| ☐ | ☒ | The statistical test(s) used AND whether they are one- or two-sided <br> *Only common tests should be described solely by name; describe more complex techniques in the Methods section.* |
| ☐ | ☐ | A description of all covariates tested |
| ☐ | ☐ | A description of any assumptions or corrections, such as tests of normality and adjustment for multiple comparisons |
| ☐ | ☐ | A full description of the statistical parameters including central tendency (e.g. means) or other basic estimates (e.g. regression coefficient) AND variation (e.g. standard deviation) or associated estimates of uncertainty (e.g. confidence intervals) |
| ☐ | ☒ | For null hypothesis testing, the test statistic (e.g. *F*, *t*, *r*) with confidence intervals, effect sizes, degrees of freedom and *P* value noted <br> *Give P values as exact values whenever suitable.* |
| ☐ | ☐ | For Bayesian analysis, information on the choice of priors and Markov chain Monte Carlo settings |
| ☐ | ☐ | For hierarchical and complex designs, identification of the appropriate level for tests and full reporting of outcomes |
| ☐ | ☐ | Estimates of effect sizes (e.g. Cohen's *d*, Pearson's *r*), indicating how they were calculated |

*Our web collection on statistics for biologists contains articles on many of the points above.*

## Software and code

Policy information about availability of computer code

Data collection

All softwares used for data collection are publicly or commercially available:
H&E stained tissue slides were generated using an Axioscan Z1 and ZEN Blue (v3.1) software. Immunofluorescence images were acquired using a Nikon Eclipse TiE inverted microscope with Andor Dragonfly 500 spinning disk system, equipped with a Zyla 4.2 PLUS sCMOS camera (Andor) coupled with Fusion software and using a 100X objective lens (NA1.4). Super-resolution images were acquired with an N-Structured illuminated Microscopy (N-SIM) microscope using a SR Apo TIRF 100× 1.49 N.A. oil objective and a DU897 Ixon camera (Andor). Raw images were computationally reconstructed using the reconstruction slice system from NIS-Elements software (Nikon). TFAM+-MDVs-related super-resolution images (Extended Data Fig. 12a) were acquired with a Zeiss LSM 880 microscope with an Airyscan detector (Carl Zeiss Microscopy, Jena, Germany), using a Zeiss 63x oil lens, numerical aperture 1.4, as the primary objective. Four colour excitation was performed with a Blue diode laser for 405 nm, Argon laser for 488 nm, He 543 laser for 561 nm and He 633 laser for 647 nm.. The Airyscan detector was used in SR mode utilising all 32 pinholes, thus increasing resolution to ~140 nm in x, y and z to capture the image. The image was reconstructed by pixel reassignment and deconvolution on Zen Black platform (Carl Zeiss Microscopy, Jena, Germany). The final image was produced on ImageJ by background subtraction (rolling ball 50) and one run of the smooth filter. MDVs super-resolution mages were acquired with a Zeiss Elyra7 equipped with Lattice SIM2 (Carl Zeiss Microscopy, Jena, Germany) with a 15 phased imaging protocol for all channels (Fig. 4h). A Zeiss 63x OIL: Plan-Apo 63x/1.4 Oil Corr WD: 0.35, was used as the primary objective to obtain representative images of individual cells. The images used for quantification of MDVs dimensions and number were acquired on a 40x OIL: Plan-APO 40x/1.4 Oil DIC (UV) VIS-IR objective using the SIM Apotome mode to maximize field of view and include more than one cell at a higher resolution.Electron microscopy images were acquired using a Tecnai G2 (FEI) transmission electron microscope operating at 100 kV equipped with a Veleta (Olympus Soft Imaging System) digital camera. RNA sequencing was performed using an Illumina HiSeq4000 sequencer. In vivo samples LC-MS was performed using a Dionex U3000 UHPLC system coupled to a Q Exactive mass spectrometer (Thermo Fisher Scientific) with a Sequant ZIC-pHILIC column (Merck Millipore). For in vitro experiments, LC-MS was performed on a QExactive Orbitrap mass spectrometer coupled to a Dionex UltiMate 3000 Rapid Separation LC system (Thermo Fisher Scientific), fitted with either a SeQuant Zic-HILIC column or a SeQuant Zic-pHILIC. Quantitative-RT

PCR was performed using an Applied Biosystems StepOne Plus or QuantSudio5 real-time PCR system. Genomic DNA amplification PCR was performed on Applied Biosystems Verity Thermo Cycler. Digital droplet PCR was performed using MODEL ASK VP (Biorad). Immunoblots images were acquired using a LICOR (model).

Data analysis | LC-MS acquired spectra were analysed using XCalibur Qual Browser and XCalibur Quan Browser software (Thermo Fisher Scientific) by referencing to an internal library of compounds. Calibration curves were generated using synthetic standards of the indicated metabolites. PCR experiments were analysed using the ∆∆Ct method. For RNA-Seq, reads were mapped to the mouse reference genome GRCm38 with the STAR (v.2.6.0c) aligner (Dobin, A. et al., 2013), filtered out using Cutadapt (version 1.10.0) (Martin, M., 2010), counted using the Bioconductor package Rsubread (v.1.28.1) (Liao, Y. et al., 2013) and gene annotated with GENCODE (release M17). Differential expression analysis was carried out with DESeq2 (v.1.18.1) (Love M. I. et al., 2014). Gene enrichment analysis was performed using the Gene Set Enrichment Analysis (GSEA) software from Broad Institute. Data compiling, processing and statistical analysis was performed using Microsoft Excel 2016 (v.16.16.27) and GraphPad Prism 7 (v.7.1.1) softwares.

For manuscripts utilizing custom algorithms or software that are central to the research but not yet described in published literature, software must be made available to editors and reviewers. We strongly encourage code deposition in a community repository (e.g. GitHub). See the Nature Portfolio guidelines for submitting code & software for further information.

## Data

Policy information about availability of data

All manuscripts must include a data availability statement. This statement should provide the following information, where applicable:
- Accession codes, unique identifiers, or web links for publicly available datasets
- A description of any restrictions on data availability
- For clinical datasets or third party data, please ensure that the statement adheres to our policy

All data are included within the article or the supplementary information. The source data for quantifications represented in all graphs plotted in figures and extended data figures are provided with this paper. Full versions of all gels and blots are provided in Supplementary Figures. Raw FastQ files for RNA-seq analyses are publicly available in the Gene Expression Omnibus (GEO) repository with the accession code GSE183745 (https://www.ncbi.nlm.nih.gov/geo/query/acc.cgi?acc=GSE183745).

## Human research participants

Policy information about studies involving human research participants and Sex and Gender in Research.

| | |
|---|---|
| Reporting on sex and gender | Findings apply to both genders and gender was not considered in the study design. Gender-specific data was not collected. Gender has no impact on the phenotype generated. |
| Population characteristics | Adult patients with suspected or confirmed inherited kidney cancer risk syndromes |
| Recruitment | patients were identified at the Specialist Multidisciplinary Team meeting and approached for informed consent at clinic consultations |
| Ethics oversight | Full ethics approval from West of Scotland Research Ethics Service, Dykebar Hospital, PAISLEY (REC reference: 16/WS/0039) |

Note that full information on the approval of the study protocol must also be provided in the manuscript.

# Field-specific reporting

Please select the one below that is the best fit for your research. If you are not sure, read the appropriate sections before making your selection.

☒ Life sciences ☐ Behavioural & social sciences ☐ Ecological, evolutionary & environmental sciences

For a reference copy of the document with all sections, see nature.com/documents/nr-reporting-summary-flat.pdf

# Life sciences study design

All studies must disclose on these points even when the disclosure is negative.

| | |
|---|---|
| Sample size | For in vivo experiments, the number of animals was determined based on the number of animals implemented in previously published papers and determined to be adequate based on the magnitude and consistency of measurable differences between groups. For all in vitro assay, n=minimum 3 and up to 6 biological replicates were used for higher reliability and sufficient size for statistical analysis. Each biological replicate is defined as an independent culture of cells. The sample size is described in the relevant Figure legends and/or method section. fig 1b,g n=7 min; fig 2a,b n=5; fig 2f,g,h,i,j,l,m,n ==3; fig 3b,c,d,e,f n=3; fig 3g,h,i n=4; fig 3k,p n=5; fig 3n n=9; fig 4d,e,g,h,j,k n=3; fig 4i,m n=4; fig 5b n=3; fig 5d n=5(N)/20(T); EDfig 2d,f n=4; EDfig 2e,f n=5; EDfig 2g,h,i,j,k,m,n,o n=3; EDfig 3 c,e,f,i,j,k,l,n n=3; EDfig 3h n=5; EDfig 4a,e,f,g,k n=3; EDfig 4h,i,j n=4; EDfig 5b,d,f n=3; EDfig 6b,c,d n=5; EDfig 6f,h n=3; EDfig 7c,d,g,i n=3; EDfig 7e,f n=5; EDfig 8b,c,d,e,f,g,h n=3; EDfig 8l n=5; EDfig 9a,c n=3; EDfig 9d,i n=5; EDfig 9e,f n=4; EDfig 9g n=9; EDfig 9l n=5; EDfig 10c,d,e,f n=3; EDfig 10i n=4; EDfig 11b,c,e,f,h,i,k,l,m,n,o,p,q n=3; EDfig 12d,f,g n=3 independent experiments. For image quantification, experiments are presented from 3 independent experiments (otherwise specified) with a total number of cells (n) |

analysed as: Fig. 2f-h: n= 173, 157, 158, 170, 160, 157, 162 for NT, vehicle, d1, d3, d6, d10, d15, respectively. Fig. 2i, j: n =30, 32, 35, 38, 32, 40, 36 and number of ROI analysed: n= 84, 85, 84, 85, 85, 84, 85 for NT, vehicle, d1, d3, d6, d10, d15, respectively. Fig. 3be, f: n=221, 192, 185 for vehicle, 200 μM MMF, 400 μM MMF, respectively. Fig. 4d, e: for TOM20-PDH+DNA+ vesicles analysis: n=166, 180, 167, 155, 170, for NT, 2d, 4d, 6d, 8d, respectively. Fig. 4e: for cytosolic DNA foci analysis: 165, 171, 162, 152, 160 for NT, 2d, 4d, 6d, 8d, respectively. Fig. 4g: n= 156; number of vesicles analysed: n=155. Fig.4h and Fig. ED12b: n= 408 ROI analysed. Fig. 4j, k: for TOM20-PDH+DNA+ vesicles analysis: n=158, 173, 158 for NT, MMF 6d si scramble, MMF 6 days si Snx9, respectively. Fig. 4k: for cytosolic DNA foci analysis: n=151, 166, 197, for NT, MMF 6d si scramble, MMF 6d si Snx9, respectively. Fig. ED2g-i: n= 166, 159, 166, 165, 175, 181, 170 for NT, vehicle, d1, d3, d6, d10, d15 respectively. Fig. ED2j, k: n =35, 42, 43, 47, 50, 48, 36 and number of ROI analysed: n= 99, 115, 137, 96, 127, 127, 76 for NT, vehicle, d1, d3, d6, d10, d15, respectively. Fig ED2l: number of mitochondria analysed: 11, 22, 79, 54, 46 for cFh1+/+, cFh1+/+ + Cre, cFh1fl/fl, cFh1-/-CL1, cFh1-/-CL19, respectively. Fig. ED3c, and h-i: n= 272, 246, 306, 300 for cFh1fl/fl, cFh1-/-CL1, cFh1-/-CL19, cFh1-/-CL1 + pFH-GFP. Fig. ED3e 30 cells were analysed per condition with a number of mitochondria analysed of n= 429, 827, 780 for cFh1fl/fl, cFh1-/-CL1, cFh1-/-CL19, respectively. Fig. ED3f: n= 33, 45, 67, 39 and number of ROI analysed: n= 79, 71, 164, 73 for cFh1fl/fl, cFh1-/-CL1, cFh1-/-CL19, cFh1-/-CL1 + pFH-GFP, respectively. Fig. ED4e-g: n=163, 161, 163 for Sdhbfl/fl, Sdhb-/-CL5, Sdhb-/-CL7, respectively. Fig. ED5d: n= 174, 165, 171 for cFh1fl/fl, cFh1-/-CL1 +pEGFP, cFh1-/-CL1 +pEGFP:NDI1, respectively. Fig. ED5f: n=221, 192, 185 for vehicle, 200μM MMF, 400μM MMF, respectively. Fig. ED6h: n= 169, 188 for DMSO and MMF 8d, respectively. Fig. ED7g: n= 182, 156, 189, 228 for cFh1fl/fl, cFh1-/-CL1, cFh1-/-CL1+pcytoFH-GFP, cFh1-/-CL1+pFH-GFP respectively. Fig. ED8b-d: n=152, 153, 162, 155, 155 for vehicle, DMS 200 μM, DMS 400μM, DMS 1mM, DMS 5mM, respectively. Fig. ED9c: n= 933, 660 for DMSO and MMF8d , respectively. Fig. ED10c: n= 175, 171, 183, 168, 166, 187, 188, 186, 154, 184, 161, 180, 169, for cFh1fl/fl, cFh1-/-CL1 si scramble, cFh1-/-CL1 si Snx9, cFh1-/-CL1 si Vdac1, cFh1-/-CL1 si Bax/Bak, cFh1-/-CL1 si Rab9, cFh1-/-CL1 si Drp1, cFh1-/-CL1 si Opa1, cFh1-/-CL1 si Mfn1, cFh1-/-CL1 si Mfn2, cFh1-/-CL 1si cGas, Fh1-/- si Sting, cFh1-/-CL1si Rig-I, respectively. Fig. ED10g: n=31, 32, for cFh1fl/fl and cFh1-/-CL1, respectively. Fig. ED11b-c: n=151, 154, 166, 159, 161, 163, for cFh1fl/fl, cFh1fl/fl si scramble, cFh1fl/fl si Snx9, cFh1-/-CL1 si scramble, cFh1-/-CL1  si Snx9, cFh1-/-CL19si scramble, cFh1-/-CL19si Snx9, respectively. Fig. ED11e, f: n= 55, 50, 53, 55, 60, 53 and number of ROI analyzed: n= 168, 162, 158, 348, 125, 155, 76 for cFh1fl/fl+ si scbl, cFh1fl/fl + si snx9, cFh1-/-CL1+si scbl, cFh1-/-CL1+ si snx9, cFh1-/-CL19+ si scbl, cFh1-/-CL19 + si snx9, respectively. Fig. ED11h, i: n=191, 167, 160, 179. 190, 147, 160, for vehicle, d6 si scramble, d6 si Snx9, d10 si scramble, d10 si Snx9, d15 si scramble, d15 si Snx9, respectively. Fig. ED11n-p: n=166, 180, 167, 155, 170, for NT, 2d, 4d, 6d, 8d MMF, respectively. Fig. ED11q: n=151, 166, 197 for NT, MMF 6d si scramble, MMF 6d si Snx9, respectively. Fig. ED12a: n=85. Fig. ED12d: n= 189, 155, 272, 159, 159, 144 for vehicle, 4OHT d1, d3, d6, d10, d15, respectively. Fig. ED12f: n= 170, 167, 145, 136, 133, 136, 139 for Fh1+/++DMSO, Fh1+/+ + MMF 6d, Fh1+/+r0+ DMSO Fh1+/+r0 + MMF 1d, 3d, 6d, 8d, respectively.

| | |
|---|---|
| Data exclusions | No data from in vivo samples were excluded from analysis but samples from some animals where the treatment had to be interrupted due to health concern were not collected.<br>In Extended Data Fig. 2c and Extended Data Fig. 6i, up to 2 biological replicates (out of 5 in total) per sample were excluded from all analyses due to bad sample quality/poor yield (data not shown). |
| Replication | In vivo and in vitro experiments were performed with at least 7 or a minimum of 3 (up to 6 times) biological replicates, respectively. All attempts at replication gave similar results and results were reliably reproduced with the same trend. |
| Randomization | Age-matched mice were randomly allocated into experimental groups.<br>For imaging, the cells and sample regions were evenly allocated and selected randomly. There was no requirement for randomization of other data. |
| Blinding | As no subjective measurements were done and the analysis were performed with quantitative instruments, no blinding was performed. |

# Reporting for specific materials, systems and methods

We require information from authors about some types of materials, experimental systems and methods used in many studies. Here, indicate whether each material, system or method listed is relevant to your study. If you are not sure if a list item applies to your research, read the appropriate section before selecting a response.

## Materials & experimental systems

| n/a | Involved in the study |
|---|---|
| ☐ | ☒ Antibodies |
| ☐ | ☒ Eukaryotic cell lines |
| ☒ | ☐ Palaeontology and archaeology |
| ☐ | ☒ Animals and other organisms |
| ☒ | ☐ Clinical data |
| ☒ | ☐ Dual use research of concern |

## Methods

| n/a | Involved in the study |
|---|---|
| ☒ | ☐ ChIP-seq |
| ☒ | ☐ Flow cytometry |
| ☒ | ☐ MRI-based neuroimaging |

## Antibodies

| | |
|---|---|
| Antibodies used | Immunobloting<br>goat polyclonal anti-FH/Fumarase (AbCam, ab113963) and anti-beta Actin [AC-15] (AbCam, ab6276), rabbit polyclonal anti-Grp75 (AbCam, ab2799), anti-Mfn1 (AbCam, ab126575), anti-P-IRF3S386 (AbCam, ab76493), and mouse monoclonal anti-VDAC1 (AbCam, ab14734). Mouse monoclonal anti-Actin (Sigma-Aldrich, A2228) and anti-Vinculin (Sigma-Aldrich, V4505). Mouse monoclonal anti-Drp1 (BD Transduction Laboratories, 611113), and anti-Opa1 (BD Transduction Laboratories, 612607). Rabbit polyclonal anti-Mfn2 (Cell Signaling Technology, 11925), anti-Bak (Cell Signaling Technology, 12105), anti-Bax (Cell Signaling Technology, 2772), anti-cGas (Cell Signaling Technology, 31659S), anti-Irf3 (Cell Signaling Technology, 4302S), anti-Stat1 (Cell Signaling Technology, 9172S), anti-P-Stat1Tyr701 (Cell Signaling Technology, 9167S), anti-TBK1/NAK (Cell Signaling Technology, 3013S), anti-Phospho-TBK1/NAKSer172 (Cell Signaling Technology, 5483S), anti-Rab9A (Cell Signaling Technology, 5118S), anti-Rig-1(Cell Signaling Technology, 3743S), and |

anti-Sting (Cell Signaling Technology, 50494S). Rabbit polyclonal anti-Snx9 (Proteintech, 15721-1-AP). Mouse monoclonal anti-Cytochrome c (BD Pharmingen, 556433). Rabbit polyclonal anti-mtTFAM (GeneTex , GTX103231).

Immunofluorescence
mouse monoclonal anti-DNA (Millipore, CBL186). Rabbit polyclonal anti-TOM20 (AbCam, ab232589), mouse monoclonal anti-TOM20 (AbCam, ab56783), and anti-PDH (AbCam, ab110333), were purchased from Abcam. Anti-Cytochrome c (BD Pharmingen, 556432). Donkey anti-mouse, goat anti-mouse IgG1, goat anti-mouse IgG2a, goat anti-mouse IgGM, and goat anti-rabbit Alexa Fluor 488, 565, 594 or 647 were used as secondary antibodies (all from Invitrogen). Rabbit anti-Cgas (Cell Signalling, D3080). Mouse anti-GM130 (BD BioSciences, 610822).

| Validation | All antibodies from commercial vendors were validated by the manufacturers on their websites. |
| --- | --- |

# Eukaryotic cell lines

Policy information about cell lines and Sex and Gender in Research

| Cell line source(s) | Cell lines source(s)<br>All cell lines were isolated from transgenic embryonic mouse kidneys. The constitutive cell lines cFh1FL/FL, cFh1-/- clones 1 and 19 are described in Frezza et al. DOI: 10.1038/nature10363. The constitutive cell line cFh1-/- clone 19 + pFH are described in Sciacovelli et al. https://doi.org/10.1038/nature19353. The inducible cell lines iFh1 clones 29 and 33 were isolated using the protocol described in Mathew at al. DOI: 10.1016/S0076-6879(08)01605-4. |
| --- | --- |
| Authentication | The cell lines were not authenticated. |
| Mycoplasma contamination | All used cell lines were routinely tested and confirmed negative for mycoplasma contamination. |
| Commonly misidentified lines<br>(See ICLAC register) | No commonly misidentified cell lines were used. |

# Animals and other research organisms

Policy information about studies involving animals; ARRIVE guidelines recommended for reporting animal research, and Sex and Gender in Research

| Laboratory animals | Mice were of mixed genetic background C57BL/6 and 129/SvJ. Animals were bred and maintained under specific pathogen-free conditions at the Breeding Unit (BRU) at the CRUK Cambridge Institute (Cambridge, UK). Fh1fl/fl and R26Creert2 mice were gifts from Prof Gottlieb (Technion, Israel Institute of Technology, Israel) and Dr Winton (CRUK, Cambridge Institute, Cambridge, UK), respectively. Experimental mice were homozygous for the conditional LoxP-exon3/4-LoxP Fh1 allele and expressed the Cre–recombinase-ert2 fusion under control of the ROSA26 promoter (Fh1fl/fl; R26 Creert2/Creert2). Littermate controls lacked the LoxP-exon3/4-LoxP allele but also expressed the Cre-ert2 allele under the control of the ROSA26 promoter (Fh1+/+; R26 Creert2/Creert2). Control mice were induced and sacrificed at the same time as their experimental littermates. In vivo experiments (tamoxifen induction) were performed under specific pathogen-free conditions at the Breeding Unit (BRU) at the CRUK Cambridge Institute (Cambridge, UK). All mouse experiments were performed in individually ventilated cages under the Animals (Scientific Procedures) Act 1986 (project licence P8A516814). The experiments were not randomised, and investigators were not blinded to treatment status during experiments and outcome assessment. |
| --- | --- |
| Wild animals | *Provide details on animals observed in or captured in the field; report species and age where possible. Describe how animals were caught and transported and what happened to captive animals after the study (if killed, explain why and describe method; if released, say where and when) OR state that the study did not involve wild animals.* |
| Reporting on sex | Findings apply to both genders and animal gender was not considered in the study design. Male and female animals were randomly assigned to study cohorts. Gender-specific data was not collected. Gender of the animals has no impact on the phenotype generated. |
| Field-collected samples | *For laboratory work with field-collected samples, describe all relevant parameters such as housing, maintenance, temperature, photoperiod and end-of-experiment protocol OR state that the study did not involve samples collected from the field.* |
| Ethics oversight | The Project Licence has been considered and granted by the UK Home Office; and ethically approved by the local establishment (LMB AWERB). |

Note that full information on the approval of the study protocol must also be provided in the manuscript.

