## [Peer Review File · Nature]

Manuscript Title: Fumarate induces vesicular release of mtDNA to drive innate immunity

Reviewer Comments & Author Rebuttals

Reviewer Reports on the Initial Version:

Referees' comments:

Referee #1 (Remarks to the Author):

This is an interesting paper that links endogenous fumarate elevation to release of mtDNA resulting in the activation of cGAS-STING dependent inflammatory response. I like the findings in the paper as it uncovers two important observations.

(1) The link between a metabolite and mtDNA is new finding. I am not aware of metabolites linked to mtDNA. Previous results have largely been confined to TFAM +/- or chemotherapeutic agents. This opens the possibility of metabolites under physiological or pathological conditions activating mtDNA dependent cGAS-STING inflammatory response. A key aspect of the paper is that elevation of fumarate but not succinate triggers release of mtDNA.

(2) Previous studies have surmised a variety of mechanisms to suggest how mtDNA is released. None of them to this reviewer are satisfactory. All of them rely on pathways linked to cell death including BAX/BAK, PTP or VDAC. This has raised the possibility that mtDNA is released in dying cells. How mtDNA is released from matrix without releasing cytochrome c has always puzzled me. However, this study links MDVs as a possible mechanism of mtDNA release. I like the Snx9-dependent release of mtDNA experiments and this could be broad mechanism by which mtDNA is released under a variety of stimuli beyond fumarate elevation.

I have a few key experiments that would bolster this excellent paper.

(1) Could the authors reconstitute the FH deficient cells with cytosolic FH? Does the fumarate action dependent on cytosolic elevation of fumarate?

(2) Previously the authors have shown that fumarate likely through succination causes defects in the mitochondrial respiratory chain in mouse cells. Human cancer cells with FH deficiency also demonstrate mitochondrial complex I defects. Could the authors reconstitute their cells with NDI1 that complements the loss of mitochondrial complex I? Does elevation of fumarate lead to complex I defects that are necessary for mtDNA release? How do the authors think the elevation of fumarate triggers MDVs?

(3) I would like to see RNAseq with siRNA against Snx9 in FH deficient cells. How much the inflammatory response is dependent on Snx9.

(4) cGAS-STING pathway occurs when cGAS-binds to STING dimers residing in the ER membrane leading to STING activation and trafficking to ER-GOLGI compartment where TBK1 is recruited. Do they observe Snx9 positive MDVs fusing or proximal to ER-GOLGI compartment?

(5) Beyond fumarate and succinate, the other key members of this metabolite family are alpha-

ketoglutarate (αKG) and L-2hydroxyglutarate (L-2HG). αKG is required for 75 plus dioxygenases including enzymes that participate in DNA, RNA, histone demethylases as well as prolyl hydroxylases for HIF and collagen regulation. Fumarate, succinate and L-2HG inhibit these enzymes. Thus, I am curious whether exogenous addition of cell permeable (octyl) L-2HG or αKG would trigger cGAS-STING inflammatory response. I would test whether this occurs before doing any experiments on mtDNA release with these two metabolites.

Referee #3 (Remarks to the Author):

The manuscript submitted by Zecchini et al focuses on the influence of Fh1 tumour suppressor and the chronology of its loss in the kidney, by developing a novel mouse model. By using this mouse model FH (in which FH loss is induced) and cell lines taken from kidneys of these mouse, the authors suggest that loss of Fh1 leads to fumarate accumulation, which results in mitochondrial network remodeling, mtDNA release into the cytosol, and activation of the innate immune response. Additionally, this work investigates the role of MDVs in fumarate accumulation, and suggests the fumarate phenotype is mediated through MDVs in a Snx9 dependent manner.

The manuscript is well explained and well written, and the experiments are mostly well planned and focused, which overall reflects the hard work of the research team. Nevertheless, I have some concerns regarding some of the methodological and analytical aspects of the research, and these should be addressed by the authors. Also, additional experiments should be done to further validate the suggested mechanism.

If the authors can address the concerns of this reviewer, I believe the manuscript potentially merits publication in Nature.

General comments:

The major concern is the physiological relevance of this work which is unclear. What would be the outcome of inhibition of the fumarate Snx9-Irf3 axis? Will inhibition of this axis effect tissue physiological outcome? Moreover, Fh1 KO causes changes in respiratory capability of the cells (Warburg effect) so it is not surprising that the expression of many genes change. In other words, it is not clear whether the loss of Fh1 directly causes the release of DNA which according to the authors, is suggested to affect the immune response. We know that loss of Fh1, causes in the long run, accumulation of fumarate, and effects histone di-methylases. The authors do not characterise the vesicles they claim are formed, whether these are specifically loaded with mtDNA, and released.

Specific comments:

Authors: Figure- 1(c) Metabolites abundance (peak intensity, arbitrary units (A.U.) in wild-type control (Fh1+/+) and Fh1-deficient (Fh1adult mouse kidney measured by liquid-chromatography mass-spectrometry (LC-MS).

Reviewer: Fig 1-(c)- The change in fumarate (the most relevant metabolite) levels, does not appear to be significant, and with such a change, one would not expect the outcome suggested by the

authors. While the other metabolites are affected, this is still not convincing. We agree with the claim that this mouse is an excellent model, but results must be refined.

Authors: Figure 1- (g) Expression levels of Interferon stimulated genes (ISGs) in Fh1^{-/-} vs Fh1^{+/+} kidney tissue at day 5 and 10 post-induction measured by qRT-PCR.

Reviewer: There is no reference in the text to the differences in the results between day 5 and day 10; on day 5 in most pro-inflammatory cytokines there is no significant difference between WT and KO mouse. What is the possible explanation?

Authors: Figure 2- (a) qRT-PCR (top) and representative immunoblots (bottom) showing expression levels of Fh1 in inducible iFh1 epithelial kidney cell lines clones 29 (iFh1fl/flCL29) treated with either vehicle (ethanol; EtOH) or 4-hydroxytamoxifen (4-OHT) (iFh^{-/-}CL29) for the indicated period of time.

Reviewer: The Western blot in EtOH, 3 days post induction shows a decrease in Fh1 levels even though it is a control and no influence should be observed.

Authors: Figure 2-(e) Representative confocal images of mitochondrial morphology and DNA foci in iFh1fl/flCL29 cells treated with vehicle (EtOH) or 4-OHT (iFh^{-/-} CL29) for 24 hours at day 6, 10 and 15 post-induction. Mitochondria and DNA were labelled using anti-TOM20 and anti-DNA antibodies, respectively. White arrows indicate cytosolic DNA foci. Scale bar: 10 μm.

Reviewer: Fig 2-(e)- The authors need to demonstrate by FISH the presence of other mitochondrial genes beside TOM20 within the cytosol. It is also important to characterize the physiological state of the cells and mitochondria, with respect to respiration and membrane potential.

Reviewer: Figures 2d and 3i- western blotting.

What are the differences between p-Tbk1 and Tbk1? This should be indicated in the figure legend (that it is the phosphorylated and unphosphorylated forms of each protein)

Authors: Figure 4. mtDNA is conveyed to the cytosol via mitochondrial-derived vesicles

Reviewer: Fig 4 - would be beneficial if the authors could comment on the average size of the vesicles and say more about their morphology.

Fig 4A- The authors claim that "double membrane-bound low-density content vesicular structures protruding from mitochondria". This is a very odd conclusion since double membrane vesicles cannot be seen in the EM Fig as indicated by the authors.

Referee #4 (Remarks to the Author):

In the manuscript entitled “Fumarate induces mtDNA release via mitochondrial-derived vesicles and drives innate immunity”, Zecchini and authors show that loss of the tumor suppressor fumarate hydratase leads to increased mtDNA release and subsequent activation of interferon stimulated genes (ISGs). They create an inducible model of FH1 deletion to investigate the temporal dynamics of fumarate induction and the downstream effects on mitochondrial dynamics and innate immunity. They use epithelial cell lines derived from the mouse kidney for the majority of their studies and show that loss of fumarate increases the number of swollen and elongated mitochondria, the release of mitochondrial DNA via mitochondrial-derived vesicles, and the induction of the cGAS-Sting-Tbk1 pathway. Mitochondrial-derived vesicles are highly understudied, and these findings indicate a new context and physiological role for these structures. While the authors have designed an inducible model to dissect the acute changes and immediate effects of FH1 loss, the timeline of events does not fully support the conclusions presented. The authors suggest that fumarate accumulation causes the effects on mitochondria, mtDNA release, and p-Stat1, but all these effects are seen between 3-6 days after induced FH1 loss, while fumarate is not increased until much later – between days 10-15. Later in their studies they add exogenous monomethylfumarate (MMF) which also affects mitochondria dynamics, mtDNA and pStat1 and ISGs. Does the dose of MMF correspond to the amount of fumarate at days 3-6, when there appears to be very little? Can the authors explain how fumarate is mediating these effects prior to its accumulation at days 3-6? Can low doses of MMF have similar effects to higher doses to recapitulate the effects of a small increase in fumarate? Because of this discrepancy, the data in the MMF treated cells is less convincing. The authors should show more of the mechanism in the inducible FH1 KO lines including the EtBr experiment to eliminate mtDNA as well as the effects on mitochondrial derived vesicles. Finally, how does this mechanism contribute to the tumor suppressive function of FH1 in HLRCC? Additional experiments to address this point would strengthen the relevance of these findings.

Referee #5 (Remarks to the Author):

Zecchini et al. report that deletion of fumarate hydratase (FH) in mouse kidneys and murine kidney epithelial cell lines leads to an accumulation of fumarate, subsequent disruptions to mitochondrial morphology, and the release of mtDNA into the cytosol. They propose that cytosolic mtDNA engages the cGAS-STING signaling axis leading to elevated expression of type I interferon responses in FH-deficient cells in vitro and both interferon and pro-inflammatory responses in FH-deficient kidneys in vivo. Furthermore, the authors show that addition of exogenous monomethylfumarate (MMF) is sufficient to phenocopy the alterations to mitochondrial morphology, mtDNA release, and innate immune activation in vitro. Finally, they report that mtDNA release is not dependent on Bax-Bak or other canonical players, but instead occurs via the Snx9-dependent generation of mitochondrial derived vesicles (MDVs). Depletion of Snx9 is sufficient to mitigate mtDNA release into the cytosol and inhibit activation of the innate immune signaling. Overall, this is an interesting study with robust datasets to support that FH deficiency triggers innate immune responses through mitochondrial stress. Although proposed by others, this paper also reveals that MDVs may be a mechanism of mtDNA release. Despite these positives, there are significant mechanistic gaps that must be further explored to solidify the proposed pathways. Specific comments follow below:

Major Comments:

1. Fig 1: The FH conditional deletion model here provides a powerful tool to examine the effects of FH loss on immune signatures in a tissue specific manner. However, the authors do not provide any data from other organs. Do the levels of fumarate increase in other organs, and if so, is this correlated with increasing immune signatures? Additionally, the authors state that there are no gross morphological changes in kidneys 10 days after tamoxifen-mediated Cre deletion of Fh1. A careful analysis of the H&E image of Figure 1 (d) seems to show many more nuclei between kidney cells. Moreover, even though FH expression is markedly down 5 days after tamoxifen exposure, the immune signature doesn't come up strongly until day 10. Both of these lines of evidence are indicative of elevated kidney-infiltrating immune cells, as opposed to a kidney epithelial cell-intrinsic immune response. Tissue RNAseq data will not distinguish kidney epithelial cell-intrinsic signatures of inflammation versus those generated by infiltrating immune cells. In agreement with the latter, the pathway analysis and gene expression signature seem to represent a generally more pro-inflammatory response as opposed to type I interferon. For example, IL-6, CD14, C3, TLR2, etc. are more consistent with a signature of macrophage/monocyte/neutrophil infiltration, as opposed to a cell-intrinsic type I interferon and interferon stimulated gene response. Given that the conditional knockout approach relies on the ubiquitously expressed ROSA-ERCre that will target FH in many body cells and tissues, it is important that the authors do more work to define the in vivo mechanisms.

2. The evidence in support of cGAS-STING-IFN-I pathway activation is generally not sufficient. Although crosses of FH cKO mice onto cGAS or STING KO mice to show ablation of the immune signature in vivo would be the most convincing, cGAS/STING inhibitors are readily available for mouse in vivo pre-clinical studies. These would be great to include as a way to link the in vivo phenotypes more cohesively with the cellular mechanisms presented in the later figures. In addition, the authors do not sufficiently explain how the kidney epithelial cell lines were immortalized and derived. This is important because many immortalization/transformation procedures render the cGAS/STING pathway non-functional at several levels (see paper from Stetson et al (Science 2015) on SV40 IgT and other oncogenes downregulating STING signaling). Finally, the authors never show that genetic ablation of cGAS/STING by siRNA, Crispr, etc is sufficient to ablate the interferon and ISG signature observed in vitro. The RU.521 experiments of Figure 2 (m) and 3 (n) do not convincingly show that cGAS inhibition abolishes the induction of immune genes after FH deletion of MMF treatment. It would be much more convincing to target these pathways using siRNA or other means (the authors use cGAS, STING, Rig-I siRNAs in Extended figure 6 but never examine immune gene expression after knockdown in FH deficient or MMF treated cells). Given that mtRNA has also been noted as a ligand driving immune activation downstream of mitochondrial stress, it is important to rule out the mtRNA sensors Rig-I/Mda5 and perhaps other nucleic acid sensing TLRs. This could be easily accomplished using the cell lines and siRNAs reported in this paper.

3. Comprehensive work from Dr. Frezza's lab and others has revealed fumarate as an oncometabolite in HLRCC via its actions on Keap1 and the NRF2 pathway, among others. Several recent papers have revealed that NRF2 pathway activation directly represses STING and counter regulates type I interferon responses (PMID: 30158636, 31487581, 31487581, 31487581). Fumarate accumulation after FH loss upregulates the NRF2 target Hmox1 in the cells used here (Extended Data Figure 2 (c)), so this suggests that NRF2 is activated. It is thus unclear how fumarate-dependent elevations in NRF2 would support robust STING activation. It is therefore important that the authors examine these connections and more thoroughly and explore innate immune pathways distinct from

cGAS-STING that might trigger the fumarate-dependent expression of innate immune genes in vitro (and in vivo).

4. The Snx9-MDV pathway as a route for mtDNA release in FH deficient cells is interesting. It is unclear, however, exactly how membrane bound mtDNA would be sensed by cGAS in the cytosol in a cell-autonomous manner. Do the authors see cGAS around the PDH/DNA+ MDVs? Is it possible these vesicles might be released and fuse with neighboring cells in vitro (or immune cells in vivo), leading to a paracrine activation of cGAS? It may be interesting to consider supernatant transfer from FH deficient cells onto WT and cGAS knockdown cells to further explore/clarify the mechanisms of mt-nucleic acid sensing. Additionally, the authors show that knockdown of Snx9 leads to a reduction in cGAS-STING pathway activation. Does this inhibition also lead to a reduction in expression of the ISGs and pro-inflammatory cytokines and chemokines reported in Figure 1 (f,g)? Similarly, does Snx9 knockdown have any effect on mitochondrial morphological changes?

5. The authors claim in the discussion that this paper represents the first report of a disease relevant mutation causing release of the mtDNA into the cytosol. This is a significant overstatement. Recently, human mutations in ATAD3A causing bona fide mitochondrial disease have been shown to liberate mtDNA into the cytosol and engage cGAS. Moreover, loss of TDP-43, CLPP, YME1L, OPA1 and others, all of which are linked to human disease, induce mtDNA release and activation of innate immunity. Perhaps the authors meant to restrict their discussion of impact to kidney diseases, but even so, loss of Tfam in kidney tubule cells has been shown to induce mtDNA leakage and cGAS-STING inflammation in vivo and in vitro. There is no evidence provided here to link the mtDNA release and immune phenotypes shown to HLRCC, so I believe it would be prudent to tone down these statements of novelty and properly reference the literature. It will not decrease the impact of the author's findings to reference these recent papers.

Technical Comments:

1. A significant portion of the analyses herein rely on manual scoring of immunofluorescence images. While manual scoring itself is not itself problematic, the individual(s) conducting the scoring should be blinded to the experimental groups wherever possible in order to minimize the impact of unintentional bias in the results. This is particularly important for subjective measures, such as mitochondrial morphology, where there is no clearly defined delineation between different morphological classifications.

2. Figures 3 (f-h, j, n), 4 (j) and Extended Data Figures 3 (j-l), 4 (i-l), 5 (a, h-k) – For these figures, each individual replicate for the control group has been set to one. This has the effect of artificially reducing the variance of the control group to zero, resulting in an inflated level of significance for groups that are being directly compared to the controls. Control values should be normalized to the mean of the control group, rather than set to one on an individual basis, preserving the variance and allowing an accurate significance level to be calculated.

4. Figures 3 (f-h), Figure 4 (j) and Extended Data Figures 2 (a), 4 (i-k) – The figure legend indicates that n = 3, but there are 4 replicates visible in the graphs.

5. Line 255-257 – Authors state that silencing of Snx9 reduces number of cytosolic DNA foci, but Extended Data Figure 7 (o) does not indicate a significant difference between scr and Snx9 siRNAs.

6. Extended Data Figure 7 (f-o) – Figure legend descriptions do not match corresponding figure panels.

7. Figure 3 (d,e) and Extended Data Figures 3 (h,i), 5 (f,g) – Each of these pairs of figures seem to present the same data (e.g., same means, same p values), just with two slightly different figure formats. What is the rationale for this duplication of data?

Author Rebuttals to Initial Comments:**Response to the referees Zecchini, Paupe et al**

Referee: plain black text

Authors: plain blue text

Reviewer #1 [mito biology/cancer metabolism]

This is an interesting paper that links endogenous fumarate elevation to release of mtDNA resulting in the activation of cGAS-STING dependent inflammatory response. I like the findings in the paper as it uncovers two important observations.

(1) The link between a metabolite and mtDNA is new finding. I am not aware of metabolites linked to mtDNA. Previous results have largely been confined to TFAM +/- or chemotherapeutic agents. This opens the possibility of metabolites under physiological or pathological conditions activating mtDNA dependent cGAS-STING inflammatory response. A key aspect of the paper is that elevation of fumarate but not succinate triggers release of mtDNA.

(2) Previous studies have surmised a variety of mechanisms to suggest how mtDNA is released. None of them to this reviewer are satisfactory. All of them rely on pathways linked to cell death including BAX/BAK, PTP or VDAC. This has raised the possibility that mtDNA is released in dying cells. How mtDNA is released from matrix without releasing cytochrome c has always puzzled me. However, this study links MDVs as a possible mechanism of mtDNA release. I like the Snx9-dependent release of mtDNA experiments and this could be broad mechanism by which mtDNA is released under a variety of stimuli beyond fumarate elevation.

I have a few key experiments that would bolster this excellent paper.

The authors thank the reviewer for their positive feedback, detailed discussion and pertinent suggestions. We believe we have addressed all the comments in the best possible way and hope the reviewer will be satisfied with our revised manuscript. To help the review of this document, we have incorporated panels from the figures as they appear now in the revised version of the manuscript. Please, note that the order of the sub-panels may differ from the figure in the manuscript for presentation purposes.

(1) Could the authors reconstitute the FH deficient cells with cytosolic FH? Does the fumarate action dependent on cytosolic elevation of fumarate?

The general issue of subcellular compartmentalization of metabolites, and, specifically, fumarate in the context of this paper, is very pertinent. As the reviewer suggested, we expressed a cytosolic form of Fh1 in our Fh1-deficient cell line to generate *Fh1*^{-/-CL1+cytoFh1} (**R#1 Figure 1a, b**). We observed a partial rescue of mitochondrial basal respiration (**R#1 Figure 1d**), a reduction in the levels of fumarate as well as a partial rescue in the levels of S-(2-succinyl)cysteine (2SC) (**R#1 Figure 1e, f**). In line with the original data we presented in the manuscript, we also observed a decrease in the number of cytosolic DNA *foci* in the *Fh1*^{-/-CL1+cytoFh1} cells (**R#1 Figure 1g**) together with a reduction in the activation of Tbk1 and Irf3 (**R#1 Figure 1h**), and a partial rescue in the transcriptional activation of downstream Interferon-stimulated genes (ISGs) (**R#1 Figure 1i**).

The overall reduction in fumarate observed in *Fh1*^{-/-CL1+cytoFh1} cells likely reflects a reduction of both mitochondrial and cytoplasmic pools because of mitochondrial fumarate moving into the cytosol. Importantly, however, immunofluorescence staining with the anti-2SC antibody confirmed a remaining positive signal in the mitochondria in the cytosolic FH rescue *Fh1*^{-/-CL1+cytoFh1} cells (**R#1 Figure 1a**), strongly suggesting an accumulation of mitochondrial fumarate in these cells. Therefore, we conclude that the cytosolic DNA *foci* phenotype observed in *Fh1*^{-/-CL1+cytoFh1} can be attributed to the remaining levels of mitochondrial fumarate. Of note, the re-expression of the full-length FH, which leads to its mitochondrial localization, fully rescued all the different phenotypes analyzed, including respiration, fumarate and 2SC levels, cytosolic DNA *foci* number, as well as the inflammation phenotype compared to the partial rescue observed in *Fh1*^{-/-CL1+cytoFh1} cells, highlighting the importance of the mitochondrial levels of fumarate in the different observed phenotypes. We have amended the manuscript and included this new important data (**revised manuscript Extended Data Fig. 7**).

R#1 Figure 1: Expression of the cytosolic Fh1 in Fh1-deficient cells. (a). Immunofluorescence for 2SC, Mitotracker, GFP and DAPI in Fh1-deficient (*Fh1*^{-/-CL1}) cells or *Fh1*^{-/-CL1} expressing a GFP-tagged cytoplasmic version of Fh1 (*Fh1*^{-/-CL1+cytoFh1-GFP}). (b). Immunofluorescence for GFP, TOM20 or DNA in control (*Fh1*^{fl/fl}), Fh1-deficient (*Fh1*^{-/-CL1}) cells or *Fh1*^{-/-CL1} expressing a GFP-tagged cytoplasmic version of Fh1 (*Fh1*^{-/-CL1+cytoFh1-GFP}) or a full version of Fh1 (*Fh1*^{-/-CL1+pFH-GFP}). (c). Fh1 mRNA expression in *Fh1*^{-/-CL1+cytoFh1-GFP} compared to *Fh1*^{fl/fl}, *Fh1*^{-/-CL1} and *Fh1*^{-/-CL1+pFH-GFP}. (d). Basal respiration in *Fh1*^{-/-CL1+cytoFh1-GFP} compared to *Fh1*^{fl/fl}, *Fh1*^{-/-CL1} and *Fh1*^{-/-CL1+pFH-GFP}. (e-f). Quantification of fumarate (e) and S-(2-succinyl)cysteine (f) levels determined by LC-MS in *Fh1*^{-/-CL1+cytoFh1-GFP} compared to *Fh1*^{fl/fl}, *Fh1*^{-/-CL1} and *Fh1*^{-/-CL1+pFH-GFP}. (g). Nucleic acid foci quantification in *Fh1*^{-/-CL1+cytoFh1-GFP} compared to *Fh1*^{fl/fl}, *Fh1*^{-/-CL1} and *Fh1*^{-/-CL1+pFH-GFP}. (h). Phosphorylation levels of Tbk1 and Irf3 in *Fh1*^{-/-CL1+cytoFh1-GFP} compared to *Fh1*^{fl/fl}, *Fh1*^{-/-CL1} and *Fh1*^{-/-CL1+pFH-GFP} as a proxy for the activity of the upstream nucleic acid sensor pathways. (i). Transcriptional activation of ISGs in *Fh1*^{-/-CL1+cytoFh1-GFP} compared to *Fh1*^{fl/fl}, *Fh1*^{-/-CL1} and *Fh1*^{-/-CL1+pFH-GFP}.

(2) Previously the authors have shown that fumarate likely through succination causes defects in the mitochondrial respiratory chain in mouse cells. Human cancer cells with FH deficiency also demonstrate mitochondrial complex I defects. Could the authors reconstitute their cells with NDI1 that complements

the loss of mitochondrial complex I? Does elevation of fumarate lead to complex I defects that are necessary for mtDNA release?

We thank the reviewer for requesting this critical control. In order to answer this comment, and as suggested by the reviewer, we constitutively expressed NDI1 in our *Fh1*-deficient cells to generate the *Fh1*^{-/-}*CL1*+EGFP:NDI1 line (R#1 Figure 2a, b). Despite a partial rescue of respiration, which was still significantly lower than control (R#1 Figure 2c) due to persistent defects in complex II and TCA cycle (as shown before in Tyrakis *et al.*, Cell Reports 2018), NDI1 expression did not significantly alter the release of mitochondrial nucleic acids into the cytosol (R#1 Figure 2d), nor the activation of the downstream cytosolic nucleic acid sensing pathway as indicated by unaffected Tbk1 and Irf3 phosphorylation (R#1 Figure 2e) or ISGs transcriptional activation (R#1 Figure 2f). Altogether, these data suggest that the phenotype we observe in *Fh1*-deficient cells is not triggered by an isolated defect in complex I. We have amended the manuscript accordingly to incorporate the data relating to NDI1 and complex I (revised manuscript Extended Data Fig. 5).

R#1 Figure 2: Expression of the NDI1 construct in *Fh1*-deficient cells. (a-b). Immunofluorescence of *Fh1*^{-/-}*CL1* cells either untransfected or expressing the GFP-only control plasmid or GFP/NDI1. pwiGFP/NDI1 contains an IRES site allowing the co-expression of GFP and NDI1 in the host cell. Red: DNA, green: GFP (a), qRT-PCR showing expression of the NDI1 construct in pwiGFP/NDI1 cells compared to untransfected cell lines (b). (c). Quantification of oxygen consumption in the indicated *cFh1* cell lines with Oroboros (upper panels); lower panels represent the percentage of ATP dependent, spare capacity

and rotenone sensitive oxygen consumption; n=3 independent experiments. (d). quantification of cytoplasmic nucleic acid foci in ND11-expressing cells compared to control cell lines. (e). Immunoblot showing the effect of ND11 expression on the activation of the Sting pathway in *Fhl1^{-/-CL1}* cells. (f). ND11 expression in *Fhl1^{-/-CL1 + ND11}* did not affect the accumulation of cytosolic mtDNA foci seen in cells not expressing it (*Fhl1^{-/-CL1}*) vs *Fhl1^{fl/fl}*.

How do the authors think the elevation of fumarate triggers MDVs?

Based on the new results, we hypothesize that a fumarate-dependent succination mechanism within the mitochondria could be involved in the formation of MDVs. Indeed, we now show that MMF treatment, which recapitulates the effects of FH loss and mtDNA release in the cytosol, induces a progressive increase of succinated proteins, accumulating particularly inside mitochondria (**revised version Extended data Fig. 6e and R#1 Figure 3**). Interestingly, the intensity of succination, here monitored by an anti-2SC antibody, correlated with the size of swollen mitochondria. Absolute metabolite quantification by LC-MS confirmed a rapid accumulation of 2SC in MMF-treated cells *comparable* to that observed in the inducible line *iFhl1^{-/-CL29}* (**R#1 Figure 3**).

a

b.

R#1 Figure 3: The effects of MMF on the fumarate pool. (a). 2SC immunofluorescence: representative confocal images of *cFhl1^{fl/fl}* treated with 400 μM MMF or vehicle (DMSO) for the indicated period. Mitochondria were labelled using anti-TOM20, and succinated substrates were labelled with an antibody recognising 2SC modifications. Scale bar: 10 μm. (b) Absolute metabolite

quantification using LC-MS of fumarate and succinate in MMF-treated cells for the indicated time period vs that observed in inducible the inducible line *iFh1^{-CL29}*.

Moreover, new experiments performed in mtDNA-depleted Rho 0 ($\rho 0$) cells showed that MMF treatment in these cells did not trigger the formation of TOM20⁺PDH⁺ MDVs (**revised manuscript Extended Data Fig. 12e, f and R#1 Figure 4**). This unexpected but crucial finding suggests that mtDNA is required for the formation of these particular MDVs. Interestingly, it has been reported that nucleoid-related proteins like TFAM and TWINKLE, can be succinated in human FH-deficient tumours and this has been associated with a loss of mtDNA copy number (Crooks *et al.*, Science Signalling 2021). While it is not clear at this point which protein is the main culprit, as cysteine residues of many proteins are succinated upon FH loss, this result lends support to the hypothesis that fumarate-driven succination may trigger MDVs formation. This hypothesis is further corroborated by the finding that only fumarate, but not other metabolites such as succinate, α KG, and 2HG, can elicit the formation of MDVs and mtDNA-release-dependent innate immunity (**revised manuscript Extended Data Fig. 8**). We have added this discussion in the revised version of the manuscript.

R#1 Figure 4: MDV formation in mtDNA-depleted cells. Right panel: representative confocal images of *iFh1^{fl/CL29} $\rho 0$* treated with 400 μ M MMF for the indicated period of time. Mitochondria were labelled using anti-TOM20 and anti-PDH antibodies, and DNA using an anti-DNA antibody. Scale bar: 10 μ m. Left panel: quantification of TOM20:PDH⁺ vesicles number; n= 3 independent experiments.

(3) I would like to see RNAseq with siRNA against Snx9 in FH deficient cells. How much the inflammatory response is dependent on Snx9.

In this study, we show that fumarate triggers the transcriptional activation of a specific subset of ISGs (see RNASeq dataset and **Fig. 1f**). ISGs expression is modulated by complex and intricate regulatory feedback loops (both positive and negative), often regulated at the post-translational level, acting at different times for different targets that make it difficult to assess this response as a whole at a single time point. In our manuscript, we observe different expression dynamics for each of our transcriptional targets that is likely to reflect such a complex transcriptional regulation. The phosphorylation of the downstream elements of the cascade *i.e.* Tbk1 and Irf3 might arguably constitute a more stable and reliable read-out of the pathway activation. Consequently, unless a time course is used, we believe that a genome-wide approach would yield limited information about the specific inflammatory pathway(s) that might be affected.

However, to better documents the inflammatory response observed upon loss of *Snx9*, a wider panel of targets where *Snx9* expression is knocked-down following treatment with MMF is now presented in **Extended Data Fig. 12g and R#1 Figure 5**.

[Text below redacted]

a

R#1 Figure 5: ISG in MMF-treated cells. (a). MMF- or vehicle-treated cells (72 hrs treatment) were treated with *Snx9* or scramble (scr) siRNA and the expression of a panel of ISGs and control genes was determined using qRT-PCR.

(4) cGAS-STING pathway occurs when cGAS-binds to STING dimers residing in the ER membrane leading to STING activation and trafficking to ER-GOLGI compartment where TBK1 is recruited. Do they observe *Snx9* positive MDVs fusing or proximal to ER-GOLGI compartment?

We thank the reviewer for this pertinent question. We have performed an additional set of experiments to address it. First, using microscopy analysis, we indeed showed the recruitment of activated p-TBK1 to the Golgi apparatus in MMF-treated cells compared to untreated cells (R#1 Figure 6), corroborating the classical cGAS/STING activation. In addition, we also analysed the presence of PDH⁺MDVs-containing DNA at the Golgi apparatus as suggested by the reviewer.

These results suggest that these *Snx9*-positive MDVs containing mtDNA are not targeted, neither fused or proximal to the ER-GOLGI compartment. For clarity and space limitation in the manuscript, we have decided to incorporate the *Tbk1*-P Golgi localisation upon MMF treatment (R#1 Figure 6), but not the negative results related to the localisation of MDVs-containing DNA at the Golgi-ER compartment (R#1 Figure 7). Further experiments using live cell imaging and super-resolution microscopy are needed to elucidate the final destination of these MDVs within the cell, and how/where the mtDNA contained in these MDVs is recognised by the DNA sensing machinery, but we believe this work requires an effort that goes beyond the scope of this manuscript.

R#1 Figure 6: Localisation of MDVs and TBK1 upon MMF treatment. (g). Representative confocal images of *cFh1^{fl/fl}* treated with 400 μ M MMF or vehicle (DMSO) for the indicated period of time. Mitochondria, Golgi apparatus, and p-Tbk1 were labelled using anti-TOM20 (magenta), GM130 (green) and p-Tbk1 (white) antibodies, respectively. Scale bar: 10 μ m. **(h)** Quantification of *cFh1^{fl/fl}* treated with 400 μ M MMF for the indicated period of time showing the percentage of cells with p-Tbk1 recruitment and co-localisation with the Golgi apparatus marker, GM130 from (f); n= 3 independent experiments.

(5) Beyond fumarate and succinate, the other key members of this metabolite family are alpha-ketoglutarate (aKG) and L-2hydroxyglutarate (L-2HG). α KG is required for 75 plus dioxygenases including enzymes that participate in DNA, RNA, histone demethylases as well as prolyl hydroxylases for HIF and collagen regulation. Fumarate, succinate and L-2HG inhibit these enzymes. Thus, I am curious whether exogenous addition of cell permeable (octyl) L-2HG or α KG would trigger cGAS-STING inflammatory response. I would test whether this occurs before doing any experiments on mtDNA release with these two metabolites.

We thank the Reviewer for this valid point that can provide an important control to our study. We followed the reviewer's advice and treated cells with the cell-permeable metabolites (octyl) L-2HG or α KG (**R#1 Figure 8i**). A recent paper assessed the role of L-2HG in lipopolysaccharide-activated macrophages (William *et al.*, J Biol Chem 2022). Using the same (octyl) L-2HG concentration range used in this paper, we showed that the addition of (octyl) L-2HG did not trigger cGas activation monitored by Tbk1 or Irf3

[Text above redacted]

phosphorylation (R#1 Figure 8j, k), nor transcriptional activation of our read-out panel of ISGs (R#1 Figure 8l). In parallel, cells were also treated with increasing concentrations of α KG. Similar to L-2HG treatment, no activation of either the signaling cascade (i.e. phosphorylation of Tbk1/Irf3) or ISGs transcription was observed in cells treated with increased α KG concentrations (R#1 Figure 8j, l). Together, these new important results indicate that the cytosolic nucleic acid sensors are not activated by either (octyl) L-2HG or α KG, and confirm the specificity of the described phenotype to increased levels of Fumarate.

We have amended the manuscript accordingly to incorporate this data (revised manuscript Extended Data Fig. 8i-l).

R#1 Figure 8: ISG expression upon incubation with α KG and 2HG. (i). schematic of the treatment. (j-k). Immunoblots showing phospho Tbk1 (p-Tbk1) and phospho Irf3 (p-Irf3) upon 2HG (j) and α KG (k) treatments, respectively. p-Tbk1 (phospho Tbk1), p-Irf3 (phospho Irf3). (l). qRT-PCR comparing the levels of transcriptional activation of ISGs upon 2HG and α KG vs MMF treatments. Note that cell-permeable esterified derivatives were used for both 2HG and α KG.

Reviewer #3 [fumarate/mito]

The manuscript submitted by Zecchini et al focuses on the influence of Fh1 tumour suppressor and the chronology of its loss in the kidney, by developing a novel mouse model. By using this mouse model FH (in which FH loss is induced) and cell lines taken from kidneys of these mouse, the authors suggest that loss of Fh1 leads to fumarate accumulation, which results in mitochondrial network remodeling, mtDNA release into the cytosol, and activation of the innate immune response. Additionally, this work investigates the role of MDVs in fumarate accumulation, and suggests the fumarate phenotype is mediated through MDVs in a Snx9 dependent manner.

The manuscript is well explained and well written, and the experiments are mostly well planned and focused, which overall reflects the hard work of the research team. Nevertheless, I have some concerns regarding some of the methodological and analytical aspects of the research, and these should be addressed by the authors. Also, additional experiments should be done to further validate the suggested mechanism. If the authors can address the concerns of this reviewer, I believe the manuscript potentially merits publication in Nature.

The authors thank the reviewer for the careful reading of the manuscript, detailed discussion and pertinent suggestions. We have now addressed all comments raised by the reviewer, which we trust has significantly increased the quality of our manuscript. To help the review of this document, we have incorporated panels from the figures as they appear in the revised version of the manuscript. Please, note that the order of the sub-panels may differ from the figure in the manuscript for presentation purposes.

General comments:

The major concern is the physiological relevance of this work which is unclear. What would be the outcome of inhibition of the fumarate Snx9-Irf3 axis? Will inhibition of this axis effect tissue physiological outcome? Moreover, Fh1 KO causes changes in respiratory capability of the cells (Warburg effect) so it is not surprising that the expression of many genes change. In other words, it is not clear whether the loss of Fh1 directly causes the release of DNA which according to the authors, is suggested to affect the immune response.

We apologise for the ambiguity in our manuscript and we have attempted to answer the reviewer's comment as best as possible below. The text has also been modified accordingly to better highlight the direct contribution of mtDNA release to the immune response, and new set of experiments/analyses have been performed to establish a link between FH-deficient renal cancer and the immune response.

(1) What would be the outcome of inhibition of the fumarate Snx9-Irf3 axis?

This is an interesting point which we also pondered on. Our study has uncovered a novel function of fumarate (*via* the loss of Fh1 activity) in the activation of Pattern Recognition Receptor (PRR) pathways both *in cellulo* and *in vivo*, that links Snx9-dependent mitochondria-derived vesicles (MDVs) formation to the downstream activation of the Sting/Tbk1/Irf3 signalling cascade that ultimately results in the stimulation of an inflammatory response *via* sustained ISGs transcriptional activation. As mutations in Fumarate Hydratase (FH) are associated with Hereditary Leiomyomatosis and Renal Cell Cancer (HLRCC), which results in the development of an aggressive form of kidney cancer, we hypothesise that a chronic low-grade inflammation resulting from fumarate-dependent activation of the Snx9-Irf3 pathway could contribute to tumorigenesis in these patients. In support of this hypothesis, we provide additional data in the revised version of the manuscript that show a robust immune signature associated with tumour tissue from HLRCC patients (**R#3 Figure 1a, b**), corroborating our *in vivo* and *in vitro* data. We also showed using deconvolution from bulk transcriptomics data (see *Material and Methods* for the experimental details) that the contribution of immune cells to this signature is higher in FH-deficient HLRCC tumours compared to other renal tumour types (**R#3 Figure 1c**); indicating an immune environment in FH-deficient tumours. In addition, we showed the presence of IL6 and IL10 in the supernatant of HLRCC tumour tissue (**R#3 Figure 1d**), corroborating the activation of the innate immune response. Together, this new set of data supports the critical role of this pathway not only *in vivo* in our mouse model, but also in patients.

We agree with the Reviewer that it would be interesting to elucidate the physiological outcome of the inhibition of this pathway, but this will require the generation of new mouse models (e.g. Fh1-deficient mouse model crossed with cGAS-KO or Sting-KO mouse model), which is not feasible in the time allowed for the revision of this manuscript, and we think beyond the scope of this initial study. It is worth noting here that currently, none of the mouse models of Fh1 loss leads to overt carcinoma, which would also complicate the design of this experiment. We are currently working to overcome this issue, and we hope to address this relevant question in a near future. The data presented below has now been incorporated in the revised manuscript (Fig. 5a-d).

R#3 Figure 1: Inflammatory signature in FH-deficient tumours. (a) Gene Set Enrichment Analysis of gene expression profiles of normal kidney (3 technical replicates) and a renal tumour (3 technical replicates) from a HLRCC patient carrying a germline mutation in the fumarate hydratase (FH) gene (Ashrafiyan *et al.*, *Cancer Res* 2010) showing enrichment in immune response pathways. NES=Normalised Enrichment Score. (b), qRT-PCR showing the expression of FH and ISGs in tumour tissue from HLRCC patients vs surrounding healthy tissue. n=3patients with 3 technical replicates. (c) Cellular composition of the bulk RNA sequencing datasets from FH deficient RCCs, SDH deficient RCCs and common RCC subtypes from TCGA (as described in Materials and Methods). (d). IL6 and IL10 ELISA on human normal vs HLRCC tumour patient samples. N=Normal (healthy individuals), T=Tumour (HLRCC patients). N=5 and 20 patients serum samples for N and T, respectively.

(2) Will inhibition of this axis effect tissue physiological outcome?

Our study shows that the Snx9-Irf3 signalling pathway is stimulated in Fh1-deficient cells and in kidney from our new generated inducible mouse model. In this revised version of our manuscript, we showed that the inhibition of the Snx9-Irf3 axis, either via the silencing of Snx9 or the *in vivo* inhibition of cGAS (revised Fig. 3n and ED Fig. 9g) leads to a significant reduction in the ISGs expression, suggesting that the inhibition of this pathway could lead to an anti-inflammatory effect that could contribute to tumour suppression. Similarly to our reply to question (1) of this reviewer, elucidating the *in vivo* outcome of inhibition of this pathway is beyond the scope of this initial study and will be investigated in a follow-up study.

(3) Moreover, Fh1 KO causes changes in respiratory capability of the cells (Warburg effect) so it is not surprising that the expression of many genes change. In other words, it is not clear whether the loss of Fh1 directly causes the release of DNA which according to the authors, is suggested to affect the immune response.

This is an appropriate comment, which we would like to elaborate on here. Several crucial findings support the specific role of Fh1 loss and increased levels of fumarate in causing the release of mtDNA to trigger the innate immune response. First, to rule out that the observed mtDNA release is a general consequence of TCA cycle inhibition and mitochondrial dysfunction (and the “Warburg effect” mentioned by the referee), we duplicated most of our experiments in cells lines harbouring a deletion of succinate dehydrogenase b (*Sdhb*), another component of the TCA cycle and complex II of the electron respiratory chain, also linked to hereditary cancers. As reported by Cardaci and co-workers, the loss of *Sdhb* activity results in a Warburg-like phenotype and a near-complete loss of oxygen consumption (Cardaci *et al.*, Nat Cell Biol, 2015). However, unlike the loss of Fh1 activity, *Sdhb* deletion did not induce the cytosolic release of mtDNA, Tbk1/Irf3 activation or transcriptional activation of ISGs (**revised manuscript Extended Data Fig. 4**). The lack of an inflammatory response in *Sdhb*-deficient cells was corroborated by the absence of immune infiltrates in SDH-deficient renal tumours in our new analyses; in contrast with the significant inflammatory response in FH-deficient tumours (**revised manuscript Extended Data Fig. 5c**). Second, we also showed that the expression of the NADH oxidase NDI1, which rescues complex I defects observed in Fh1-deficient cells (Tyrakis *et al.*, Cell Reports, 2017 and response to referee 1, point (2)), did not prevent mtDNA release, activation of TBK1/Irf3 nor the transcriptional upregulation of ISGs, corroborating the prerequisite and specificity of Fh1 loss and fumarate accumulation - rather than a bioenergetic defect - for mtDNA release driving the immune response. In further confirmation of this hypothesis, our study reveals that the observed effect is mediated specifically by fumarate but not succinate, which accumulates upon loss of *Sdhb*. Indeed, in contrast to fumarate treatment, succinate treatment showed no effect on the cytosolic release of mtDNA, Tbk1/Irf3 phosphorylation nor ISGs transcriptional activation (**revised manuscript Extended Data Fig. 8a-h**). In the revised version of the manuscript, we have also extended the treatment to other known “oncometabolites”, namely α KG and 2HG (see response to referee 1, point (5)). These metabolites are also generated due to alterations to the TCA cycle. However, treatment with these metabolites had no effect on our specific read-outs (**revised manuscript Extended Data Fig. 8i-l**). Finally, we also show that re-expression of full-length and also cytosolic-only Fh1 in Fh1-deficient cells rescues the phenotype (**revised manuscript Extended Data Fig. 7**). Of note, the cytosolic rescue of Fh1, which still accumulates mitochondrial fumarate even though to a lower level than the *Fh1*^{-/-}, reduces only partially the inflammatory phenotype, thus ascribing to mitochondrial fumarate the trigger of the cascade we describe. Altogether, these results indicate that the phenotype we observe in Fh1-deficient cells is not triggered indirectly by changes in the respiratory capability of the cell but are due to the specific loss of Fh1 activity and fumarate accumulation. We thank the reviewer for this useful comment and trust that the new data described here and that we have now incorporated in the revised manuscript has alleviated the reviewer’s concerns.

(4) We know that loss of Fh1, causes in the long run, accumulation of fumarate, and effects histone dimethylases. The authors do not characterise the vesicles they claim are formed, whether these are specifically loaded with mtDNA, and released.

We agree with the reviewer that the specific loading of vesicles with mtDNA is an important point for this work. Mitochondrial-derived vesicles (MDVs) are characterised by their size but, more importantly, by cargo specificity. Indeed, MDVs only harbour a subset of mitochondrial proteins, which allows their identification and characterization compared to mitochondria. In the manuscript, we have shown that upon fumarate accumulation, cells exhibited the presence of MDVs in the cytosol compared to control cells. These MDVs are characterized by the presence of the mitochondrial matrix marker, Pyruvate Dehydrogenase (PDH), but negative for the outer mitochondrial membrane marker, TOM20, two well-known markers of different types of MDVs (Sugiura *et al.*, EMBO J 2014, König *et al.*, Nat Cell Biol. 2021, Soubannier *et al.*, Curr Biol. 2012)

To the question of whether the vesicles are actually loaded with mtDNA, our original microscopy data showed co-localisation of DNA *foci* with PDH, but not TOM20, in the vicinity of MMF-treated mitochondria, indicating that these DNA *foci* contained in MDVs are likely of mtDNA origin (**revised manuscript Fig. 4c** and image below, red arrows).

Representative N-Structured illumination microscopy super-resolution images of cFh1fl/fl cells treated with 400 μ M MMF for 6 days. Red arrows indicate TOM20-PDH+DNA+ vesicles budding events; blue arrows indicate TOM20-PDH+DNA+ released vesicles. Scale bar: 5 μ m. Magenta square: magnification of the panels above.

We have also corroborated the specific presence of mtDNA in the cytosol using ddPCR on cytosolic fractions with specific DNA probes targeting different genes of the mtDNA: Cox3, ND1 and D-loop (**revised manuscript Fig. 2l-n, Fig. 3g-i, Fig. 4i, Extended Data Fig. 2m-o, Extended Data Fig. 3j-l, Extended Data Fig. 4h-j and Extended Data Figure 8e-g**).

Nevertheless, to address the reviewer's request, we have performed additional experiments which show that these MDVs also contain TFAM, an mtDNA-binding protein that is essential for mtDNA packaging into nucleoids: TFAM was observed in around 65% of these MDVs (**R#3 Figure 2 and revised manuscript Fig. 4f,g**) and was shown to colocalize with DNA inside the PDH⁺DNA⁺TOM20⁻ MDVs by immunofluorescence and super-resolution microscopy (**R#3 Figure 3a, revised manuscript Extended Data Fig. 12a**). Given the known strong binding activity of TFAM to mtDNA, this latter data mitigates the question about the origin of DNA observed in these MDVs.

We also have shown that these MDVs containing mtDNA were released from the mitochondria since silencing of Snx9, a critical protein regulating MDVs release (Matheoud *et al.*, Cell 2016) inhibited not only the presence of DNA *foci* and PDH⁺DNA⁺TOM20⁻ MDVs in the cytosol (measured by immunofluorescence analysis, **revised manuscript Fig. 4j,k, Extended Data Fig. 10a, Extended Data Fig. 11a-c, h-i, g, n-q**) and presence of mtDNA in the cytosol (measured by ddPCR, **revised manuscript Fig. 4i and extended Data Fig. 10d-f**), but also the immune response monitored by Tbk1/Irf3 activation by immunoblots (**revised manuscript Fig. 4l and Extended Data Fig. 11d,j**) and ISGs expression measured by RT-qPCR (**revised manuscript Fig. 4m and Extended Data Fig. 13**). Together, these data strongly support the role of Snx9 in the release of mtDNA-loaded MDVs and driving the immune response.

Finally, as suggested by the reviewer, we have strengthened the characterization of these MDVs using Lattice NSIM super-resolution microscopy analysis. Our new data (**R#3 Figure 4 and revised manuscript Fig. 4h and Extended Data Fig. 12a**) not only confirmed the presence of TFAM and DNA in PDH⁺TOM20⁻ negative MDVs, but also fully characterised their size, diameter, and shape.

With this new set of experiments, we trust that we have addressed this reviewers' concern, and demonstrate that upon fumarate accumulation, mtDNA-loaded MDVs are released from the mitochondria in a Snx9-dependent manner, to drive the innate immunity.

#R 3 Figure 2. MDVs elicited by fumarate contain TFAM. Left panel Representative confocal images of *cFh1^{fl/fl}* cells treated with 400 μ M MMF for 6 days. Mitochondria were labelled using anti-TOM20 and anti-PDH antibodies, and TFAM using anti-TFAM antibody. White arrows indicate TOM20⁺PDH⁺TFAM⁺ vesicles. Scale bar: 10 μ m. Right panel: Quantification of TOM20⁺PDH⁺ vesicles number positive for TFAM in *cFh1^{fl/fl}* cells treated with 400 μ M MMF for 6 days; n= 3 independent experiments.

#R 3 Figure 3. Colocalization of TFAM with DNA inside the MDVs elicited by fumarate. Representative Airyscan super-resolution images of *cFh1^{fl/fl}* cells treated with 400 μ M MMF for 6 days. Mitochondria were labelled using anti-TOM20 (magenta), anti-PDH (white) anti-TFAM (turquoise) and anti-DNA (yellow) antibody. White arrows indicate TOM20⁺PDH⁺DNA⁺TFAM⁺ vesicles. Scale bar: 5 μ m.

#R3 Figure 4. Characterization of the morphology of MDVs elicited by fumarate by lattice Structured illumination microscopy super resolution. Left panel: representative image of *cFh1^{fl/fl}* cells treated with 400 μ M MMF for 6 days. Mitochondria were labelled with anti-TOM20 and anti-PDH antibodies, and DNA with an anti-DNA antibody. Squares indicate examples of TOM20⁺PDH⁺DNA⁺ vesicles analysed. Scale bar: 5 μ m; magnification : scale bar: 1 μ m. Right panels represent the average maximal Feret's diameter and average area of MDVs; n=3 independent experiments.

We have included this information in the revised version of the manuscript (revised manuscript Fig. 4c, f-h and Extended Data Fig. 12a).

Specific comments:

(1) Authors: Figure- 1(c) Metabolites abundance (peak intensity, arbitrary units (A.U.) in wild-type control (Fh1+/+) and Fh1-deficient (Fh1adult mouse kidney measured by liquid-chromatography mass-spectrometry (LC-MS).

Reviewer: Fig 1-(c)- The change in fumarate (the most relevant metabolite) levels, does not appear to be significant, and with such a change, one would not expect the outcome suggested by the authors. While the other metabolites are affected, this is still not convincing. We agree with the claim that this mouse is an excellent model, but results must be refined.

We apologise for the ambiguity regarding this important point, and we are providing here a detailed answer to the reviewer's comment to clarify our results.

Upon loss of Fh1 activity, fumarate cannot enter the enzymatic reaction that converts it into succinate but, instead, reacts chemically with the thiol side chain within cysteine in proteins *via* a Michael addition reaction (Blatnik *et al.*, Diabetes 2008). This chemical modification that adds fumarate to cysteine residues is called succination. This non-enzymatic post-translational modification of cysteine also generates S-

2(succinyl)cysteine (2SC), which is considered a metabolic marker of loss of FH activity. Through the process of succination, fumarate alters protein function (Reviewed in Schmidt *et al.*, Semin Cell Dev Biol 2020). Another consequence of fumarate accumulation in the cytosol is the reversal of argininosuccinate lyase, a urea cycle enzyme, that results in the generation of argininosuccinate from arginine and fumarate (Zheng *et al.*, Cancer Metabolism 2013). Succination and reversal of argininosuccinate lyase (ASL) are adaptive mechanisms that allow the cell to limit the accumulation of fumarate to survive the mutation. Indeed, the inhibition of either pathway is lethal to cells (Reviewed in Schmidt *et al.*, Semin Cell Dev Biol, 2020). These reactions can be seen as “buffering tanks” that mop up the excess of fumarate. Consistently, with our new inducible model, both *in cellulo* and *in vivo*, we show that 2SC and argininosuccinate levels increase well before free fumarate is detected, and only when these “buffering tanks” are at near-saturation, we observe a more significant accumulation of fumarate in the tissue. Consequently, a modest increase in fumarate can still be accompanied by a strong downstream effect due to protein succination. A stronger accumulation of fumarate is seen later on (>>>day 10) in kidney tissue from our mouse model, and the higher levels observed in the different chronic models (Reviewed in Schmidt *et al.*, Semin Cell Dev Biol, 2020) reached upon several months of culture. We have amended the manuscript and provided a cartoon (R#3 Figure 5) below that has also been integrated into the manuscript (revised manuscript Extended Data Fig.1c) to illustrate the concept and better explain the sequence of events leading to 2SC, argininosuccinate and fumarate levels accumulation.

R#3 Figure 5: The chronology of Fh1 loss. In Fh1-deficient cells, fumarate enters a chemical reaction (Michael addition) whereby it is chemically added to thiol residues of proteins to form S-2(succinyl)cysteine (2SC). This modification alters protein function and is responsible for the oncogenic effect of the loss of FH activity. Only when succination is “at saturation” with the 2SC intracellular sinks (or “buffering tanks”) full can fumarate start to accumulate within the cell. The green and blue lines illustrate how the levels of 2SC and fumarate, respectively, may evolve with time within the cell from the moment of Fh1 loss of activity.

(2) Authors: Figure 1- (g) Expression levels of Interferon stimulated genes (ISGs) in Fh1^{-/-} vs Fh1^{+/+} kidney tissue at day 5 and 10 post-induction measured by qRT-PCR.

Reviewer: There is no reference in the text to the differences in the results between day 5 and day 10; on day 5 in most pro-inflammatory cytokines there is no significant different between WT and KO mouse. What is the possible explanation?

We thank the referee for raising this concern, which allows us to explain more in details the mouse model we used in this study. This newly generated mouse is a tamoxifen-inducible model (see revised Fig. 1a and related text for details), which allows recombination and thus the progressive loss of Fh1 levels in the adult animal. Our protocol involves 3 injections every other day (revised Fig. 1a). Upon Tamoxifen injection, the *locus* between exons 2 and 3 of *Fh1* is deleted. Over the course of the 3 injections, more Tamoxifen will accumulate, more cells within the tissues will be targeted and the recombination rate will

increase. There is a time lag before the effects of Fh1 deletion can be observed because, whilst the *locus* will be affected, some protein is still present within the cells. This explains why the effect *in vivo* is stronger at day 10 compared to day 5. In addition, secondary effects, such as activation of downstream signalling cascades and regulatory feedback loops are triggered later on. The molecular events we report here are a snapshot of the transcriptional landscape at days 5 and 10. A more refined picture would require many more time points. We hope that these more detailed explanations address the question about the difference between day 5 and day 10.

(3) Authors: Figure 2- (a) qRT-PCR (top) and representative immunoblots (bottom) showing expression levels of Fh1 in inducible *iFh1* epithelial kidney cell lines clones 29 (*iFh1*fl/flCL29) treated with either vehicle (ethanol; EtOH) or 4-hydroxytamoxifen (4-OHT) (*iFh*^{-/-}CL29) for the indicated period of time.

Reviewer: The Western blot in in EtOH, 3 days post induction shows a decrease in Fh1 levels even though it is a control and no influence should be observed.

We thank the reviewer for pointing what is likely to be a technical issue during the protein transfer. We have repeated the immunoblot and have inserted the new one in the revised version of the manuscript (R#3 Figure 6 and revised Fig. 2a).

R#3 Figure 6: Dynamics of *Fh1* deletion in *iFh1*^{fl/flCL29} cells. Representative immunoblots showing expression levels of *Fh1* in inducible *iFh1*^{fl/fl} epithelial kidney cell lines clones 29 (*iFh1*^{fl/flCL29}) treated with either vehicle (ethanol; EtOH) or 4-hydroxytamoxifen (4-OHT) (*iFh*^{-/-}CL29) for the indicated period of time.

(4) Authors: Figure 2-(e) Representative confocal images of mitochondrial morphology and DNA foci in *iFh1*fl/flCL29 cells treated with vehicle (EtOH) or 4-OHT (*iFh*^{-/-} CL29) for 24 hours at day 6, 10 and 15 post-induction. Mitochondria and DNA were labelled using anti-TOM20 and anti-DNA antibodies, respectively. White arrows indicate cytosolic DNA foci. Scale bar: 10 μm.

Reviewer: Fig 2-(e)- The authors need to demonstrate by FISH the presence of other mitochondrial genes beside TOM20 within the cytosol. It also important to characterize the physiological state of the cells and mitochondria, with respect to respiration and membrane potential.

Based on this comment and the other comment of the same reviewer, we realized that there is some ambiguity regarding this specific point in the manuscript and will attempt to clarify the issue below.

Following the observation of nucleic acid fragments in the cytosol of Fh1-deficient cells and given the striking morphology of Fh1-deficient mitochondria, we hypothesized that mitochondrial material may be leaking into the cytosol. We used digital PCR (ddPCR) to demonstrate and quantify the presence of mtDNA fragments of mitochondrial origin in the cytosol using primers that cover three different regions of the mitochondrial genome – namely the D-loop, Co3 and ND1 *loci*. Although these three *loci* do not cover the entirety of the mitochondrial genome, they are almost equidistant within the circular mtDNA molecule (and thus provide a good coverage; R#3 Figure 7) and the experiments we performed confirmed the accumulation of these mtDNA fragments into the cytosol of Fh1-deficient cells and under other conditions of fumarate accumulation.

R#3 Figure 7: mtDNA genes that were targeted in our ddPCR. Schematic showing the location of mitochondrial genome-encoded genes co3, ND1 and D-loop that were chosen as targets for quantification by digital droplet PCR (ddPCR).

The presence of whole mitochondrial genes, let alone the whole mitochondrial genome, is not necessary to elicit the response we observe; fragments of nucleic acids are sufficient to activate the cytosolic DNA sensors and the downstream Sting/Tbk1/Irf3 cascade. For example, Xian *et al.* (Immunity 2022) showed that FEN1-cleaved 500–650 bp fragments of oxidized mtDNA can exit mitochondria via mPTP- and VDAC-dependent channels to stimulate the activation of NLRP3 inflammasome. Released oxidized mtDNA fragments can also activate cGAS-STING signalling and stimulate a pro-inflammatory response. Thus, expanding our read-out to additional targets will not affect the conclusion of the experiment. The reviewer suggests using FISH to detect the presence of genes. Whilst this is a valid approach to the question, the technique we have used – ddPCR – is equally valid in addressing the question. By combining ddPCR on the cytosolic fraction and immunofluorescence showing the specific presence of mtDNA in the cytosol, we believe we confirmed the mitochondrial nature of the cytosolic DNA.

In addition, based on the reviewer's comment about TOM20, we feel it is necessary to amend the text to dispel any confusion with regards to the MDVs' cargo specificity. The cargo that is incorporated into MDVs is highly selective and can include mitochondrial nucleic acid fragments and specific proteins. For example, though both are mitochondrial outer membrane proteins, previous studies have showed that some vesicles carry MAPL but not TOM20 (Neuspiel *et al.*, Curr Biol 2008; Braschi *et al.*, Curr Biol 2010). Accordingly, the MDVs we characterised in our models of Fh1-deficiency or fumarate (MMF)-induced exhibit some cargo specificity whereby TOM20 is actually absent, while these MDVs are positive for the matrix protein PDH (see the quantification of TOM20⁻ PDH⁺ DNA⁺ vesicles in revised manuscript Fig. 4j,k and Extended Data Fig. 11n,o). Moreover, we have now shown additional mitochondrial proteins, which are present in these MDVs. As detailed in point 4 of Reviewer 2, the mtDNA-related protein, TFAM, is also present in the PDH⁺DNA⁺TOM20⁻ MDVs (revised manuscript Fig. 4f,g and Extended Data Fig. 12a), corroborating the presence of mtDNA in these MDVs released in the cytosol.

Finally, we thank the reviewer for pointing out the necessity to characterize the physiological state of the cells and mitochondria, with respect to respiration and membrane potential. Previous publications (Frezza *et al.*, Nature 2011; Sciacovelli *et al.*, Nature 2016) have already documented this for the chronic model (*cFh1^{-/-}CL1* and *cFh1^{-/-}CL19*). Following the reviewer's recommendation, we measured OCR in our inducible model (R#3 Figure 8) using the Agilent Seahorse XF Analyzer and showed a decrease in OCR consumption in our cells lacking Fh1 at all time points analysed. Membrane potential in the inducible cell lines was also visualized by microscopy using MitoTracker Red CMXRos (R#3 Figure 8). This showed reduced levels of mitochondrial potential in the Fh1-deficient cells vs control cells at day 15 after recombination.

We have amended the manuscript accordingly to incorporate this data (revised manuscript Extended Data Fig. 2a-c).

R#3 Figure 8. Further characterisation of the inducible Fh1-deficient cell model. (a). Schematic diagram illustrating the induction protocol in *iFh1^{fl/fl}* cell lines. (b). Oxygen Consumption Rate (OCR) in the inducible clones 29 and 33 at 3, 10, 15 and 21 days after treatment with either vehicle (ethanol; in blue; *Fh1^{fl/fl}* cells) or 4-OHT (in red; *Fh1^{-/-}*) measured using an Agilent Seahorse XF Analyzer. (c). MitoTracker Red CMXRos was used to visualise differences in membrane potential between vehicle- or 4OHT-treated *iFh1^{fl/fl}CL29* and *iFh1^{fl/fl}CL33* cells at day15.

Reviewer: Figures 2d and 3i- western blotting.

What are the differences between p-Tbk1 and Tbk1? This should be indicated in the figure legend (that it is the phosphorylated and unphosphorylated forms of each protein)

We thank the reviewer for pointing out this oversight. We have amended the figure legend accordingly: p-Tbk1 and p-Irf3 are detailed as phospho-Ser172 Tbk1 and phospho-Ser396 Irf3, respectively.

Authors: Figure 4. mtDNA is conveyed to the cytosol via mitochondrial-derived vesicles

Reviewer: Fig 4 - would be beneficial if the authors could comment on the average size of the vesicles and say more about their morphology.

We have now acquired additional images of the MDVs in MMF-treated cells using lattice NSIM super-resolution microscopy and have provided more information regarding their size, diameter and morphology in the revised version of the manuscript (see point 4 Reviewer 2 for detailed answer).

Fig 4A- The authors claim that “double membrane-bound low-density content vesicular structures protruding from mitochondria”. This a very odd conclusion since double membrane vesicles cannot be seen in the EM Fig as indicated by the authors.

We thank the reviewer for this comment and apologize for the misunderstanding in the manuscript. The reviewer is correct, that fully released vesicles are not observed in our TEM images. From our TEM pictures (**R#3 Figure 9** and Fig. 4a in the original manuscript), we refer to a double membrane structure with low-density content starting to emerge/bud from the mitochondria. This is typical from MDVs formation (Sugiura *et al.*, Nature 2017). In the figure below, we have highlighted the mitochondrial membrane (pink line); this indicates a continuous double membrane that includes the potential budding MDV. Nevertheless, we have modified the text accordingly to dampen our conclusions.

R#3 Figure 7. EM of Fh1-deficient cells. TEM images of Fh1-deficient mouse kidney tissue at day 10 post-induction highlighting the double mitochondrial membrane (pink line) around a potential budding MDV.

Referee #4 (Remarks to the Author):

In the manuscript entitled "Fumarate induces mtDNA release via mitochondrial-derived vesicles and drives innate immunity", Zecchini and authors show that loss of the tumor suppressor fumarate hydratase leads to increased mtDNA release and subsequent activation of interferon stimulated genes (ISGs). They create an inducible model of FH1 deletion to investigate the temporal dynamics of fumarate induction and the downstream effects on mitochondrial dynamics and innate immunity. They use epithelial cell lines derived from the mouse kidney for the majority of their studies and show that loss of fumarate increases the number of swollen and elongated mitochondria, the release of mitochondrial DNA via mitochondrial-derived vesicles, and the induction of the cGas-Sting-Tbk1 pathway. Mitochondrial-derived vesicles are highly understudied, and these findings indicate a new context and physiological role for these structures. While the authors have designed an inducible model to dissect the acute changes and immediate effects of FH1 loss, the timeline of events does not fully support the conclusions presented. The authors suggest that fumarate accumulation causes the effects on mitochondria, mtDNA release, and p-Stat1, but all these effects are seen between 3-6 days, after induced FH1 loss, while fumarate is not increased until much later – between days 10-15.

Later in their studies they add exogenous monomethylfumarate (MMF) which also affects mitochondria dynamics, mtDNA and pStat1 and ISGs. Does the dose of MMF correspond to the amount of fumarate at days 3-6, when there appears to be very little? Can the authors explain how fumarate is mediating these effects prior to its accumulation at days 3-6? Does the dose of MMF correspond to the amount of fumarate at days 3-6, when there appears to be very little?

We thank Reviewer 4 for their comments, and we hope that they will be satisfied by our new set of experiments addressing their concerns. We have incorporated in the response the figures as they appear now in the revised version of the manuscript. To help the review of this document, we have incorporated panels from the figures as they appear now in the revised version of the manuscript. Please, note that the order of the sub-panels may differ from the figure in the manuscript for presentation purposes.

Based on the reviewer's first comment, which is similar to the comment of Reviewer 2 specific point 1, we realise there might be some ambiguity regarding the effects of fumarate accumulation and the timing of events. This is indeed an important issue and we apologise for not making it clearer.

Upon loss of Fh1 activity, fumarate cannot enter the enzymatic reaction that converts it into succinate but, instead, reacts chemically with the thiol side chain within cysteine in proteins *via* a Michael addition reaction (Blatnik *et al.*, Diabetes 2008). This chemical modification that adds fumarate to cysteine residues is called succination. This non-enzymatic post-translational modification of cysteine also generates S-2(succinyl)cysteine (2SC), which is considered a metabolic marker of loss of FH activity. Through the process of succination, fumarate alters protein function (Reviewed in Schmidt *et al.*, Semin Cell Dev Biol 2020). Another consequence of fumarate accumulation in the cytosol is the reversal of argininosuccinate lyase, a urea cycle enzyme, that results in the generation of argininosuccinate from arginine and fumarate (Zheng *et al.*, Cancer Metabolism 2013). Succination and reversal of argininosuccinate lyase (ASL) are adaptive mechanisms that allow the cell to limit the accumulation of fumarate to survive the mutation. Indeed, the inhibition of either pathway is lethal to cells (Reviewed in Schmidt *et al.*, Semin Cell Dev Biol, 2020). These reactions can be seen as "buffering tanks" that mop up the excess of fumarate. Consistently, with our new inducible model, both *in cellulo* and *in vivo*, we show that 2SC and argininosuccinate levels increase well before free fumarate is detected, and only when these "buffering tanks" are at near-saturation, we observe a more significant accumulation of fumarate in the tissue. Consequently, a modest increase in fumarate can still be accompanied by a strong downstream effect due to protein succination. A stronger accumulation of fumarate is seen later on (>>>day 10) in kidney tissue from our mouse model, and the higher levels observed in the different chronic models (Reviewed in Schmidt *et al.*, Semin Cell Dev Biol, 2020) reached upon several months of culture. We have amended the manuscript and provided a cartoon (**R#4 Figure 1**) below that has also been integrated into the manuscript (**revised manuscript Extended Data Fig.1c**) to illustrate the concept and better explain the sequence of events leading to 2SC, argininosuccinate and fumarate levels accumulation.

R#4 Figure 1: The chronology of Fh1 loss. In Fh1-deficient cells, fumarate enters a chemical reaction (Michael addition) whereby it is chemically added to thiol residues of proteins to form S-2(succinyl)cysteine (2SC). This modification alters protein function and is responsible for the oncogenic effect of the loss of FH activity. Only when succination is “at saturation” with the 2SC intracellular sinks (or “buffering tanks”) full can fumarate start to accumulate within the cell. The green and blue lines illustrate how the levels of 2SC and fumarate, respectively, may evolve with time within the cell from the moment of Fh1 loss of activity.

The reason MMF is used here is that, unlike fumarate, the molecule is cell-permeable due to the presence of the methyl ester group, but this group also moderately increases the molecule’s reactivity with regards to the Michael addition reaction (Schmidt *et al.*, Bioorg Med Chem 2007; Kulkarni *et al.*, Nature Chemical Biology 2019). Thus, the dynamics of MMF will be different from that of endogenous fumarate produced by Fh1-deficient cells and exogenous fumarate treatment of the cells is a compromise between cell permeability and the reactivity of the molecule used. Importantly, however, the reactivity of MMF is much closer to that of fumarate than other permeable esters such as dimethyl fumarate (RA Kulkarni *et al.* 2019). In addition, we tested several doses of MMF (see the following point) and found 400 μM to be the lowest dose that resulted in an effect without showing toxicity to the cell. However, to address the referee’s point, we now provide new evidence with regard to the comparison between fumarate accumulated in the inducible model, and MMF treatment (R#4 Figure 2). Of note, we found that MMF leads to a comparable increase in fumarate and 2SC within the time frames tested (R#4 Figure 2). Furthermore, we found that MMF- leads to succination preferentially in the mitochondria (R#4 Figure 3). Overall, these new lines of evidence strongly support the use of MMF as a surrogate of fumarate in the context explored in this work.

R#4 Figure 2. Comparison of fumarate and S-(2-succinyl)cysteine (2SC) accumulation in the inducible Fh1 deletion model and MMF treatment. (c-d) Absolute concentration of fumarate (c) and 2SC (d) measured by LC-MC in *iFh1CL29^{fl/fl}* cells treated

with either vehicle or 4-OHT (right-hand side plots) or in cells treated with either vehicle or 400 μ M MMF for the indicated period of time (left-hand side plots).

R#4 Figure 3. MMF-mediated mitochondrial succination. 2SC immunofluorescence: representative confocal images of *cFh^{fl/fl}* treated with 400 μ M MMF or vehicle (DMSO) for the indicated period. Mitochondria were labelled using anti-TOM20, and succinated substrates were labelled with an antibody recognising 2SC modifications. Scale bar: 10 μ m.

Can low doses of MMF have similar effects to higher doses to recapitulate the effects of a small increase in fumarate? Because of this discrepancy, the data in the MMF treated cells is less convincing

To address this point, we have broadened range of MMF concentrations used to treat the cells, particularly lower concentrations (**R#4 Figure 4**) as suggested by the reviewer. Our new results show a clear dose-dependent effect on the transcriptional activation of target ISGs with the strongest effect at the higher dose (400 μ M). For most of the targets, the lower concentrations (25 μ M) show no or little effect. In line with the succination process detailed above, the lower dose is not sufficient to trigger a consistent response compared to higher doses. This data has now been incorporated into a new figure (**revised manuscript Extended Data Fig. 8I**) that combines the effect of lower doses of MMF on the ISG response and that of other metabolites (α KG and 2HG; see response to reviewer 1 point 5).

R#4 Figure 4. Effects of MMF on ISG are dose-dependent. (a). qRT-PCR showing the expression levels of our target ISGs in *Fh1^{Δ/Δ}* cells treated with a range of MMF, αKG or 2HG concentrations. A clear dose-dependent effect is observed with the higher doses resulting in a stronger ISG activation.

The authors should show more of the mechanism in the inducible *Fh1* KO lines including the EtBr experiment to eliminate mtDNA as well as the effects on mitochondrial derived vesicles.

We acknowledge this important point raised by the reviewer: Here, we provide additional experiments performed in the inducible *Fh1*-deficient model and characterize the time course of MDVs formation during the course of 15 days post deletion. We now show that TOM20⁺PDH⁺ MDVs are detectable early on at 3 days after deletion of *Fh1* (R#4 Figure 5 and revised manuscript Extended Data Fig. 12c,d), consistent with an early accumulation of 2SC and activation of inflammation (revised manuscript Extended Data Fig. 6a-d and 9a, respectively).

R#4 Figure 5. TOM20-PDH⁺ MDVs are detectable early on at 3 days after deletion of *Fh1*. Left panel: Representative confocal images of *iFh1^{Δ/Δ}CL20* cells treated with either vehicle (ethanol; EtOH) or 4-hydroxytamoxifen (4-OHT) (*iFh1^{Δ/Δ}CL20*) for the indicated period of time. Mitochondria were labelled using anti-TOM20 and anti-PDH antibodies, and DNA using an anti-DNA antibody. White arrows indicate TOM20⁺PDH⁺DNA⁺ vesicles. Scale bar: 10 μm. Right panel: Quantification of TOM20-PDH⁺DNA⁺ vesicles from (c); n=3 independent experiments.

The reviewer raises an important question about the presence of MDVs in the Rho 0 cell line we generated. Here, we present additional experiments performed in Rho 0 cells showing that the treatment with MMF in these cells didn't trigger the formation of TOM20-PDH⁺ vesicles (regardless of the presence of DNA) at any time point (R#4 Figure 6 and revised manuscript Extended Data Fig. 12e, f) suggesting the requirement of the presence of mtDNA for the formation of these specific vesicles. Therefore, we postulate

that alterations of mtDNA or its nucleoid structure caused by fumarate is at least in part responsible for the formation of MDVs.

R#4 Figure 6. mtDNA is required for fumarate-driven MDVs formation. Left panel. representative confocal images of *iFh1^{fl/fl};CL29^{p0}* treated with 400 μ M MMF for 1-8 days. Mitochondria were labelled using anti-TOM20 and anti-PDH antibodies, and DNA using an anti-DNA antibody. Scale bar: 10 μ m. Right panel: Quantification of TOM20-PDH⁺ vesicle number from (e). n=3 independent experiments

Finally, how does this mechanism contribute to the tumor suppressive function of FH1 in HLRCC? Additional experiments to address this point would strengthen the relevance of these findings

This is a very pertinent comment and we thank the reviewer for suggesting the additional experiments. Indeed, mutations in Fumarate Hydratase (FH) are associated with Hereditary Leiomyomatosis and Renal Cell Cancer (HLRCC), which can ultimately result in the development of an aggressive form of kidney cancer. In our study, we hypothesize that the chronic low-grade inflammation resulting from the early activation of the innate immune response may play a role in the development of tumours in FH-deficient kidney tissue.

We want to emphasise there that there are no *in vivo* models so far that recapitulate the type of tumours observed in HLRCC patients and the mouse model we used only develops premalignant cysts only in the late stage of the mouse life. Therefore, while working on generating a more suitable mouse model, we resorted to human data. First, GSEA analysis of a published expression profiling dataset (Ashrafian *et al.*, Cancer Research, 2010; GSE20896) of HLRCC vs Normal tissue shows a strong enrichment in the innate immune response (**R#4 Figure 7a**). Second, despite the paucity of HLRCC tumour material, we teamed with the group of Dr Maxine Tran to provide additional data showing a transcriptional activation of ISGs in tumour tissue from HLRCC patients vs normal tissue that is in line with the phenotype we report in the manuscript in across our models (**R#4 Figure 7b** and **revised manuscript Fig. 5**). Third, using a deconvolution method (<https://github.com/Danko-Lab/TED>) to determine the cellular composition of kidney tumour tissue across a panel of FH-deficient RCCs and other renal cancer subtypes including, ccRCC, papillary-type RCC (pRCC), SDH deficient tumours (SDH_renal), chromophobe RCCs (chRCC) and metabolically different chRCC (MD-chRCC) compared to normal tissue (Normal), we confirmed an elevated contribution in lymphocytes, reflecting an inflammatory microenvironment, in FH-deficient tumours but not in SDH-deficient tissues vs normal tissue (**R#4 Figure 7c**) in line with our hypothesis. Fourth, high levels of Interleukin-6 (IL6) have been shown to be present in the tumour microenvironment; thereby reflecting the strong association between inflammation and cancer (Kumari *et al.*, Tumour Biol., 2016; Hirano *et al.*, Int Immunol, 2021). Its overexpression has been reported in almost all types of tumours and it has been shown to play a role in the pathogenesis of chronic inflammatory diseases, autoimmune diseases and cancer. In line with this, we also observed a slight but significant elevation in IL6 and IL10 levels in the FH-deficient HLRCC tumours by ELISA (**R#4 Figure 7d**). This data is now incorporated in the revised version of the manuscript in the **revised Fig. 5**. Of note, as reported in a recent publication from the Linehan lab (Crooks *et al.*, Science Signalling, 2021), human HLRCC kidney tissue displays a mitochondrial phenotype that is identical to that we report in our study in Fh1-deficient animals. We now refer to this phenotype in the revised version of the manuscript. Altogether, this new dataset indicates that

some of the features we observe in our *in vivo* and *in vitro* models of Fh1-deficiency are recapitulated in human tumour tissue associated with loss of Fumarate Hydratase (FH), the human ortholog of mouse Fh1.

R#4 Figure 7: Inflammation in HLRCC. **a.** Lymphocyte contribution in FH-deficient RCCs and other renal cancer subtypes including, ccRCC, papillary-type RCC (pRCC), SDH deficient tumours (SDH_{renal}), chromophobe RCCs (chRCC) and metabolically different chRCC (MD-chRCC) compared to normal tissue (Normal). Top: Wilcoxon rank sum test, $p < 0.05$ Benjamini-Hochberg adjustment). Bottom: plot highlighting an increased lymphocyte contribution in FH-deficient tumours. **b.** Gene Set Enrichment Analysis of gene expression profiles of normal kidney (3 technical replicates) and a renal tumour (3 technical replicates) from a HLRCC patient carrying a germline mutation in the fumarate hydratase (FH) gene (Ashrafian *et al.*, *Cancer Res* 2010) showing enrichment in immune response pathways. NES=Normalised Enrichment Score. **c.** qRT-PCR showing the expression of FH and ISGs in tumour tissue from HLRCC patients vs surrounding healthy tissue. **d.** IL6 ELISA on human normal vs HLRCC tumour patient tissue. **e.** TEM of the mitochondria in FH-deficient tissue. The phenotype is similar to that observed in our mouse models.

Referee #5 (Remarks to the Author):

Zecchini et al. report that deletion of fumarate hydratase (FH) in mouse kidneys and murine kidney epithelial cell lines leads to an accumulation of fumarate, subsequent disruptions to mitochondrial morphology, and the release of mtDNA into the cytosol. They propose that cytosolic mtDNA engages the cGAS-STING signaling axis leading to elevated expression of type I interferon responses in FH-deficient cells in vitro and both interferon and pro-inflammatory responses in FH-deficient kidneys in vivo. Furthermore, the authors show that addition of exogenous monomethylfumarate (MMF) is sufficient to phenocopy the alterations to mitochondrial morphology, mtDNA release, and innate immune activation in vitro. Finally, they report that mtDNA release is not dependent on Bax-Bak or other canonical players, but instead occurs via the Snx9-dependent generation of mitochondrial derived vesicles (MDVs) Depletion of Snx9 is sufficient to mitigate mtDNA release into the cytosol and inhibit activation of the innate immune signaling.

Overall, this is an interesting study with robust datasets to support that FH deficiency triggers innate immune responses through mitochondrial stress. Although proposed by others, this paper also reveals that MDVs may be a mechanism of mtDNA release. Despite these positives, there are significant mechanistic gaps that must be further explored to solidify the proposed pathways. Specific comments follow below:

We thank the Reviewer to find our study of interest and supported by robust datasets, and by their relevant comments that we have addressed in details. We hope that the reviewer will be satisfied by the new version of our manuscript. To help the review of this document, we have incorporated parts of figures as they appear now in the revised version of the manuscript. Please, note that the order of the sub-panels may differ from the figure in the manuscript for presentation purposes.

Major Comments:

1. Fig 1: The FH conditional deletion model here provides a powerful tool to examine the effects of FH loss on immune signatures in a tissue specific manner. However, the authors do not provide any data from other organs.

[Text above redacted]

[Figure above redacted]

Do the levels of fumarate increase in other organs, and if so, is this correlated with increasing immune signatures?

Additionally, the authors state that there are no gross morphological changes in kidneys 10 days after tamoxifen-mediated Cre deletion of Fh1. A careful analysis of the H&E image of Figure 1 (d) seems to show many more nuclei between kidney cells. Moreover, even though FH expression is markedly down 5 days after tamoxifen exposure, the immune signature doesn't come up strongly until day 10.

Both of these lines of evidence are indicative of elevated kidney-infiltrating immune cells, as opposed to a kidney epithelial cell-intrinsic immune response.

Tissue RNAseq data will not distinguish kidney epithelial cell-intrinsic signatures of inflammation versus those generated by infiltrating immune cells. In agreement with the latter, the pathway analysis and gene expression signature seem to represent a generally more pro-inflammatory response as opposed to type I interferon. For example, IL-6, CD14, C3, TLR2, etc. are more consistent with a signature of macrophage/monocyte/neutrophil infiltration, as opposed to a cell-intrinsic type I interferon and interferon stimulated gene response.

Given that the conditional knockout approach relies on the ubiquitously expressed ROSA-ERCre that will target FH in many body cells and tissues, it is important that the authors do more work to define the in vivo mechanisms.

The reviewer is absolutely correct that tissue RNASeq will not distinguish kidney epithelial cell-intrinsic signatures of inflammation vs those generated by infiltrating immune cells. Indeed, to rule out these non-cell autonomous consequences of Fh1 loss we developed the epithelial kidney cell models, where we could confirm the cell intrinsic activation of innate immunity. Nonetheless, the reviewer's comment is very pertinent, and we addressed it as follows. First, we used a deconvolution software (<https://github.com/Danko-Lab/TED>, manuscript accepted for publication) to extract the potential contribution of infiltrating immune cells from our RNASeq mouse data. A detailed description of the method is provided in Materials and Methods in the revised version of the manuscript. This analysis showed that immune cells contributed to a very low fraction of the signature in both wild-type control (*Fh1*^{+/+}) and Fh1-deficient (*Fh1*^{-/-}) kidney tissues. In addition, no significant difference in lymphocytes or myeloid lineage contribution was observed between *Fh1*^{+/+} and *Fh1*^{-/-} tissue (R#5 Figure 2e). Second, we stained the kidney tissue for a marker of immune cells (CD14) (R#5 Figure 2f). Again, we observed no differences between *Fh1*^{+/+} and *Fh1*^{-/-} tissue, arguing against the immune signature reflecting recruitment of immune cells to the tissue. However, we believe that ultimately, some immune cells would be recruited to the Fh1-deficient tissue, as seen in human HLRCC, but that this would be a later event. At the early time point we investigated (day 10 post-induction), we are confident that the immune signature we observe originates from the Fh1-deficient kidney epithelial cells. We have now incorporated this data in the revised version of the manuscript in Extended Data Fig. 1e, f)

[Text above redacted]

R#5 Figure 2. Analysis of inflammation in Fh1-deficient kidney tissue. (e). We applied a deconvolution method (<https://github.com/Danko-Lab/TED>) to determine cellular composition of mouse kidney tissue at day 5 and day 10 post-induction. This approach indicated no significant differences in lymphocytes or myeloid lineage contribution between the control and Fh1-deficient mouse kidney tissues (pairwise comparison using Wilcoxon rank sum exact test and Benjamini-Hochberg p-value adjustment shows no significant differences between the control and Fh1-deficient mouse kidney tissues). (f). Immunohistological staining of *Fh1*^{+/+} vs *Fh1*^{-/-} mouse kidney tissue at day 10 post-induction. Left: 2SC staining showing an accumulation of 2SC in *Fh1*^{-/-} mouse kidney tissue (bottom) vs *Fh1*^{+/+} control animals (top) in line with our metabolomics data confirming the Fh1-deficient status of the tissue. Right: no difference in staining for CD14, a marker for myeloid cells, was observed between *Fh1*^{+/+} control and *Fh1*^{-/-} mouse kidney tissue.

2. The evidence in support of cGAS-STING-IFN-I pathway activation is generally not sufficient. Although crosses of FH cKO mice onto cGAS or STING KO mice to show ablation of the immune signature in vivo would be the most convincing, cGAS/STING inhibitors are readily available for mouse in vivo pre-clinical studies. These would be great to include as a way to link the in vivo phenotypes more cohesively with the cellular mechanisms presented in the later figures.

We thank the reviewer for this suggestion. Following their advice, we designed an *in vivo* study using a Sting inhibitor (**R#5 Figure 3a**). Mice that were treated with the combination of Tamoxifen and Sting inhibitor showed a dose-dependent reduction in ISGs expression. Noticeably Sting inhibitor H-151 had a strong effect, reducing the levels of the cytokines *Cxcl10*, *Ccl20* and *Ccl2* but a milder effect on other markers like *Ifim10*, *Ifi202b* and *Areg*. Although the levels of *Ifnb1* were decreased, the use of Sting inhibitor didn't fully abrogate its production. This suggested that other pathways could also contribute to the type I response, a point that we are now discussing in the manuscript and in other answers to this reviewer. This data has now been incorporated in the revised manuscript in **Fig. 3n** and **Extended Data Fig. 9g**.

R#5 Figure 3. Sting inhibition in *Fh1*-deficient animals. (n,g). *in vivo* Sting inhibition in *Fh1*-deficient animals. *Fh1^{fl/fl}* animals we treated with either vehicle (for control animals) or 2mg/animal Tamoxifen (3 doses every other day from day 0) to induce recombination and loss of *Fh1*. Animals also received a dose of vehicle or a low/high dose of Sting inhibitor H-151 (7mg/animal/day or 1.4mg/animal/day) every day up to day 9. Kidneys were collected at day 10. This generated four groups: *Fh1^{fl/fl}* control (●), *Fh1^{-/-}* (●), *Fh1^{-/-}* + 1.4mg/kg H-151 (●) and *Fh1^{-/-}* + 7mg/kg H-151 (●) animals. qRT-PCR was performed on the kidney tissue to determine the expression of various targets. As expected, *Fh1* expression was strongly reduced in animals treated with Tamoxifen vs vehicle controls. Expression of ISGs is shown in *Fh1^{-/-}* animals and is somewhat rescued in a dose-dependent manner in animals treated with Sting inhibitor H-151. Although the p-value is not <0.05 in all conditions, a downward trend is clearly seen for all ISGs measured. *Tfam* and *Hmgb1*, used as a negative controls, do not show any effect associated with H-151 treatment.

In addition, the authors do not sufficiently explain how the kidney epithelial cell lines were immortalized and derived. This is important because many immortalization/transformation procedures render the cGAS/STING pathway non-functional at several levels (see paper from Stetson et al (Science 2015) on SV40 IgT and other oncogenes downregulating STING signaling).

This is an important point and we thank the reviewer for pointing it out. Indeed, the Stetson lab identified the oncogenes of DNA tumour viruses, including E7 from human HPV and E1A from adenovirus, as potent and specific inhibitors of the cGAS-STING pathway (Lau et al., Science 2015). For immortalization of the kidney epithelial cell lines, we followed the protocol described in Mathew *et al.* (PMID: 18603117) and referred to it in the Material and Methods section. This protocol does indeed involve expression of E1A and a dominant negative mutant of p53. However, we would argue that the data presented in the manuscript indicate that the pathway is functional in our cell lines and can be activated as shown by the upregulation of *cGas* (revised manuscript Extended Data Fig. 9h) and its translocation in the cytosol (revised manuscript Extended Data Fig. 9b,c), the upregulation of Sting (revised manuscript Fig. 1f, Extended Data Fig. 3m), the phosphorylation of Tbk1 (revised manuscript Fig. 2d, Fig. 3j,m, Fig.4 l, Extended Data Fig. 7h, Extended Data Fig. 8j,k and Extended Data Fig. 9h), and its recruitment to the Golgi apparatus (revised manuscript Data Fig. 6g,h) (see new experiments presented in the next section). Moreover, the phenotype we report in our study was initially identified *in vivo* and treatment of *Fh1^{-/-}* animals with a Sting inhibitor, H-151, rescued ISGs upregulation to some extent (see point 2 above). We have also shown that Sting knockdown using siRNAs results in a decrease in the pathway activation both at the level of the signalling cascade *i.e.* phosphorylation of Tbk1 and Irf3 but also in the transcriptional activation of target ISGs in *iFh1^{-/-c129}* cell-lines and upon MMF treatment (revised manuscript Extended Data Fig. 9h and Fig. 3o, p respectively). Overall, these results should mitigate the concerns raised by the referee about an effect of immortalisation in dampening the signalling evoked by STING.

Finally, the authors never show that genetic ablation of cGAS/STING by siRNA, Crispr, etc is sufficient to ablate the interferon and ISG signature observed invitro. The RU.521 experiments of Figure 2 (m) and 3 (n) do not convincingly show that cGAS inhibition abolishes the induction of immune genes after FH deletion of MMF treatment. It would be much more convincing to target these pathways using siRNA or other means (the authors use cGAS, STING, Rig-I siRNAs in Extended figure 6 but never examine immune gene expression after knockdown in FH deficient or MMF treated cells). Given that mtRNA has also been noted as a ligand driving immune activation downstream of mitochondrial stress, it is important to rule out the mtRNA sensors Rig-I/Mda5 and perhaps other nucleic acid sensing TLRs. This could be easily accomplished using the cell lines and siRNAs reported in this paper.

We thank the reviewer for mentioning the possibility that other pathways involved in cytosolic nucleic acid recognition upstream TBK1/IRF3 signalling pathway could be involved in our mechanism. Indeed, recent publications have acknowledged that several PRR proteins can cooperate in the regulation of ISG responses (Zevini *et al.*, Trends Immun, 2017). Indeed, as the reviewer underlined here, both our cellular models and in the new *in vivo* experiments highlight the inability to completely rescue the ISG response observed with the cGas or Sting inhibitors, RU.521 or H-151, respectively. Therefore, we decided to investigate more broadly other PRR proteins potentially involved in the signalling of the ISG response. We conducted a siRNA targeted screen in the inducible model of *Fhl* deletion using phosphorylation of Irf3 as a read out, since it showed the strongest activation in our cellular models and is a common marker of most PRR driven type I inflammation. (R#5 Figure 4). This experiment revealed that in addition to silencing *cGas* and *sting*, siRNA directed against *Rig-I* was also able to mitigate Irf3 Phosphorylation; while silencing of *Mda5*, *Mavs*, *Ifi202b*, or *Tlr* receptors had no effect. Although this result does not completely rule out a role for Toll-like receptors in the immune response we observe, considering the strong effect of the combination of *Rig-I* and *cGas* silencing (see also R#5Figure 5), we chose not to investigate their potential role at this moment and decided instead to investigate further a potential role of RIG-I.

R#5Figure 4. PRR siRNA screen: Immunoblots of Irf3 phosphorylation in *iFhl^{1/CL29}* (lane 1, control siRNA scbl) compared to *iFhl^{-/-CL29}* (ko ind15) cells at 15 days post induction and transfected with indicated siRNAs.

We further confirmed these results in MMF-treated cells where we show that silencing *cGas* and *Rig-I* individually had a mild effect on Tbk1/Irf3 activation but strikingly, combined treatment of siRNAs against *cGas* and *Rig-I* or *cGas*, *Rig-I* together with *Sting* obliterated Tbk1/Irf3 activation (R#5 Figure 5 and revised manuscript Fig. 3o, p). Based on these results, we concluded that 1) when cGas is silenced, Rig-I may be able to compensate for the absence of the protein and vice versa; and 2) that Rig-I and cGas are the major regulators involved in the ISGs response we observe. RIG-I recognizes short viral double-stranded RNA, However, strikingly, the silencing of *Mda5* and *Mavs*, the other key components involved

in the RNA-mediated response, had no effect on Irf3 phosphorylation in our conditions (R#5 Figure 5). Although this does not exclude a possible role of mtRNA as the trigger of the cascade, our results also indicate that release of mtDNA is the major initiating factor suggesting a potential non-canonical function of Rig-I. These new results are now discussed in the manuscript.

R#5 Figure 5. Contribution of PPRs to innate immunity triggered by fumarate. (o). Immunoblots of indicated proteins in *cFh1fl/fl* cells incubated with either DMSO or MMF for 8 days and treated with combinations of indicated siRNA. (p). mRNA expression of a panel of ISGs in *cFh1fl/fl* cells incubated with either DMSO or MMF for 8 days and treated with combinations of indicated siRNA.

3. Comprehensive work from Dr. Frezza's lab and others has revealed fumarate as an oncometabolite in HLRCC via its actions on Keap1 and the NRF2 pathway, among others. Several recent papers have revealed that NRF2 pathway activation directly represses STING and counter regulates type I interferon responses (PMID: 30158636, 31487581, 31487581, 31487581). Fumarate accumulation after FH loss upregulates the NRF2 target Hmox1 in the cells used here (Extended Data Figure 2 (c)), so this suggests that NRF2 is activated. It is thus unclear how fumarate-dependent elevations in NRF2 would support robust STING activation. It is therefore important that the authors examine these connections and more thoroughly and explore innate immune pathways distinct from cGAS-STING that might trigger the fumarate-dependent expression of innate immune genes *in vitro* (and *in vivo*).

The reviewer raises an important point that deserves investigating in our model. Indeed, Olganier *et al.* and Sun *et al.* both showed that Nrf2 activation decreases STING expression by decreasing *Sing* mRNA stability. A more recent study in David Olganier's lab uncovered a role for Nrf2 as a negative regulator of the early innate immune response without affecting *Sting* mRNA or protein expression levels.

Using our *in vitro* models, the data presented in the manuscript show that phosphorylation of Tbk1, the downstream target of Sting, in Fh1-deficient cells does occur. Moreover, new experiments presented above with Sting inhibitor *in vivo* also confirmed that Sting is a component of the ISGs response we observed. However, from this data, we cannot rule out the possibility that the signal we observed could be stronger in

the absence of Nrf2. To test this hypothesis, we performed Nrf2 KD using siRNA in Fh1-deficient cell lines (R#5 Figure 6).

The immunoblot presented in R#5 Figure 6a shows that Nrf2 stabilization is indeed increased in *Fh1* KO but we could see only a modest decrease in Sting protein expression levels compared to control. As for the mRNA stability, rather than a reduction, the RT-qPCR in R#5 Figure 6b indicates a significant increase in *Sting* mRNA in *Fh1*^{-/-CL1} Fh1-deficient cells vs *Fh1*^{fl/fl} controls. In line with this observation, our *in vivo* RNASeq data shows a of 2.98-fold (p-val=0.051) upregulation of *Sting* mRNA in Fh1-deficient kidney tissue at day 10. Furthermore, *Nrf2* silencing did not increase Sting protein expression (compare lanes 1 and 2 in immunoblot R#5 Figure 6a) nor activation of the downstream cascade (as indicated by Tbk1/Irf3 phosphorylation) (R#5 Figure 6a). Finally, whilst *Nrf2* KD indeed resulted in a reduction in *Nqo1* expression, it did not result in a significant increase in the expression of different ISGs overall (R#5 Figure 6c), as would be expected if it inhibited the pathway. Thus, Nrf2 does not appear to act negatively on the activation of the pathway in our model. Our data clearly indicates a role for Sting in the activation of the innate immune response in our models, both *in vivo* and *in vitro*. However, we acknowledge that this result does not necessarily rule out a contribution of Sting-independent pathway(s). Indeed, Sting silencing using siRNA (see R#5 Figure 4) only results in a partial loss of ISGs activation and, although combined KD of *cGas*, *Sting* and *Rig-I* appears to enhance this rescue, this indicates that other pathways are likely to play a role in the response. Due to space constraints, we consider this part of the investigation not to be essential to the message we convey here, so we have not included the data in the manuscript. We will leave the referee and the editor to decide whether this is relevant to include.

R#5Figure 6. The role of NRF2 in inflammation in Fh1-deficient cells. (a). effect of Nrf2 KD on the activation of the Sting/Tbk1/Irf3 cascade. *Scr* or *Snx9* siRNA was transfected (details in the revised Materials and Methods section) in *Fh1*^{fl/fl} or *Fh1*^{-/-CL1} cells. Phosphorylation of Tbk1 and Irf3 was not affected; as were Sting protein levels. (b). qRT-PCR showing expression of *Nrf2* and downstream target *Nqo1* in *Fh1*^{fl/fl} or *Fh1*^{-/-CL1} cells treated with either *scr* or *Snx9* siRNA. The strong increase in *Nqo1* levels observed in *Fh1*^{-/-CL1} cells is, as expected, rescued following *Nrf2* KD. In addition, *Nrf2* KD does not result in *Sting* mRNA loss of stability as indicated by qRT-PCR (of note, *Sting* expression is actually upregulated in Fh1-deficient cells). (c). qRT-PCR showing no effect of *Nrf2* KD on ISG expression. *Tfam* expression is shown as negative control.

4. The Snx9-MDV pathway as a route for mtDNA release in FH deficient cells is interesting. It is unclear, however, exactly how membrane bound mtDNA would be sensed by cGAS in the cytosol in a cell-autonomous manner. Do the authors see cGAS around the PDH/DNA+ MDVs?

The reviewer raises an important point here. According to the literature, it is assumed that cGAS would only recognize free dsDNA in the cytosol and has never been shown to be recruited to a membrane bound organelle. Indeed, at this step, we can only speculate about how mtDNA present in MDVs ends up being released in the cytosol. It is possible that MDVs directly release the DNA in the cytosol as around 30% of the TOM20⁻ PDH⁺ MDVs we observed do not contain mtDNA (Extended Data Fig.11o). We are currently working on different hypothesis and hope to elucidate the full mechanism of cGAS access to mtDNA in the near future. Although we did not observe a striking colocalization of cGAS with MDVs, we report here a consistent retention of cGAS in the cytosol in MMF treated (42% cytosolic cGAS) vs control (18% cytosolic cGAS) cells, indicative of a cytosolic activation of cGAS (R#5 Figure 7).

R#5 Figure 7. MMF-mediated nucleus-to-cytoplasm translocation of cGAS. (b) Representative confocal images of cFh1fl/fl cells treated with 400 μ M MMF or vehicle (DMSO) for 8 days. cGAS and nucleus were labelled using an anti-cGAS antibody and DAPI staining, respectively. Scale bar: 10 μ m. (c) Quantification of cFh1fl/fl treated with 400 μ M MMF for 8 days showing the percentage of cells with cytosolic cGAS translocation.

Is it possible these vesicles might be released and fuse with neighboring cells in vitro (or immune cells in vivo), leading to a paracrine activation of cGAS? It may be interesting to consider supernatant transfer from FH deficient cells onto WT and cGAS knockdown cells to further explore/clarify the mechanisms of mt-nucleic acid sensing.

[Text above redacted]

Additionally, the authors show that knockdown of *Snx9* leads to a reduction in cGAS/STING pathway activation. Does this inhibition also lead to a reduction in expression of the ISGs and pro-inflammatory cytokines and chemokines reported in Figure 1 (f,g)?

We thank the reviewer for pointing out this crucial control. We believe our data clearly shows a reduction in the Tbk1-Irf3 pathway activity (**revised manuscript Fig. 4l**) and *Ifnb1* (**revised manuscript Fig. 4m**) following *Snx9* knockdown. ISGs expression is modulated by complex and intricate regulatory feedback loops (both positive and negative), often regulated at the post-translational level, acting at different times for different targets that make it difficult to assess this response as a whole at a single time point. In our manuscript, we observe different expression dynamics for each of our transcriptional targets that is likely to reflect such a complex transcriptional regulation. The phosphorylation of the downstream elements of the cascade *i.e.* Tbk1 and Irf3 might arguably constitute a more stable and reliable read-out of the pathway activation.

To address the referees' request, we knocked down *Snx9* expression using siRNA in cells treated with 400 μ M monomethyl fumarate (MMF) for 72 hours. RT-qPCR was then used to assess the impact of the loss of *Snx9* on ISGs expression at this specific time point. The results are reported below (**R#5 Figure 9**) and

[Figure and text above redacted]

show a range of dysregulated targets upon *Snx9* loss, corroborating the specificity and critical role of *Snx9*-dependent mtDNA-loaded MDVs in the inflammation answer observed. This data is now included in revised ED Fig. 12g.

R#5 Figure 9. ISG in MMF-treated cells. MMF- or vehicle-treated cells (72hrs treatment) were treated with *Snx9* or scramble (scr) siRNA and the expression of a panel of ISGs and control genes was determined using qRT-PCR.

Similarly, does *Snx9* knockdown have any effect on mitochondrial morphological changes?

Our results show a correlation between mitochondria swelling and mtDNA release. However, we have shown that preventing mitochondria fusion or fission using siRNA against *Drp1*, *Opa1*, *Mfn1*, *Mfn2* does not affect mtDNA release. We now present quantification of mitochondrial morphology monitoring the effect of the silencing of *Snx9* in the Fh1-deficient cells. Our results confirmed that loss of Fh1 leads to mitochondrial morphology changes (enlarged and elongated) characterized by a decreased number of mitochondria and increased area compared to control cells. While silencing of *Snx9* in Fh1-competent control cells reduced mitochondrial number but not size (R#5 Figure 10 and Extended Data Fig. 11e,f), suggesting a mild effect leading to mitochondrial elongation, loss of *Snx9* does not rescue mitochondrial morphology in Fh1-deficient cells. Overall, this suggests that mitochondria swelling could happen independently of *Snx9*-dependent mtDNA release.

R#5 Figure 10: Effect of Snx9 knock down on mitochondrial morphology. Quantification of mitochondrial number and area in indicated cFh1 cell lines transfected with scramble (scr) or Snx9 siRNA; n=3 independent experiments.

5. The authors claim in the discussion that this paper represents the first report of a disease relevant mutation causing release of the mtDNA into the cytosol. This is a significant overstatement. Recently, human mutations in ATAD3A causing bona fide mitochondrial disease have been shown to liberate mtDNA into the cytosol and engage cGAS. Moreover, loss of TDP-43, CLPP, YME1L, OPA1 and others, all of which are linked to human disease, induce mtDNA release and activation of innate immunity. Perhaps the authors meant to restrict their discussion of impact to kidney diseases, but even so, loss of Tfam in kidney tubule cells has been shown to induce mtDNA leakage and cGAS-STING inflammation *in vivo* and *in vitro*.

We thank the reviewer for pointing this out and apologise for the oversight. The point we sought to make was that it is the first time that an alternative mechanism for mtDNA release other than involving mitochondria loss of integrity, and therefore compatible with cell survival and a persistent immune signature, is proposed. We have amended the text and references accordingly.

There is no evidence provided here to link the mtDNA release and immune phenotypes shown to HLRCC, so I believe it would be prudent to tone down these statements of novelty and properly reference the literature. It will not decrease the impact of the author's findings to reference these recent papers.

We understand the reviewer's comment and will modify the text accordingly. Nevertheless, we now provide additional data obtained from human patients showing a similar immune response signature and cytokines production profile in HLRCC tumours to that seen in our different mouse models (**R#5 Figure 11** and **revised manuscript Fig. 5a-d**). We hope that our findings will catalyse an in-depth characterisation of the role of fumarate-dependent inflammation in the aetiology of HLRCC, and its progression.

R#5 Figure 11: Inflammatory signature in FH-deficient tumours. (a) Gene Set Enrichment Analysis of gene expression profiles of normal kidney (3 technical replicates) and a renal tumour (3 technical replicates) from a HLRCC patient carrying a germline mutation in the fumarate hydratase (FH) gene (Ashrafiyan *et al.*, *Cancer Res* 2010) showing enrichment in immune response pathways. NES=Normalised Enrichment Score. (b). qRT-PCR showing the expression of FH and ISGs in tumour tissue from HLRCC patients vs surrounding healthy tissue. n=3 individual patients, in three technical replicates. (c) Cellular composition of the bulk RNA sequencing datasets from FH deficient RCCs, SDH deficient RCCs and common RCC subtypes from TCGA (as described in Materials and Methods). (d). IL6 and IL10 ELISA on human normal vs HLRCC tumour patient samples. N=Normal (healthy patients), T=Tumour (HLRCC patients). N=5 and 20 patients serum samples for N and T, respectively.

Technical Comments:

1. A significant portion of the analyses herein rely on manual scoring of immunofluorescence images. While manual scoring itself is not itself problematic, the individual(s) conducting the scoring should be blinded to the experimental groups wherever possible in order to minimize the impact of unintentional bias in the results. This is particularly important for subjective measures, such as mitochondrial morphology, where there is no clearly defined delineation between different morphological classifications.

We appreciate the reviewer's concerns, and we wish to explain in detail here why we choose to report the mitochondrial morphology phenotype this way: first, due to the unusual mitochondrial morphology observed in these cells, we wanted to provide a comprehensive assessment. In addition, with our classification, we wanted to describe the heterogeneity of the mitochondrial network in different cells, which wouldn't otherwise be reflected by assessing average size of mitochondria.

Nevertheless, we now provide new unbiased quantification of mitochondria area and number (**R#5 Figure 12**) using Fiji software calculated from randomized region of interest analyses (see Material and Methods section). These new results show that the loss of Fh1 induces a decrease of mitochondrial count mirrored by an increase of mitochondrial area., corroborating the changes in the mitochondrial network in *cFh1^{CL1}* and *cFh1^{CL19}* (**revised Extended Data fig. 3f**), *iFh1^{-CL29}* and *iFh1^{-CL33}* (**revised Fig. 2i, j** and **Extended Fig. 2j,k**) and in MMF treated cells (**revised Fig3c,d**).

R#5 Figure 12: Mitochondrial morphology quantification with Fiji: **f**. Quantification of mitochondrial number (left) and area (right) in indicated *cFh1* cell lines from. **i,j**. Quantification of mitochondrial number (i), and area (j) in *iFh1^{β/βCL29}* cells treated with either vehicle (EtOH) or 4-OHT (*iFh1^{-/-CL29}*) for the indicated period of time. **J,k**. Quantification of mitochondria number (j), and area (k), in untreated and *iFh1^{β/βCL29}* cells treated with either vehicle (EtOH) or 4-OHT (*iFh1^{-/-CL29}*) for the indicated period of time. **c,d**. Quantification of mitochondrial number (c), and area (d) in untreated, vehicle (DMSO) or MMF-treated *cFh1^{β/β}* cells. n=3 independent experiments for each panel

2. Figures 3 (f-h, j, n), 4 (j) and Extended Data Figures 3 (j-l), 4 (i-l), 5 (a, h-k) For these figures, each individual replicate for the control group has been set to one. This has the effect of artificially reducing the variance of the control group to zero, resulting in an inflated level of significance for groups that are being directly compared to the controls. Control values should be normalized to the mean of the control group, rather than set to one on an individual basis, preserving the variance and allowing an accurate significance level to be calculated.

We understand the concern of the reviewer and we want to reassure here that we didn't manipulate the data to change the significance of our results. The digital droplet PCR experiments assess cytosolic mtDNA copy after a brief cell extraction with digitonin and we define mtDNA release by the fold change observed in different samples to this baseline. Thus, we would argue that Fold Change to control is the biological value we want to report here. We report below the statistics as they would be calculated plotting each control value (R#5 Figure 13). Also, we have documented the absolute copy number data in the raw data section.

R#5 Figure 13: Quantification of absolute mitochondrial copy number by Digital Droplet PCR in cytosolic fractions of indicated samples: In indicated *cFhl* cell lines (Upper panel) and *cFhl* ^{Δ/Δ} cells treated with vehicle (DMSO) or the indicated concentration of MMF during 8 days (lower panel). n=3 independent experiments for each panel. Results are expressed as copy number of analysed mitochondrial fragment/ μ g of protein. P-values were determined by one-way ANOVA with Tukey's multiple comparison test.

4. Figures 3 (f-h), Figure 4 (j) and Extended Data Figures 2 (a), 4 (i-k) – The figure legend indicates that n = 3, but there are 4 replicates visible in the graphs.

We thank the reviewer for pointing out this. We have amended accordingly in the figure legend.

5. Line 255-257 – Authors state that silencing of *Snx9* reduces number of cytosolic DNA foci, but Extended Data Figure 7 (o) does not indicate a significant difference between scr and *Snx9* siRNAs.

We omitted to indicate it on the graph. MMF 6 days scramble (scr) siRNA vs MMF 6 days *Snx9* siRNA: p<0,0001.

6. Extended Data Figure 7 (f-o) – Figure legend descriptions do not match corresponding figure panels.

We thank the reviewer for pointing out this oversight. We have amended accordingly in the figure legend.

7. Figure 3 (d,e) and Extended Data Figures 3 (h,i), 5 (f,g) – Each of these pairs of figures seem to present the same data (e.g., same means, same p values), just with two slightly different figure formats. What is the rationale for this duplication of data?

We meant to show the distribution of number of *foci* in each cell. We acknowledge the redundancy of these graphs and now show one graph representing distribution and mean.

Reviewer Reports on the First Revision:

Referees' comments:

Referee #1 (Remarks to the Author):

I am satisfied!

Referee #3 (Remarks to the Author):

After reading the response of the authors to my comments I conclude that the authors have satisfyingly addressed all my concerns. With regard to the response of the authors to comments of the other reviewers, I again think they have addressed all the concerns and have done an excellent job. I recommend that this manuscript be accepted for publication in Nature.

Referee #4 (Remarks to the Author):

The authors have provided a substantial amount of new data and I can appreciate the amount of work that has gone into this revision. It is still unclear to me how these effects are mediated by fumarate if fumarate is not yet accumulated when the key phenotypes are observed (i.e. phospho-IRF3, cytosolic DNA fragments, and induction of MDVs). The MMF experiments do not clarify this point as the addition of MMF causes an increase in fumarate. The new schematic presented (Extended data Figure 1C animation) suggests that the early (and most important) phenotypes are not mediated by fumarate itself but instead by the adaptive buffering response to increased fumarate, perhaps the increased 2SC. It seems the more appropriate experiments would include manipulation of 2SC instead of MMF. The MMF experiments would only make sense with a dose that causes an increase in 2SC but no change in fumarate as that would best mimic the induced loss of Fh1. The results are very striking, but the mechanisms is still unclear as to what is mediating which phenotypes. Clearly MMF has an effect, but there is also clearly an effect before fumarate is accumulated. The data shown do not differentiate these two mechanisms, nor does the data show they are one in the same.

Referee #5 (Remarks to the Author):

I appreciate the authors' thorough responses to my questions and those of other reviewers. The authors provide much new data that solidifies key mechanistic details related to mtDNA release and sensing. The MDV story is very nice and novel. I am intrigued by the involvement of RIG-I, but not MAVS, in mediating innate immune responses in Fh1-/- and MMF treated cells. AT-rich cytoplasmic DNA can be transcribed into RNA for activation of RIG-I, so perhaps this is occurring to provide cytosolic dsRNA ligands that synergize with mtDNA/cGAS/STING. However, the RIG-I response should require MAVS, yet the supplemental response Figure 4 shows no role for MAVS. There are some earlier reports of RIG-I/MAVS/STING interactions upon overexpression, but it is my

understanding that this complex is not a major driver of IFN and NF- κ B signaling in physiological conditions. However, it might be nice to cite these reports and add a few lines to the discussion. Given this unique RIG-I-dependent but MAVS-independent result, I request that the authors include the the p-IRF3 data from response Fig. 4 to the paper, and also add MAVS siRNA knockdown experimental data to Fig. 3 or ED Fig. 9. I think it is important to clarify that this is a unique response downstream of FH inhibition and fumarate accumulation, which reveals unexpected complexity in the innate immune response influenced by mitochondrial metabolites and mtDNA release. After incorporating these data, references, and discussion points, I will support publication of the paper in Nature.

Author Rebuttals to First Revision:

Response to the referees Zecchini, Paupe et al

Referee: plain black text

Authors: plain blue text

Referee #4 (Remarks to the Author):

The authors have provided a substantial amount of new data and I can appreciate the amount of work that has gone into this revision. It is still unclear to me how these effects are mediated by fumarate if fumarate is not yet accumulated when the key phenotypes are observed (i.e. phosphor-IRF3, cytosolic DNA fragments, and induction of MDVs). The MMF experiments do not clarify this point as the addition of MMF causes an increase in fumarate. The new schematic presented (Extended data Figure 1C animation) suggests that the early (and most important) phenotypes are not mediated by fumarate itself but instead by the adaptive buffering response to increased fumarate, perhaps the increased 2SC. It seems the more appropriate experiments would include manipulation of 2SC instead of MMF. The MMF experiments would only make sense with a dose that causes an increase in 2SC but no change in fumarate as that would best mimic the induced loss of Fh1. The results are very striking, but the mechanisms is still unclear as to what is mediating which phenotypes. Clearly MMF has an effect, but there is also clearly an effect before fumarate is accumulated. The data shown do not differentiate these two mechanisms, nor does the data show they are one in the same.

We thank the reviewer for assessing our revised manuscript critically and appreciating our efforts to address their comments. The referee points out that “*the new schematic presented (Extended data Figure 1C animation) suggests that the early (and most important) phenotypes are not mediated by fumarate itself but instead by the adaptive buffering response to increased fumarate, perhaps the increased 2SC*”. Based on this comment, we need to clarify some important aspects of our findings.

First, we would like to elaborate on an important point regarding succination, which may not have been sufficiently clear in our earlier response. In FH-deficient cells, fumarate cannot be converted into malate and we surmise that the loss of Fh1 leads to its increased availability. On one hand, this initial fumarate accumulation leads to the formation of the metabolite 2SC, i.e. the adduct between fumarate and free cysteine, which can be detected by mass spectrometry. On the other hand, fumarate can lead to the succination of cysteine residues of proteins that can be visualised using the dedicated anti-2SC antibody. Based on our current model, in FH-deficient conditions, fumarate would only start to accumulate, as measured by mass spectrometry, once a significant pool of available cysteine residues has undergone succination, and other buffering systems have reached saturation. This model is corroborated by the observation that *in vivo*, at day 5 upon Fh1 loss, even though fumarate levels are not yet significantly up by that time 2SC is increased. Thus, fumarate-mediated effects of FH loss on protein can appear before fumarate accumulates in the cell. Yet, increased succination arises from the dysregulated homeostasis of fumarate levels, so we postulate that the effects are *de facto* mediated by fumarate. We have amended the text to clarify this point and have been careful in distinguishing between free 2SC and protein succination.

In reply to the point as to whether more appropriate experiments would include manipulation of “*2SC instead of MMF*”, we want to emphasise that succination, and therefore the production of 2SC, are inescapably linked and can only be achieved *in vivo* or *in vitro* by the addition of (exogenous or endogenous) fumarate (or MMF) to cysteine thiol residues. It is, therefore, technically not possible to manipulate protein succination without first increasing fumarate. Yet, we agree with the reviewer that our study does not provide definitive evidence to dissect the effects of fumarate accumulation from fumarate-driven succination, and which fumarate by-product is responsible for the observed phenotype. We have now amended the text and discussion to mitigate any ambiguity about our conclusions.

Referee #5 (Remarks to the Author):

I appreciate the authors' thorough responses to my questions and those of other reviewers. The authors provide much new data that solidifies key mechanistic details related to mtDNA release and sensing. The MDV story is very nice and novel. I am intrigued by the involvement of RIG-I, but not MAVS, in mediating innate immune responses in *Fh1*^{-/-} and MMF treated cells. AT-rich cytoplasmic DNA can be transcribed into RNA for activation of RIG-I, so perhaps this is occurring to provide cytosolic dsRNA ligands that synergize with mtDNA/cGAS/STING. However, the RIG-I response should require MAVS, yet the supplemental response Figure 4 shows no role for MAVS. There are some earlier reports of RIG-I/MAVS/STING interactions upon overexpression, but it is my understanding that this complex is not a major driver of IFN and NF-κB signaling in physiological conditions. However, it might be nice to cite these reports and add a few lines to the discussion. Given this unique RIG-I-dependent but MAVS-independent result, I request that the authors include the the p-IRF3 data from response Fig. 4 to the paper, and also add MAVS siRNA knockdown experimental data to Fig. 3 or ED Fig. 9. I think it is important to clarify that this is a unique response downstream of FH inhibition and fumarate accumulation, which reveals unexpected complexity in the innate immune response influenced by mitochondrial metabolites and mtDNA release. After incorporating these data, references, and discussion points, I will support publication of the paper in Nature.

We thank the Reviewer for their positive comments on our revised manuscript and for supporting the publication of our manuscript upon minor modifications.

We agree with Reviewer 5 on the unexpected nature of our new results suggesting a Rig-I contribution to the inflammation observed but in a MAVS-independent manner. Based on Reviewer's comments and suggestions, we have now added to the manuscript the immunoblots showing no effect of MAVS-silencing on p-Irf3 levels upon MMF treatment, together with the graph showing the efficiency of the siRNA used to silence *Mda5* and *Mavs* expression (**NEW Extended Data Fig. 9h**). We also have amended the results and discussion sections, by suggesting a potential Mavs-independent Rig-I activity in our model and incorporated relevant additional references asked by the Reviewer.

Reviewer Reports on the Second Revision:

Referees' comments:

Referee #4 (Remarks to the Author):

The authors have addressed my concerns in this revised version of the manuscript.

Referee #5 (Remarks to the Author):

The authors have responded to my final critiques. I strongly support publication in Nature.